# Structural basis for RAD18 regulation by MAGEA4 and its implications for RING ubiquitin ligase binding by MAGE family proteins

Simonne Griffith-Jones [ID][1], Lucía Álvarez [ID][2,4], Urbi Mukhopadhyay[1,4], Sarah Gharbi [ID][1], Mandy Rettel [ID][2], Michael Adams[1], Janosch Hennig [ID][2,3] & Sagar Bhogaraju [ID][1✉]

## Abstract

**MAGEA4 is a cancer-testis antigen primarily expressed in the testes but aberrantly overexpressed in several cancers. MAGEA4 interacts with the RING ubiquitin ligase RAD18 and activates trans-lesion DNA synthesis (TLS), potentially favouring tumour evolution. Here, we employed NMR and AlphaFold2 (AF) to elucidate the interaction mode between RAD18 and MAGEA4, and reveal that the RAD6-binding domain (R6BD) of RAD18 occupies a groove in the C-terminal winged-helix subdomain of MAGEA4. We found that MAGEA4 partially displaces RAD6 from the RAD18 R6BD and inhibits degradative RAD18 autoubiquitination, which could be countered by a competing peptide of the RAD18 R6BD. AlphaFold2 and cross-linking mass spectrometry (XL-MS) also revealed an evolutionary invariant intramolecular interaction between the catalytic RING and the DNA-binding SAP domains of RAD18, which is essential for PCNA mono-ubiquitination. Using interaction proteomics, we found that another Type-I MAGE, MAGE-C2, interacts with the RING ubiquitin ligase TRIM28 in a manner similar to the MAGEA4/RAD18 complex, suggesting that the MAGEA4 peptide-binding groove also serves as a ligase-binding cleft in other type-I MAGEs. Our data provide new insights into the mechanism and regulation of RAD18-mediated PCNA mono-ubiquitination.**

**Keywords** Ubiquitin; RAD18; MAGEA4; RAD6; Trans-lesion DNA Synthesis
**Subject Categories** Post-translational Modifications & Proteolysis; Structural Biology

## Introduction

Protein ubiquitination is a conserved post-translational modification, exclusive to eukaryotes, that regulates numerous cellular processes including protein turnover, cell cycle, DNA repair, intracellular trafficking and innate immunity (Pickart, 2001). Ubiquitination is mediated by a three-enzyme cascade (E1, E2, E3) that begins with the attachment of ubiquitin (Ub), a 76 amino acid protein, to E1. E1 then transfers Ub to a catalytic cysteine within E2, which subsequently catalyses the formation of an isopeptide bond between Ub and a lysine within the substrate. Finally, E3 ligases bring specificity to the reaction and mediate the transfer of Ub to the substrate (Pickart, 2001). Of the ~600 E3 ligases, really interesting new gene (RING) E3 ligases represent the largest family and are characterized by the presence of the RING domain (Deshaies and Joazeiro, 2009). RING E3 ligases do not contain a catalytic cysteine residue, unlike HECT E3 ligases, another major family of E3s (Yang et al, 2021). Instead, RING E3 ligases mediate ubiquitination by binding to both the E2 loaded with Ub (E2~Ub) and substrate and bringing them into close proximity (Yang et al, 2021).

Ubiquitination plays a key role in orchestrating a series of responses to damaged DNA. UV induced DNA damage can lead to bulky lesions, for example cyclobutane pyridimine dimers (Mouret et al, 2006), which cannot be accommodated by replicative polymerases, causing the replication fork to stall and subsequently causing double-stranded breaks, chromosome instability or cell death (Andersen et al, 2008). Therefore, eukaryotes have evolved mechanisms to bypass damaged DNA, termed DNA damage tolerance (DDT). The accumulation of single-stranded DNA resulting from replication fork stalling recruits the RING E3 ubiquitin ligase RAD18, in complex with the E2 RAD6 (Davies et al, 2008; Notenboom et al, 2007). RAD18/RAD6 monoubiquitinates the DNA sliding clamp PCNA, inducing a polymerase switching event in which the replicative polymerase is substituted for a member of the Y family of polymerases (Polη, Polκ, Polι and REV1) (Kannouche et al, 2004). Y family polymerases are able to facilitate trans-lesion synthesis (TLS) past bulky lesions and allow replication to resume in an error-prone or error-free manner, depending on the polymerase (Prakash et al, 2005). RAD18 and RAD6 are both upregulated following exposure to DNA damaging agents (Jones and Prakash, 1991). RAD18 levels in the cell are tightly controlled in order to prevent PCNA monoubiquitination and the recruitment of error-prone Y-family polymerases in the

[1]European Molecular Biology Laboratory, 71 Avenue des Martyrs, 38042 Grenoble, France. [2]European Molecular Biology Laboratory, Meyerhofstraße 1, 69117 Heidelberg, Germany. [3]Biochemistry IV, Biophysical Chemistry, University of Bayreuth, Universitätsstrasse 30, 95447 Bayreuth, Germany. [4]These authors contributed equally: Lucía Álvarez, Urbi Mukhopadhyay. ✉E-mail: bhogaraju@embl.fr

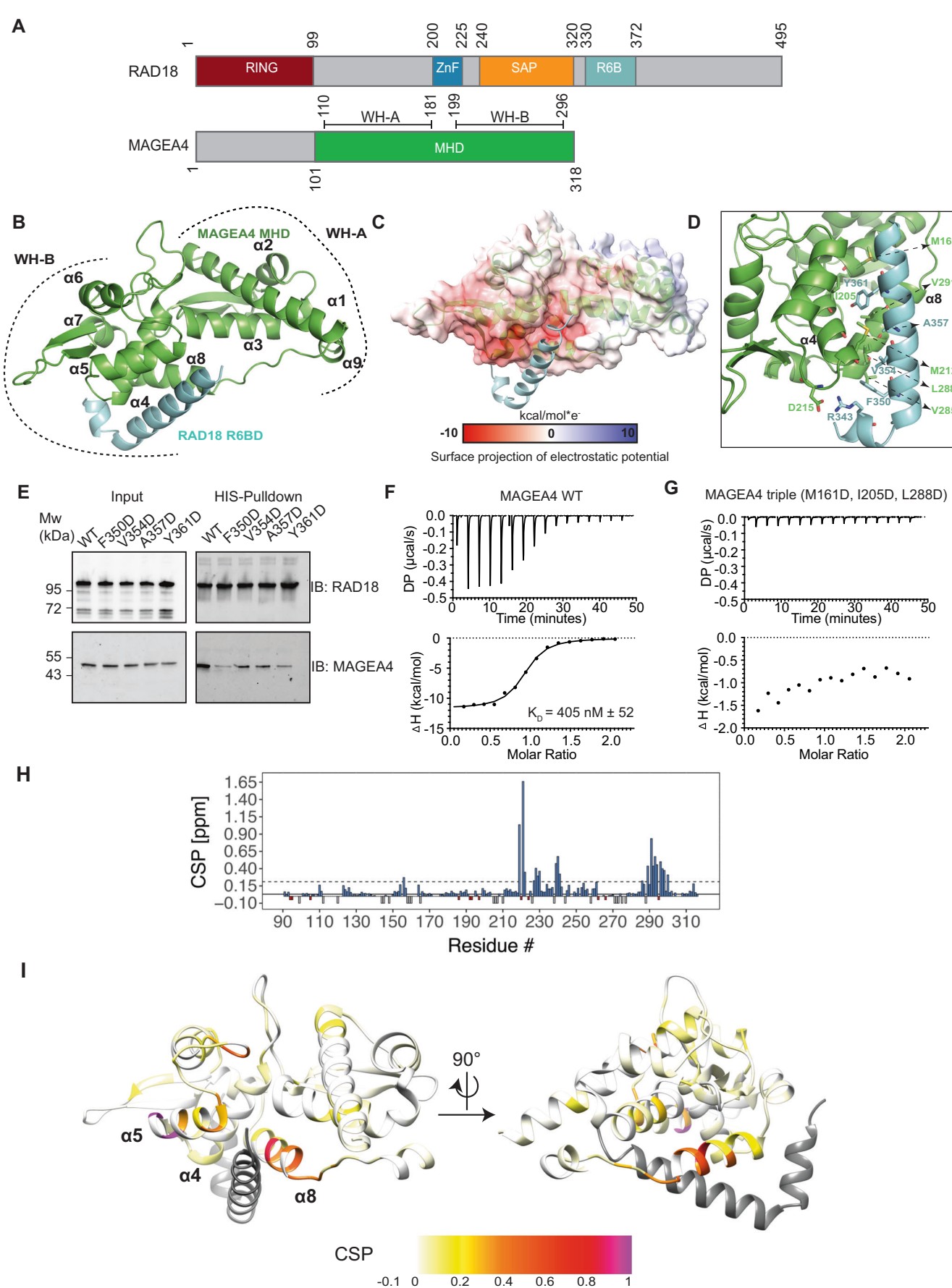

◄ **Figure 1. The RAD18 R6BD binds to the C-terminal WH-B motif of MAGEA4.**

(A) Domain organization of RAD18 (top) as inferred from the AlphaFold model of full-length RAD18 (Appendix Fig. S1). The domain organization of MAGEA4 is shown below as inferred from the crystal structure of the MAGEA4 MHD (PDB: 2WA0). (B) AlphaFold2 model of the MAGEA4 MHD (green) bound to the RAD18 R6BD (cyan). (C) Electrostatic potential mapped on to the surface of the MAGEA4 MHD bound to the RAD18 R6BD. (D) Close-up of the MAGEA4/RAD18 interaction interface, showing the binding residues involved. (E) HIS-tagged wild type (WT) and mutant RAD18 proteins were co-expressed with WT untagged MAGEA4. Cleared lysates were subjected to HIS pulldown using Talon beads. Both cleared lysates and Talon-bound proteins were subjected to SDS-PAGE followed by western blotting using the indicated antibodies. (F) ITC experiment performed between MAGEA4 WT and a 27 amino acid peptide corresponding to the RAD18 R6BD to measure the binding affinity. The ITC experiment was performed three times with similar results. (G) ITC experiment performed between MAGEA4 Triple mutant (M161D, I205D, L288D) and the RAD18 R6BD peptide to measure the binding affinity. The ITC experiment was performed three times with similar results. (H) Histogram of the chemical shift perturbations for MAGEA4 MHD upon binding to RAD18 R6BD. The CSPs were determined at a protein:peptide ratio of 1:3. The continuous line shows the average of CSPs over all residues and the dashed line indicates the value of the average plus one standard deviation. Residues for which there was no assignment in both free and bound bars are shown in grey, whereas residues only assigned to the free state are shown in red. (I) MAGEA4/RAD18 complex structure prediction with CSP magnitude mapped onto the structure from low CSPs (yellow) to high CSPs (purple). Source data are available online for this figure.

absence of DNA damage. This control is mediated by RAD18 autoubiquitination and subsequent proteasomal degradation (Miyase et al, 2005; Masuyama et al, 2005).

RAD18 is a 56.2 kDa, multi-domain protein that is conserved from yeast to humans (Fig. 1A) (Notenboom et al, 2007). The N-terminal RING domain (residues 1–99) of RAD18 acts as the catalytic domain and is also responsible for RAD18 dimerization (Notenboom et al, 2007; Huang et al, 2011). Binding to PCNA was mapped to the N-terminus of RAD18, although the exact residues involved, and the region on PCNA to which RAD18 binds, remain unknown (Notenboom et al, 2007). The C-terminal RAD6 binding domain (R6BD) (residues 340–395) and the RING domain of RAD18 form non-overlapping interactions with RAD6 (Notenboom et al, 2007). Conflicting reports propose that the zinc finger (ZnF) domain (residues 201–225) has roles in DNA binding, dimerization and binding to Ub (Miyase et al, 2005; Jones et al, 1988; Tateishi et al, 2000). The SAP domain (residues 248–282) has been shown to be involved in DNA binding (Notenboom et al, 2007). The structures of the ZnF domain, R6BD and RING domain have been reported (Pickart, 2001; Huang et al, 2011; Hu et al, 2017; Hibbert et al, 2011; Rizzo et al, 2014).

Melanoma antigens (MAGEs) are a family of ~40 proteins in humans, many of which are selectively expressed in germ-line and cancer cells (Scanlan. et al, 2002). Numerous MAGEs bind RING E3 ubiquitin ligases and modulate their activities through an unknown mechanism (Doyle et al, 2010). For example, MAGE-A3/6 binds TRIM28 and promotes ubiquitination of AMPKα1 (Pineda et al, 2015). MAGEA4 was shown to bind and stabilise RAD18 in mammalian cells, even in the absence of UV-induced DNA damage (Gao et al, 2016). The MAGEA4 interaction surface was mapped to the R6BD of RAD18. Despite high sequence and structural similarity amongst MAGEA proteins, RAD18 binding was found to be highly specific to MAGEA4 (Scanlan et al, 2002; Gao et al, 2016). Normally, MAGEA4 expression is confined to the testes (Fon Tacer et al, 2019). Likewise, RAD18 is highly expressed in the testes (Van Der Laan et al, 2000) and is important for the long-term maintenance of spermatogenesis (Sun et al, 2009), indicating that the MAGEA4/RAD18 interaction may also be relevant during spermatogenesis. Aberrant expression of MAGEA4 in cancer cells may encourage PCNA monoubiquitination and the recruitment of error-prone polymerases, even in the absence of DNA damage, through the stabilisation of RAD18. This could potentially result in mutagenesis and drive cancer progression (Gao et al, 2016). MAGE proteins have been shown to bind to RING E3 ligases via a conserved MAGE homology domain (MHD) (Doyle et al, 2010; Barker and Salehi, 2002). Crystal structures of the MAGE homology domains (MHD) of MAGEA4, -A3, -A11 and -G1 have been reported (Doyle et al, 2010; Newman et al, 2016). However, these structures fail to explain how MAGEs influence RING ligase activity. Structural characterisation of MAGEA4/RAD18 would give general insights into how MAGE proteins interact with, activate and stabilise RING ligases, as well as further elucidating the specific role of MAGEA4 in DDT pathways in germ-line and cancer cells. Furthermore, the MAGEA4/RAD18 interface represents a potential drug target for cancer therapies.

In this study, we used AlphaFold2 (AF), Nuclear Magnetic Resonance (NMR) spectroscopy and mutagenesis experiments to dissect the interaction between the MAGEA4 MHD and the RAD18 R6BD. We identified a peptide-binding groove in MAGEA4 which specifically binds to RAD18. MAGEA4 also outcompetes RAD6 and displaces it from the R6BD of RAD18. We find that MAGEA4 sterically blocks the degradative autoubiquitination activity of RAD18 and thereby stabilizes it. We also defined an evolutionarily invariant intramolecular interaction between the RING and the SAP domains of RAD18 that together play a critical role in PCNA monoubiquitination. Finally, we show that the peptide-binding groove identified in MAGEA4 is conserved in at least one other Type-I MAGE protein, MAGEC2, and possibly also others. Our data provide unique insights into RAD18-driven PCNA monoubiquitination and the regulation of this process by MAGEA4 in cancers.

## Results

### The AF model and validation of the MAGEA4/RAD18 R6B domain complex

MAGEA4 is reported to bind to a RAD18 region corresponding to residues 340–395, which harbours the RAD6 binding domain (R6BD) of RAD18 (Gao et al, 2016) (Figs. 1A and S1). To understand the structural basis of this interaction, we used AlphaFold2 (AF) to generate a predicted model of the RAD18 R6BD (residues 340–400) in complex with the MAGEA4-MHD (residues 101–311) (Fig. 1B). Supporting the accuracy of our AF model of the MAGEA4/RAD18 complex, we found that the per residue confidence values (pLDDT scores) of MAGEA4 and RAD18 are high (ranging between 70 and 90) at the predicted

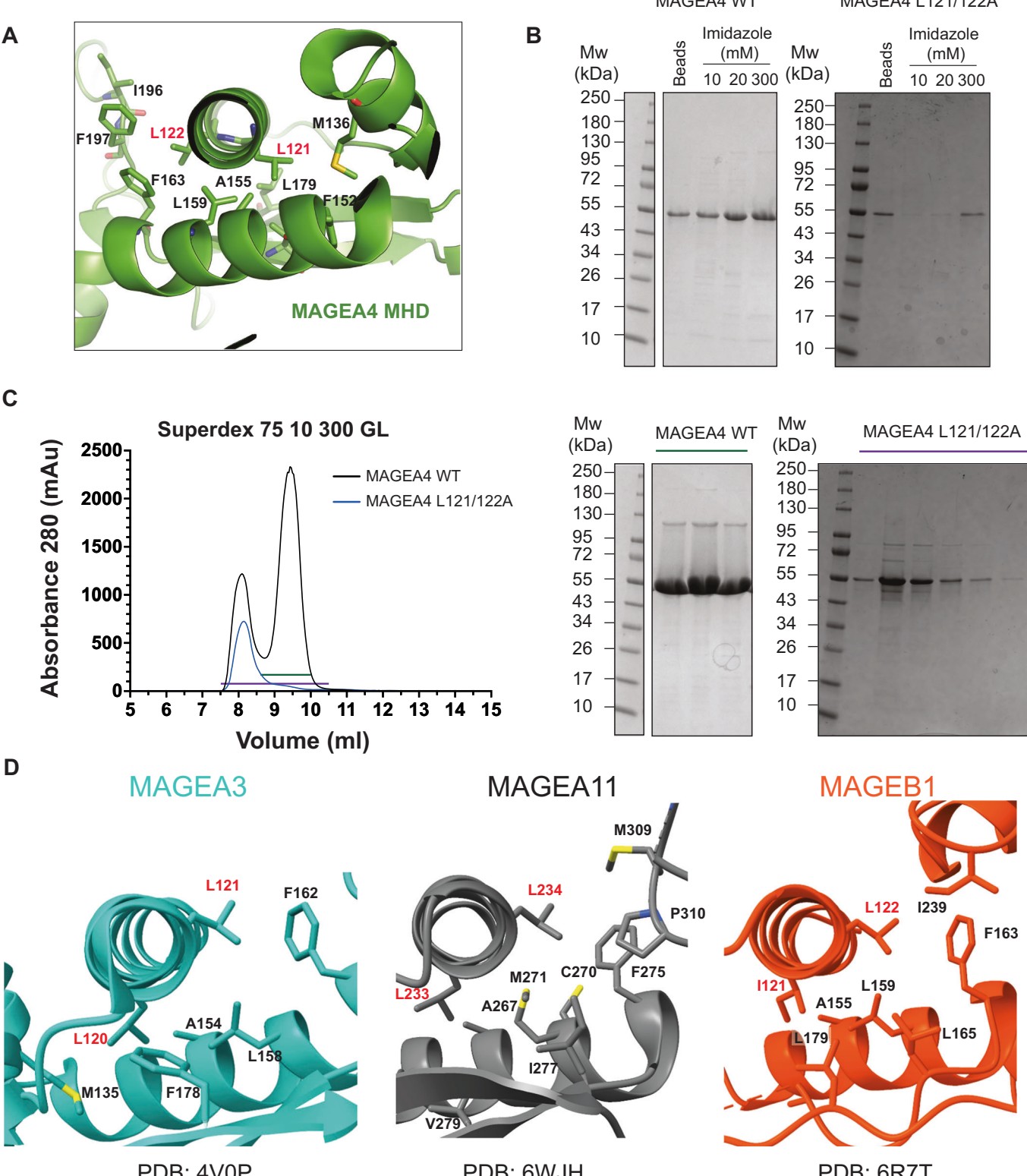

Figure 2. The "dileucine motif" of MAGEA4 contributes towards MAGEA4 stability.

(A) A model of the crystal structure of the MAGEA4 MHD showing that L121 and L122 exist inside the hydrophobic core of WH-A of MAGEA4. (B) Comparison of affinity purification of MAGEA4 WT and MAGEA4 L121/122A mutant from 2 l E. coli. Both proteins harbour N-terminal His-tags and were loaded onto TALON beads and eluted with increasing concentrations of imidazole. Samples were loaded onto SDS PAGE gels and visualized by Coomassie staining. (C) The 300 mM imidazole fraction of MAGEA4 WT or MAGEA4 L121/122A from (B) were subjected to Size Exclusion chromatography (left). Fractions (indicated by the solid lines) were loaded onto SDS PAGE gels and visualized by Coomassie staining (right). (D) Crystal structures of the MHDs of MAGEA3 (PDB: 4V0P), MAGEA11 (PDB: 6WHJH) and MAGEB1 (PDB: 6R7T), showing that the dileucine motif contributes towards the hydrophobic core of these proteins. The dileucine motif of each protein is indicated in red. Source data are available online for this figure.

interface of these molecules (Appendix Fig. S1). As seen in the crystal structure of the MAGEA4-MHD (Newman et al, 2016), our model predicts that the MHD adopts two highly ordered winged-helices (WHs), WH-A ($\alpha$-helices 1–3) and WH-B ($\alpha$-helices 4–8), linked by an extended $\beta$-hairpin. Residues 340–366 of the RAD18 R6BD form two short $\alpha$-helices which pack against a hydrophobic and negatively charged peptide binding groove lining $\alpha$4, $\alpha$5 and $\alpha$8 in WH-B of MAGEA4 (Fig. 1B,C). AF predicts with low confidence that the C-terminus of the RAD18 R6BD (residues 366–400) associates with WH-A of MAGEA4, but this is predicted to be disordered in the AF model and does not participate in specific interactions with the MAGEA4-MHD (Fig. EV1A,B).

The interface of MAGEA4/RAD18 is composed of mostly hydrophobic interactions (Fig. 1D). A strong hydrophobic patch is formed by Y361 of RAD18 and M161 ($\alpha$3), I205 ($\alpha$4), M212 ($\alpha$4) and V291 ($\alpha$8) of MAGEA4. Further hydrophobic interactions exist between F350, V354 and A357 of RAD18 and L288 ($\alpha$8), V285 ($\alpha$8) of MAGEA4 (Fig. 1D). The interaction between MAGEA4 and RAD18 is further stabilized by a salt bridge between R343 of RAD18 and D215 (loop $\alpha$4–$\alpha$5) of MAGEA4. We introduced charged residues replacing the hydrophobic residues at the interface of RAD18 and MAGEA4 and subsequently performed in vitro HIS pulldown assays by co-expressing His-tagged full-length (FL) wild type (WT) or mutant RAD18 with untagged FL MAGEA4 WT in E. coli. All of the tested RAD18 mutants (F350D, V354D, A357D and Y361D) displayed significantly reduced binding to MAGEA4, with F350D and Y361D almost completely abolishing the interaction (Fig. 1E). We also generated the MAGEA4 mutants M161D and the triple mutant (M161D, I205D, L288D) to examine the impact of mutating MAGEA4 residues on RAD18 binding (Fig. EV1C). We next performed isothermal titration calorimetry (ITC) experiments in which we compared the binding affinity of a synthesized RAD18 R6BD peptide (residues 340–366) to MAGEA4 WT and the mutants. We determined a dissociation constant ($K_D$) of 405 nM between MAGEA4 WT and the RAD18 R6BD peptide (Fig. 1F). M161D showed a marked reduction in binding affinity ($K_D = 1.66\,\mu M$) to the RAD18-R6BD peptide (Fig. EV1D), but failed to fully eliminate binding. The triple mutant of MAGEA4 (M161D, I205D, L288D) completely loses binding to the RAD18 R6BD peptide (Fig. 1G), indicating that these hydrophobic residues are important for the RAD18 R6BD-MAGEA4 interaction.

To further validate the AF model experimentally, we performed NMR titrations in which we titrated the RAD18 R6BD peptide used in the ITC titration above, into $^2$H, $^{13}$C, $^{15}$N labelled MAGEA4-MHD (Fig. 1H). We were able to assign ~75% of the MAGEA4-MHD (Fig. EV1E). Missing assignments are due to the large size of the MAGEA4-MHD (~27 kDa), which causes fast

transverse relaxation and therefore signal decrease beyond detection for 25% of residues, despite deuteration of the protein. Furthermore, at pH 7.5, the chemical exchange rates of amide protons in loop regions are not optimal for NMR studies and most missing assignments are therefore in these loop regions. However, we could observe strong chemical shift perturbations (CSPs) in the MAGEA4-MHD upon titration. The CSPs were in the slow exchange regime, indicating strong affinity between MAGEA4 and RAD18 R6BD. This necessitated the assignment of the backbone resonances in the bound state. Strong CSPs were found within $\alpha$8 (E289, V291, V292, R293, V294) and in the loop between $\alpha$8 and $\alpha$9 (V298 and I300) of the MAGEA4 MHD upon titration with the RAD18 R6BD peptide (Figs. 1H,I and EV1E). In addition, MAGEA4 V228 and M229 at the C-terminus of $\alpha$5 displayed significant CSPs, as did residues within the loop connecting $\alpha$5 to $\alpha$6 (G230, V239, Y240 and G241). Although it is not possible to differentiate between CSPs caused by direct interactions and those resulting from allosteric changes, the large CSPs detected in MAGEA4 for $\alpha$8 and the C-terminus of $\alpha$5 are consistent with these regions being involved in binding to the RAD18 R6BD, as suggested by the AF model. Furthermore, these findings are in accordance with MAGEA4 WH-B being primarily responsible for binding to the RAD18 R6BD.

The crystal structure of MAGE-G1 in complex with the RING ligase NSE1 revealed the presence of a dileucine motif at residues L96 and L97 of MAGE-G1 that is involved in the interaction, as well as partially contributing to the hydrophobic core of the MAGE-G1 MHD (Doyle et al, 2010). Mutating these residues in vitro abolished the ability of MAGE-G1 to enhance NSE1 ubiquitination (Doyle et al, 2010). The dileucine motif is conserved throughout the MAGE family, leading to the proposition that the dileucine motif is a general feature utilized by MAGE proteins to bind to partner RING ligases (Doyle et al, 2010). Gao et al mutated the corresponding dileucine motif in MAGEA4 (L121 and L122) to alanine residues and showed that the mutant MAGEA4 was unable to stabilize RAD18 in HEK293T cells (Gao et al, 2016). However, analysis of our AF model and the crystal structure of MAGEA4 shows that these leucine residues are buried within the hydrophobic core of WH-A (Fig. 2A) and could be important for the correct folding of the MAGEA4 MHD. Accordingly, we mutated these residues and expressed the double mutant MAGEA4 L121/122A in E. coli. We found that expression of the MAGEA4 L121/122A mutant was significantly reduced in comparison to MAGEA4 WT, despite both proteins being extracted from the same volume of E. coli culture (2 l) (Fig. 2B, C), indicating that the mutant is less stable than the WT. Furthermore, the MAGEA4 L121/122A mutant eluted in the void volume upon size-exclusion chromatography, indicating that the protein is aggregated (Fig. 2C).

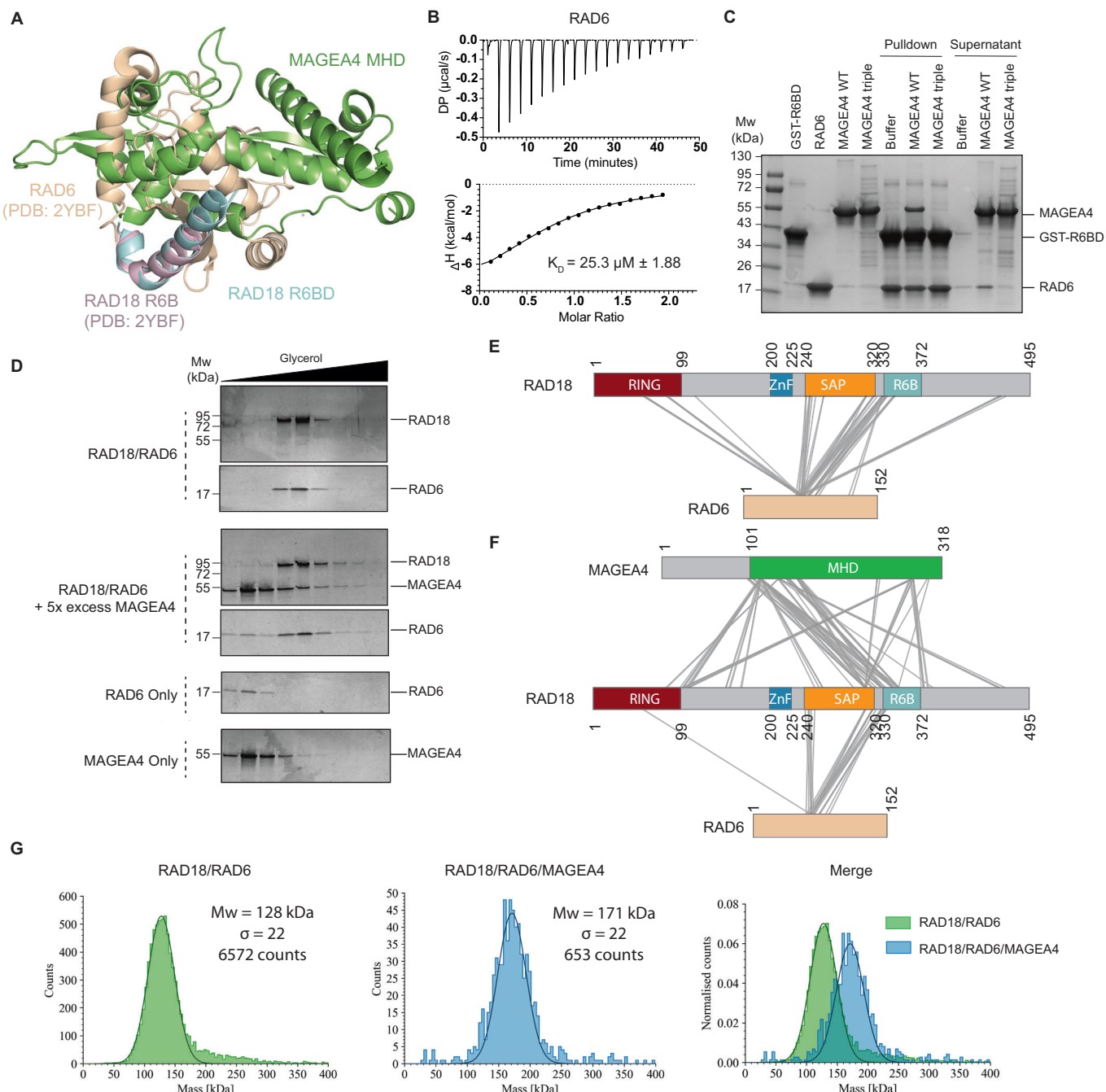

This suggests that the dileucine motif is important for the correct folding of the MAGEA4 MHD and likely only plays an indirect role in the RAD18 interaction. In fact, the crystal structures of MAGEA3, MAGEA11 and MAGEB1 also show that the dileucine motif exists within the hydrophobic core of WH-A, with one of the leucine residues exchanged for isoleucine in MAGEB1 (Newman et al, 2016; Yang et al, 2020a) (PDB: 6R7T) (Fig. 2D), further supporting the notion that the dileucine motif is not a general feature utilized by MAGEs for binding to partner RING ligases. Instead, in many MAGE proteins, the conservation of the dileucine motif is likely to be necessary for the correct folding of the MHD.

## MAGEA4 alters RAD18/RAD6 complex architecture

The E3/E2 complex of RAD18/RAD6 was shown, through the use of multiple RAD18 deletion constructs and cross-linking mass spectrometry experiments (XL-MS), to adopt an intricate architecture where the catalytic N-terminal RING domain and the C-terminally located R6BD both bind non-exclusively to RAD6 (Fig. 1A) (Notenboom et al, 2007; Back et al, 2002). The superposition of the crystal structure of RAD6/RAD18 R6BD (PDB: 2YBF) (Hibbert et al, 2011) with the AF model of MAGEA4/ RAD18 R6BD reveals that the simultaneous binding of both RAD6 and MAGEA4 to the RAD18 R6BD would result in steric clashes

**Figure 3.   MAGEA4 causes conformational changes in the RAD18/RAD6 complex.**

(A) AlphaFold2 model of MAGEA4 MHD/RAD18 R6BD superimposed with the crystal structure of the RAD6/RAD18 R6B domain (PDB: 2YBF). The superimposition indicates clashes between MAGEA4 and RAD6 if they were to bind the RAD18 R6BD at the same time. All proteins are marked in different colours as depicted in the figure. (B) ITC experiment performed between RAD6 and the RAD18 R6BD peptide to measure the binding affinity. This experiment was conducted twice with similar results. (C) GST-tagged RAD18 R6BD was immobilized onto glutathione beads and incubated with a 3x molar excess of RAD6. The immobilized GST-RAD18 R6BD/RAD6 was then incubated with buffer or with a 3x molar excess of MAGEA4 WT or MAGEA4 triple mutant, as indicated. The supernatant, containing proteins not bound to the GST-RAD18 R6BD, and the pulldown, containing proteins bound to the GST-RAD18 R6BD, from each condition were loaded onto a SDS PAGE gel and subjected to Coomassie staining to monitor RAD6 dissociation from the RAD18 R6BD. (D) RAD18/RAD6, RAD18/RAD6 + 5x excess MAGEA4, RAD6 alone or MAGEA4 alone were loaded on 5–20% glycerol gradients. After ultracentrifugation, fractions were collected and subjected to SDS-PAGE followed by detection of proteins using silver staining. (E) The purified RAD18/RAD6 complex was subjected to cross-linking as detailed in the Methods. Cross-linked RAD18/RAD6 was subjected to tryptic digestion and mass spectrometry. Identified inter-molecular cross-links are indicated using solid lines (Dataset EV1). (F) FL MAGEA4 was added in excess to the purified RAD18/RAD6 complex and the mixture was subjected to XL-MS as detailed in the Methods. Cross-links identified between RAD18 and RAD6 or MAGEA4 in the RAD18/RAD6/MAGEA4 complex are indicated using solid lines (Dataset EV2). (G) Mass distributions of the RAD18/RAD6 (left) and RAD18/RAD6/MAGEA4 (middle) complexes measured by Mass Photometry. The solid line shows the best-fit Gaussian distribution, the counts indicate the integral of the Gaussian curve and the σ value gives the Standard Deviation. The mass distributions of both complexes are normalized and overlayed (right). Source data are available online for this figure.

between RAD6 and MAGEA4 (Fig. 3A). Therefore, we postulate that MAGEA4 displaces RAD6 from the RAD18 R6BD, whilst still allowing RAD6 to bind to the RING domain of RAD18. To test this, we performed ITC experiments between RAD6 and the RAD18 R6BD peptide and obtained a Kd of 25.3 μM between these two proteins (Fig. 3B). This means that MAGEA4 binds to the R6BD of RAD18 (Kd of 405 nM, Fig. 1F) with ~60-fold higher affinity than RAD6 does (Figs. 1F and 3B).

To examine the competition between RAD6 and MAGEA4 for the R6BD of RAD18, we performed direct competition assays in which GST-tagged RAD18 R6BD was immobilized on glutathione beads, followed by loading of RAD6 onto the immobilized RAD18 R6BD. The immobilized GST-RAD18 R6BD/RAD6 was then incubated with a 3x molar excess of either MAGEA4 WT or MAGEA4 triple mutant and the dissociation of RAD6 was monitored. A notable amount of RAD6 was displaced by MAGEA4 WT, though a large amount of RAD6 remained bound to the RAD18 R6BD, indicating that MAGEA4 is unable to fully outcompete RAD6 from the RAD18 R6BD (Fig. 3C). As expected, the MAGEA4 triple mutant failed to displace RAD6 from the RAD18 R6BD (Fig. 3C). Furthermore, increasing the concentration of MAGEA4 did not lead to full dissociation of RAD6 from the RAD18 R6BD (Fig. EV2A). It is important to note here that RAD6 has two binding sites on RAD18, namely, the RING domain and the R6BD which would result in much stronger avidity of RAD6 towards full-length (FL) RAD18. To assess the competition between RAD6 and MAGEA4 for FL RAD18, we subjected the purified RAD18/RAD6 complex to glycerol gradient ultracentrifugation to monitor the co-elution of RAD6 with RAD18 in the presence or absence of excess MAGEA4. We found that a small but notable amount of RAD6 dissociated from RAD18 in the presence of MAGEA4, indicating that the RAD18/RAD6 interaction is likely to be weaker in the presence of MAGEA4 due to the partial displacement of RAD6 from the RAD18 R6BD (Fig. 3D). In addition, MAGEA4 co-sediments with the RAD18/RAD6 complex at a higher glycerol (11–18%) concentration compared to RAD18/RAD6 alone (11–14%), consistent with the idea that RAD6 does not dissociate completely from RAD18 in the presence of MAGEA4. This is also consistent with the previous observations that RAD18 is still able to perform RAD6-dependent monoubiquitination of PCNA in the presence of MAGEA4 (Gao et al, 2016). Our pulldown and density gradient data reveal that at least a fraction of MAGEA4 binds to the same monomer of RAD18 as

RAD6, resulting in partial release of RAD6 from the R6BD of RAD18.

To further analyse how the binding of MAGEA4 alters the architecture of RAD18/RAD6, we performed cross-linking mass spectrometry (XL-MS) experiments whereby the RAD18/RAD6 complex was cross-linked using BS3 (bis(sulfosuccinimidyle) subérate) in the presence or absence of excess MAGEA4 (Figs. 3E,F and EV2B; Dataset EV1 and 2) followed by the analysis of the cross-linked peptides using mass spectrometry. As an internal validation of our XL-MS data, we found that RAD6 is in close proximity to both the RING domain and the R6BD of RAD18, as expected (Fig. 3E). Interestingly, we found that a major portion of the cross-links between RAD18 and RAD6 occur in a region N-terminal to the R6BD, this is likely because the R6BD is buried within the interaction interface with RAD6 and is not accessible to the cross-linker. We also detected a number of cross-links between the MAGEA4 MHD and the R6BD of RAD18, in agreement with the AF model of MAGEA4/RAD18 (Figs. 1 and 3F). Surprisingly, MAGEA4 also cross-linked to multiple other regions of RAD18 in addition to the R6BD, particularly to the DNA-binding SAP domain and to the C-terminus of the RING domain. It appears that although MAGEA4 primarily interacts with the R6BD of RAD18, there is close proximity (or association) between the MAGEA4 MHD and the RING and SAP domains of RAD18. From our cross-linking data alone, it is not possible to determine whether the RING and SAP domains that are in close proximity to MAGEA4 originate from the same RAD18 monomer to which MAGEA4 is bound, or whether these domains belong to the other monomer. We failed to detect any cross-links between MAGEA4 and RAD6, in agreement with the notion that both RAD6 and MAGEA4 interact with the RAD18 R6BD mutually exclusively (Fig. 3A; Dataset EV2). Since the XL-MS data on RAD18/RAD6 alone and MAGEA4 were collected separately in two different experiments, the absolute number of cross-links or the intensity of the cross-linked peptides cannot be compared between these two scenarios. However, the pulldown (Fig. 3C) and the glycerol gradient experiments (Fig. 3D) are consistent with the idea that the avidity between RAD6 and RAD18 is reduced in the presence of MAGEA4.

RAD18 reportedly exists as an asymmetric dimer, whereby only one RAD6 protein is capable of binding to dimeric RAD18 (Huang et al, 2011; Masuda et al, 2012). The stoichiometry of MAGEA4, however, is unknown in the RAD18/RAD6/MAGEA4 complex. Firstly, in support of the reported asymmetry of the RAD18 dimer

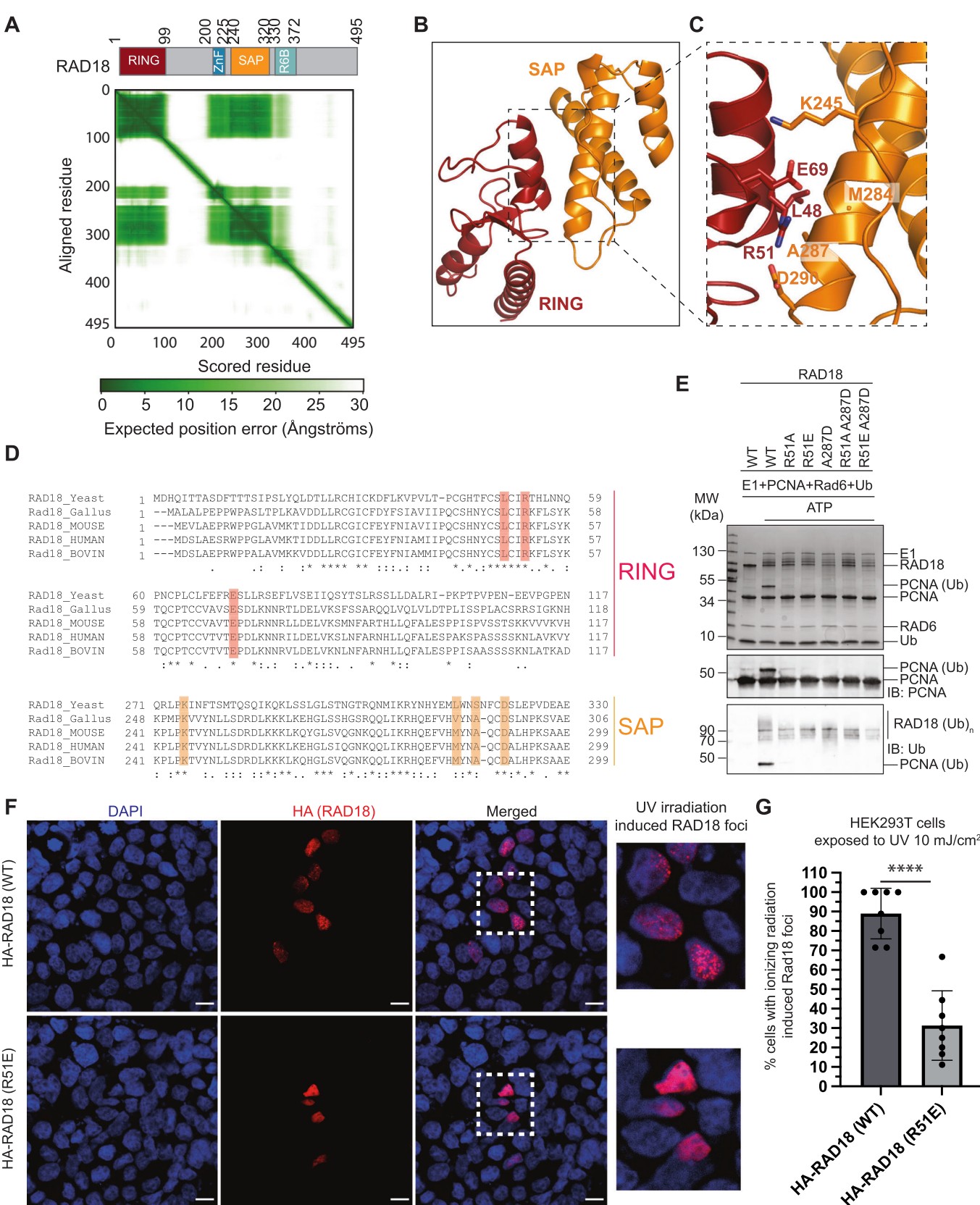

**Figure 4. Interactions between RING and SAP domains of RAD18 are essential for PCNA monoubiquitination.**

(A) Predicted alignment error (PAE) plot of one RAD18 monomer from the AlphaFold (AF) prediction of dimeric RAD18 (Appendix Fig. S1). Above the PAE plot is a re-drawn domain organization of RAD18 as predicted by AF. (B) Interaction interface of human RAD18 RING and SAP domains as predicted by the AF model. (C) Close-up of the RAD18 RING and SAP interface showing various residues from both the domains involved in the interaction. (D) Multiple sequence alignment of the RAD18 RING and SAP domains from multiple species, the residues involved in the RING and SAP domain interaction are highlighted. (E) In vitro ubiquitination assays performed with WT RAD18/RAD6 or the RAD18/RAD6 complexes with various RING-SAP disrupting RAD18 mutants, showing that PCNA ubiquitination is lost upon disrupting the RING-SAP interaction interface. (F) HEK293T cells were transfected with WT or R51E HA-RAD18. Cells were exposed to 10 J/m² and fixed 4 h later. HA-RAD18 was immunostained using anti-HA antibody and the nucleus was stained using DAPI. Scale bar 12 μm. (G) Quantification of the experiment shown in (F). Percentage of transfected cells with ionizing radiation induced foci (IRIF) were counted as described in the Methods for both WT and R51E and plotted $p < 0.0001$ (calculated by an unpaired T-test). Error bars represent Standard Deviation. The centre of the error bars is the mean. The total number of transfected cells analysed ($n$) was 63 for HA-RAD18 WT and 66 for HA-RAD18 R51E. Source data are available online for this figure.

in RAD18/RAD6, we performed analytical ultracentrifugation (AUC) on the RAD18/RAD6 complex and measured a sedimentation coefficient s20w of 6.2S for 92% of the sample, non-interactive species analysis indicated a molecular weight of 131.7 kDa, consistent with an elongated complex containing two RAD18s and one RAD6 (theoretical Mw = 134.6 kDa, derived frictional ratio being 1.6) (Fig. EV2C). However, we were unable to accurately examine the stoichiometry of the RAD18/RAD6/MAGEA4 complex by AUC, perhaps due to the more dynamic nature of this complex compared to RAD18/RAD6 alone. Therefore, to determine the stoichiometry of the RAD18/RAD6/MAGEA4 complex, we performed Mass Photometry on RAD18/RAD6 and RAD18/RAD6/MAGEA4. We determined a molecular weight of 128 kDa for RAD18/RAD6, which is in line with our AUC experiment and previous findings (Fig. 3G) (Masuda et al, 2012). In order to stabilize the RAD18/RAD6/MAGEA4 complex, we first subjected the protein to BS3 cross-linking, followed by size exclusion chromatography to remove the uncross-linked proteins. The peak fraction was taken for Mass Photometry analysis (Fig. EV2D). We determined a molecular weight of 171 kDa for RAD18/RAD6/MAGEA4, indicating that the RAD18/RAD6/MAGEA4 complex exists as a tetramer containing two RAD18s, one RAD6 and one MAGEA4 (Fig. 3G). However, since MAGEA4 only displaces a small fraction of RAD6 from the RAD18/RAD6 complex (Fig. 3C,D), it is unclear whether MAGEA4 directly competes with RAD6 for the same RAD18 R6BD, or whether MAGEA4 binds to the unoccupied R6BD and this results in conformational changes within RAD18 that reduces the affinity between RAD18 and RAD6. Future efforts in experimentally determining the structure of RAD18/RAD6 alone and in complex with MAGEA4 would shed light on this aspect.

## Intramolecular interaction between the RING and the SAP domains of RAD18

In the XL-MS experiment of the RAD18/RAD6 complex, we found notable cross-links between RAD6 and the SAP domain of RAD18 (Fig. 3E,F). To assess if these cross-links reflect direct interactions between RAD6 and the SAP domain of RAD18, or mere physical proximity in the complex, we analysed the AF model of FL dimeric RAD18 (Varadi et al, 2022; Jumper et al, 2021). The predicted aligned error (PAE) plot of the RAD18 model suggests an interdomain interaction between the RING and the SAP domains of RAD18 (Fig. 4A; Appendix Fig. S1). Consistently, we also detected multiple cross-links between the RING domain and the SAP domain of RAD18 in the XL-MS analysis (Dataset EV1 and 2). In

the AF model of RAD18, the central α-helix of the SAP domain packs against the sole α-helix of the core RING domain (Fig. 4B). Importantly, the AF predicted interface of the RAD18 RING and SAP domains is compatible with the RING-RAD6 interface, as deduced from a model created based on the crystal structure of a closely related RING-E2 (PDB:6W9A) structure (Fig. EV3A). The SAP-RING dimer is also compatible with the previously determined crystal structure (PDB:2Y43) of the dimeric RAD18 RING domain (Fig. EV3B) (Huang et al, 2011). The AF model of dimeric RAD18 indicates that the RING and SAP domain from the same RAD18 monomer interact (Appendix Fig. S1), however, it is entirely feasible that the RAD18 dimer adopts a conformation in which the RING and SAP domains from different monomers interact.

The predicted interface of RING and SAP includes two salt bridges; one between R51 of RING and D290 of SAP, and another between E69 of RING and K245 of SAP (Fig. 4C). A hydrophobic interaction network composed of residues A287, M284 of SAP and L48 of the RING domain further strengthens the RING-SAP interaction (Fig. 4C). Importantly, multiple sequence alignment of RAD18 revealed conservation of residues involved in the RING-SAP interface and confirmed that this RING-SAP interaction is evolutionarily conserved from yeast to humans (Fig. 4D). We generated several mutants of RAD18 (R51A, R51E, A287D, R51A_A287D and R51E_A287D) to disrupt the interface between the RING and SAP domains. We purified these mutant RAD18/RAD6 complexes and conducted PCNA monoubiquitination assays (Fig. 4E). Strikingly, none of the mutants exhibited any PCNA monoubiquitination activity, while they retained RAD18 auto-ubiquitination activity to a large extent. Following exposure to UV radiation, RAD18 has been shown to localize to sites of replication fork stalling and form distinct nuclear foci, termed ionizing radiation induced foci (IRIF) (Watanabe et al, 2009) (Fig. 4F). In the absence of UV radiation, RAD18 is more diffuse within the nucleus (Fig. EV3C). Strikingly, RAD18 with a disrupted RING-SAP interface was also defective in the formation of IRIF in irradiated HEK293T cells, indicating that the SAP-RING interaction might also play a role in RAD18 recruitment to damaged DNA sites (Figs. 4F,G and EV3C). Interestingly, PCNA was previously shown to interact with the N-terminus of RAD18, however, binding experiments with individual domains of RAD18 failed to detect any direct interaction with PCNA (Notenboom et al, 2007). Considering the SAP-RING interaction reported here, along with the previous studies that implicated the SAP domain in interactions with DNA (Notenboom et al, 2007), we propose that the RING and SAP domains cooperate through intramolecular interactions to

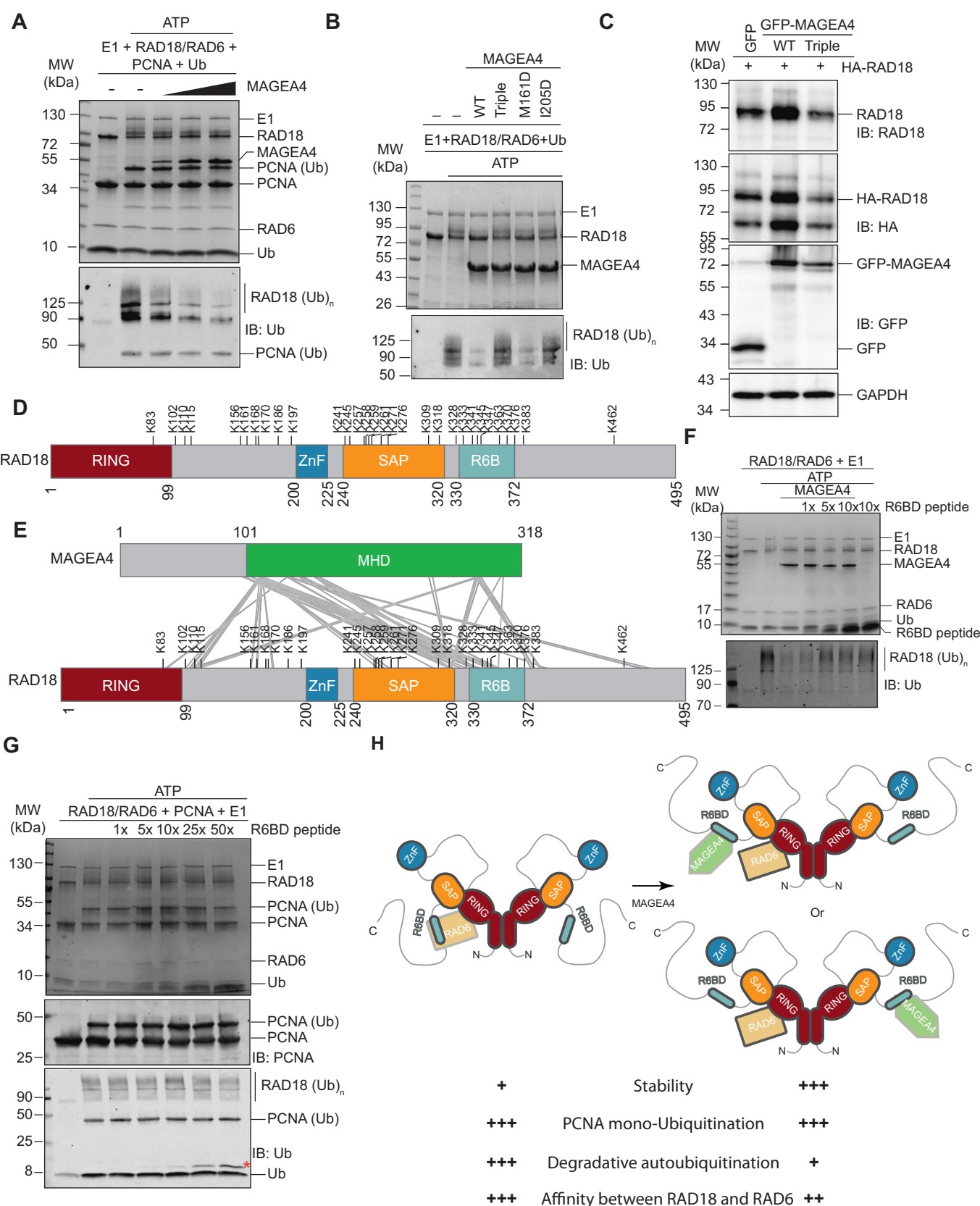

**Figure 5.   MAGEA4 inhibits RAD18 autoubiquitination without hindering PCNA ubiquitination.**

(A) In vitro ubiquitination reactions performed using the RAD18/RAD6 complex in the presence of increasing amounts of MAGEA4 (1, 3 or 5 μM). Reactions were subjected to SDS-PAGE followed by Coomassie staining (Top), part of the reactions were subjected to SDS-PAGE followed by WB. Autoubiquitination of RAD18 and PCNA monoubiquitination were monitored using an antibody against Ub (Bottom). (B) In vitro ubiquitination reactions were performed using the RAD18/RAD6 complex in the presence of wild-type MAGEA4 or the indicated mutants. 5X molar MAGEA4 to RAD18/RAD6 was added. Reactions were subjected to SDS-PAGE followed by Coomassie staining (Top), part of the reactions were subjected to SDS-PAGE followed by WB. Autoubiquitination of RAD18 was monitored using an antibody against Ub (Bottom). (C) HEK293T cells were co-transfected with HA-tagged RAD18 and GFP alone, GFP-tagged MAGEA4 WT or GFP-tagged MAGEA4 triple mutant. The soluble fraction was isolated and loaded onto Coomassie gels, followed by WB with antibodies against RAD18, to monitor endogenous RAD18, HA, to monitor HA-tagged RAD18 levels, GFP, to monitor GFP-MAGEA4 levels. GAPDH was used as a loading control. (D) Autoubiquitinated RAD18 was subjected to LC-MS/MS to identify ubiquitination sites (Appendix Table S1). The identified Lysine residues are depicted here on a cartoon of RAD18 showing various domains. (E) Overlay of the cross-links identified between RAD18 and MAGEA4 and the lysine residues targeted for RAD18 autoubiquitination. (F) In vitro ubiquitination reactions were performed using the RAD18/RAD6 complex in the presence of MAGEA4 and/or increasing amounts of the RAD18 R6BD peptide. A 5X molar excess of MAGEA4 to RAD18/RAD6 was added. The RAD18 R6BD peptide was added in an increasing molar ratio to MAGEA4, as indicated. Reactions were subjected to SDS-PAGE followed by Coomassie staining (Top), part of the reactions were subjected to SDS-PAGE followed by WB. Autoubiquitination of RAD18 was monitored using an antibody against Ub (Bottom). (G) In vitro ubiquitination reactions were performed using the RAD18/RAD6 complex to test the effect of the R6BD peptide's presence on PCNA monoubiquitination. The R6BD peptide was added in an increasing molar ratio to RAD18/RAD6, as indicated. Reactions were subjected to SDS-PAGE, followed by Coomassie staining (Top), part of the reactions were subjected to SDS-PAGE followed by WB. PCNA ubiquitination was monitored using an antibody against PCNA. Autoubiquitination of RAD18 and PCNA monoubiquitination were monitored using an antibody against Ub (Bottom). *Ubiquitinated R6BD peptide. (H) A model depicting (not drawn to scale) the organization of various domains of the RAD18/RAD6 complex in the presence and absence of MAGEA4. Without MAGEA4, RAD6 is bound to both the R6BD and RING domains of RAD18 and in this configuration, RAD18 is prone to degradative autoubiquitination, presumably to regulate its cellular levels. MAGEA4 triggers the formation of a tetrameric complex whereby MAGEA4 replaces RAD6 at the R6BD of RAD18, whilst the RAD6-RAD18 RING domain interaction is retained. In this configuration, the RAD18/RAD6/MAGEA4 complex is inefficient at autoubiquitination. RAD18 is an asymmetric dimer composed of two RAD18 molecules and one RAD6 molecule. It is unclear whether MAGEA4 binds to the RAD18 monomer that is engaged with RAD6, or whether MAGEA4 interacts with the free RAD18 monomer, indicated by the two models of MAGEA4 binding shown. Various properties of the RAD18/RAD6 complexes are indicated below the cartoons and their respective intensities (+++ or +) are attributed to both configurations of the complex. Although the model shows the RING and SAP domains from a single monomer of RAD18 interacting, it is also possible that the RING domain of one monomer interacts with the SAP domain of the other monomer. Source data are available online for this figure.

specifically recognize and engage PCNA loaded onto the DNA for ubiquitination. RAD18 is a key factor contributing to increased resistance to radio/chemotherapy in several cancers (Xie et al, 2014; Wu et al, 2019; Li et al, 2022; Du et al, 2022), therefore it is of high therapeutic interest to inhibit RAD18 in these scenarios. Inhibiting the interaction of the RING and SAP domains offers a pragmatic approach to silencing RAD18, without affecting the function of RAD6, which plays an important role in transcription (Kim et al, 2009).

## MAGEA4 stabilizes RAD18 by inhibiting RAD18 autoubiquitination

MAGEA4 is known to mediate stabilization of RAD18 through inhibiting its ubiquitin-dependent proteolysis and promoting trans-lesion synthesis in the adenocarcinoma cell line H1299 (Gao et al, 2016). RAD18 stability is also regulated through the action of USP7, which cleaves the RAD18-made ubiquitin chains and inhibits the proteasomal degradation of RAD18 (Zlatanou et al, 2016). To understand the molecular basis of RAD18 regulation through ubiquitination, we performed in vitro ubiquitination assays with the RAD18/RAD6 complex with and without MAGEA4. In agreement with the previous reports, we show that the presence of MAGEA4 does not alter the RAD18/RAD6-mediated monoubiquitination of PCNA (Fig. 5A) (Gao et al, 2016). However, RAD18 autoubiquitination was markedly reduced in the presence of MAGEA4 in a concentration-dependent manner (Fig. 5A). Although RAD18 forms distinct foci following exposure to UV, MAGEA4 reportedly remains soluble irrespective of UV treatment, indicating that MAGEA4 binds to soluble RAD18 (Gao et al, 2016). Moreover, ubiquitinated or ubiquitin-fused RAD18 remains soluble, even following UV treatment, suggesting that RAD18 ubiquitination also likely occurs to soluble RAD18 (Zeman et al, 2014). To determine whether the effect of MAGEA4 on RAD18

autoubiquitination is dependent on the latter's binding to DNA (Tsuji et al, 2008), we performed in vitro ubiquitination assays in the presence of single-stranded DNA. The addition of DNA did not impact the ability of MAGEA4 to bind and protect RAD18 from autoubiquitination (Fig. EV4A).

To test if the reduction in the autoubiquitination of RAD18 is a direct result of MAGEA4 binding to RAD18, we conducted in vitro ubiquitination assays using MAGEA4 mutants that are deficient in binding to RAD18 (Fig. 1). MAGEA4 M161D retained the ability to block RAD18 autoubiquitination, though to a milder extent compared to MAGEA4 WT, consistent with the reduced binding of this mutant observed through ITC (Figs. 5B and EV1D). However, the addition of the MAGEA4 triple mutant (M161D, I205D, L288D) and the I205D mutant showed RAD18 autoubiquitination levels similar to those in the absence of MAGEA4 (Fig. 5B), suggesting that the MAGEA4/RAD18 interaction directly inhibits RAD18 autoubiquitination. To understand the impact of MAGEA4 on RAD18 stability in the cellular context, we co-transfected HEK293T cells with HA-tagged RAD18 and either GFP-tagged MAGEA4 WT, GFP-tagged MAGEA4 triple mutant or GFP alone and monitored both endogenous RAD18 levels and HA-tagged RAD18 levels (Figs. 5C and EV4B). In agreement with the idea that MAGEA4 protects RAD18 from degradation, we found higher levels of RAD18 in the presence of GFP-MAGEA4 WT in comparison to cells transfected with GFP-alone or the GFP-MAGEA4 triple mutant, which is deficient in RAD18 binding and fails to inhibit RAD18 autoubiquitination in vitro (Fig. 5B,C).

To understand how MAGEA4 is able to block autoubiquitination of RAD18, we sought to identify the sites of RAD18 autoubiquitination. Towards this, we performed an in vitro ubiquitination assay and subjected the smeared band corresponding to the autoubiquitinated fraction of RAD18 to mass spectrometry analysis (Fig. EV4C). We found that numerous lysine residues are targeted for autoubiquitination along RAD18 (Fig. 5D).

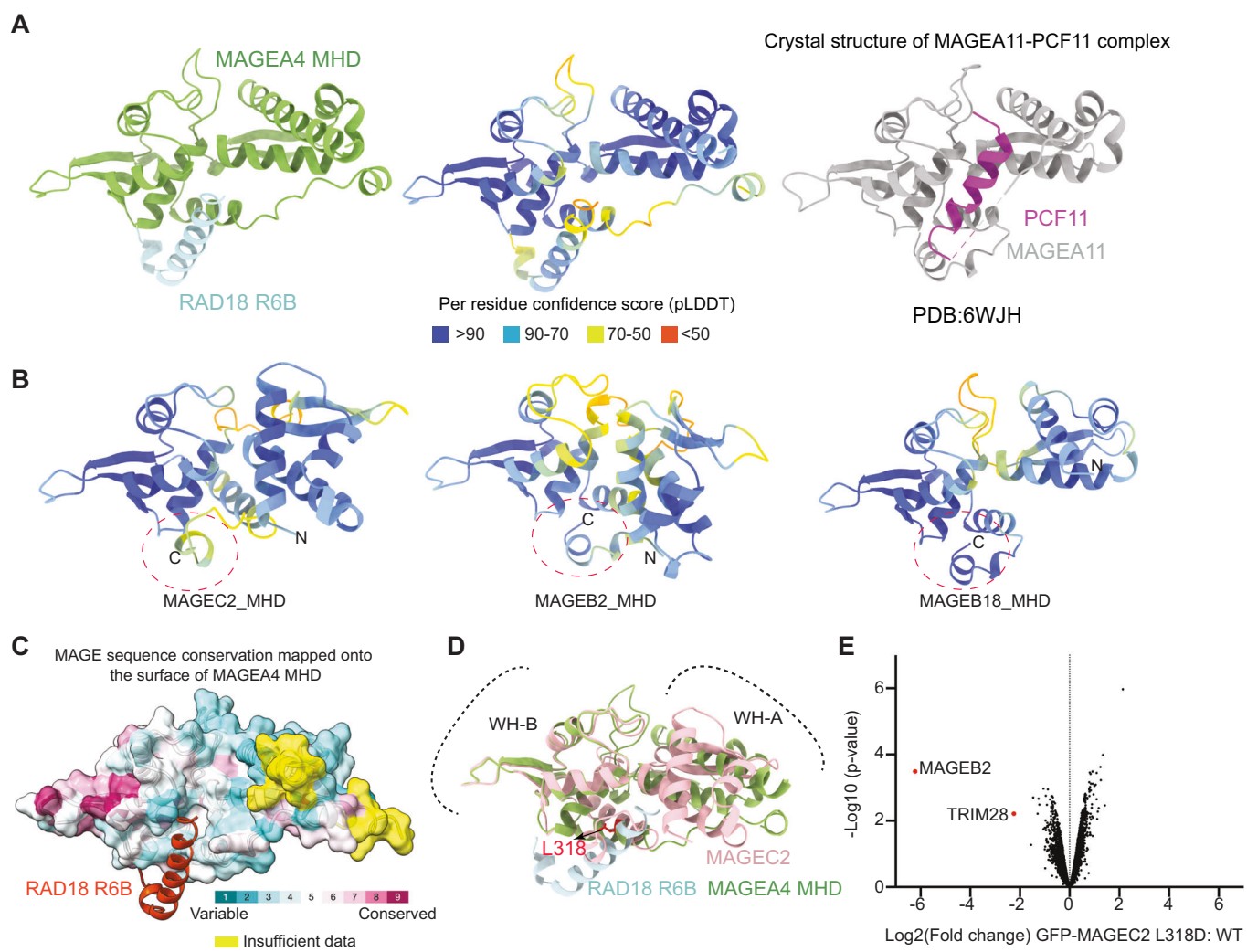

**Figure 6. Identification of a potential "Ligase-binding cleft" in type-I MAGEs.**

(A) AlphaFold model of MAGEA4/RAD18 (left). AlphaFold model of MAGEA4/RAD18 with per residue confidence scores (pLDDT) mapped onto the model (Middle). MAGEA11/PCF11 crystal structure (right). (B) AlphaFold models of MAGEC2, MAGEB2 and MAGEB18 with per residue confidence scores (pLDDT) mapped onto the model. An equivalent region corresponding to the R6BD-binding groove (circled in dotted red line) of MAGEA4 is occupied by a C-terminal helical peptide in MAGEC2, MAGEB2 and MAGEB18. (C) AF structure-based sequence alignment of all type-I MAGEs was performed using PROMALS3D (Pei et al, 2008). The resulting conservation of residues is mapped onto the surface of the MAGEA4 MHD using the Consurf server (Landau et al, 2005; Glaser et al, 2003). Dark magenta indicates high conservation whereas dark cyan indicates high variability of residues. Yellow indicates that the sequence alignment data is not available for these regions of MAGEA4. The RAD18 R6BD peptide is coloured in red. (D) WH-B focused superimposition of the Alphafold models of MAGEA4/RAD18 and MAGEC2. Residue L318 present in the peptide-binding groove of MAGEC2 is coloured in red. (E) GFP-tagged MAGEC2 WT and MAGEC2 L318D were expressed in HEK293T cells. Immunoprecipitation (IP) of the recombinant proteins was performed, followed by their analysis using quantitative mass spectrometry. A volcano plot shows the enrichment of various proteins bound to WT vs L318D MAGEC2 proteins. MAGEB2 and TRIM28 proteins, which are enriched in WT but not in MAGEC2 L318D IP, are labelled. This experiment was conducted in triplicate (biological replicates). P-values were estimated from t-values calculated by a moderated t-test (limma) with the fdrtool R package.

Previously, sites of autoubiquitination of murine RAD18 were mapped to K161, K261, K309 and K318 (Notenboom et al, 2007). All these lysine residues are conserved in human RAD18 and interestingly, we also found them as targets of autoubiquitination. In addition, we also identified multiple ubiquitin chain types in the sample including K48, K63, K11 and K6, indicating that RAD18 autoubiquitination is likely degradative in nature (Appendix Table S1). Intriguingly, polyubiquitination of RAD18 was previously shown to accumulate in cells (GM637 and Mori SV) treated with the proteasomal inhibitor MG132. Moreover, upon autoubiquitination in the presence of E1 and RAD6, RAD18 acts as a

substrate for the proteasome in vitro (Miyase et al, 2005). Strikingly, overlaying the cross-linking sites of MAGEA4/RAD18 onto the autoubiquitination sites of RAD18 reveals notable overlap, hinting that MAGEA4 sterically disallows the autoubiquitination of RAD18 (Fig. 5E) and thereby prevents its degradation in cells.

MAGEA4-induced stabilization of RAD18 has been proposed to cause increased tolerance to DNA damage and replicative stress to drive tumour progression (Gao et al, 2016). Therefore, it is of interest to disrupt this interaction, especially in combination with the genotoxic stress inducing radio/chemo therapy in cancers that are aberrantly overexpressing MAGEA4 (Bhan et al, 2012;

Fujiwara-Kuroda et al, 2018; Sani et al, 2018). As proof of principle, we tested if a peptide consisting of the R6BD of RAD18 is able to inhibit the interaction between MAGEA4 and RAD18. We performed in vitro ubiquitination using RAD18/RAD6 and MAGEA4 in the presence of increasing amounts of the R6BD peptide. The R6BD peptide effectively reduces the ability of MAGEA4 to inhibit RAD18 autoubiquitination (Fig. 5F). Expectedly, the R6BD peptide also inhibits RAD6-driven autoubiquitination of RAD18 (Fig. 5F), since the R6BD peptide also binds to RAD6, albeit with lower affinity compared to MAGEA4 (Figs. 1F and 3B). Intriguingly, addition of the R6BD peptide did not affect RAD18-mediated monoubiquitination of PCNA (Fig. 5G), consistent with the observation that MAGEA4 also does not affect this process in vitro (Fig. 5A). Our data suggest that MAGEA4 induces architectural changes in RAD18/RAD6 (Fig. 3) and inhibits RAD18 degradative autoubiquitination, resulting in higher stability of RAD18 in cells (Fig. 5H) (Gao et al, 2016).

## A potential ligase-binding cleft in MAGEs

Potts and colleagues have discovered an ubiquitination substrate-binding cleft (SBC) in MAGEA11 which is sandwiched between the two Winged-helix sub domains of the MHD (Yang et al, 2020a). This SBC has been shown to be important for substrate-binding in MAGEA11, MAGEA3 and MAGEF1 proteins. Moreover, the sequence variability in the SBC across MAGEs was proposed to be responsible for the selectivity of different MAGEs towards particular substrates. However, it is not known whether MAGEs use a specific structural feature to bind their partner ubiquitin ligases. Different MAGEs have been shown to recognize various regions of RING ubiquitin ligases using a mechanism that is not understood (Doyle et al, 2010). The R6BD of RAD18 binds to a distinct peptide-binding groove in WH-B of the MAGEA4 MHD, which is contiguous to the SBC identified in MAGEA11 (Fig. 6A). Interestingly, analysis of the crystal structure of MAGEA11 (Fig. 6A) and the AlphaFold structures of a number of Type-I MAGEs revealed that the equivalent of the R6BD-binding groove observed in MAGEA4 is occupied with a small helical peptide from the C-terminus of the respective MAGEs (Figs. 6B and EV5). Based on this, we hypothesized that this region in WH-B has a propensity to interact with short helical peptides. Moreover, analogous to the SBC in MAGEs, we found that the residues lining this peptide-binding groove (PBG) in WH-B are some of the least conserved in the otherwise highly homologous MHDs, suggesting that the PBG could be responsible for the selectivity of MAGEs towards their ligase partners (Fig. 6C; Appendix Fig. S2). We sought to test the role of this PBG in another MAGE protein, MAGEC2, the expression of which is a poor prognostic marker in many cancers and which also acts as an oncogene (Bhatia et al, 2013; Yang et al, 2007). MAGEC2 interacts with the ubiquitin ligase Tripartite motif-containing protein 28 (TRIM28), which plays crucial roles in transcriptional regulation and DNA repair (Iyengar and Farnham, 2011). MAGEC2, in association with TRIM28, mediates the ubiquitination and degradation of the tumour suppressors Fructose-1,6-biphosphatase (FBP1) and p53 (Yang et al, 2007; Jin et al, 2017). We sought to understand how MAGEC2 specifically recognizes TRIM28 as this will provide a strategy to target MAGEC2 in various cancer therapeutics. We superposed the AF model of MAGEC2 with the model of R6BD bound MAGEA4

(Fig. 6D). Based on this, we chose to mutate L318 of MAGEC2 which lies in the PBG region. We ectopically expressed GFP-tagged WT and L318D MAGEC2 in HEK293T cells and immunoprecipitated GFP-tagged proteins, followed by analysis of their interactomes in a quantitative mass spectrometry experiment (Fig. 6E). The mutation of L318 in MAGEC2 resulted in specific loss of binding to the endogenous partner E3 ligase TRIM28, confirming that the PBG in WH-B is involved in binding to the ligase partner in MAGEC2. Interestingly, we also found specific loss of binding to MAGEB2 in the MAGEC2 L318D mutant compared to the WT. It is unclear whether there is a direct interaction between MAGEB2 and MAGEC2 or if their interaction is mediated through TRIM28, a shared interaction partner of both these MAGEs (Yang et al, 2007). Overall, our data indicate that akin to MAGEA4, MAGEC2 interacts with TRIM28 using a PBG in its WH-B subdomain. Importantly, targeting the PBG of MAGEs might provide a viable therapeutic approach in cancers overexpressing these tumour-driving MAGEs (Weon and Potts, 2015).

## Discussion

Here we employed Alphafold, XL-MS, mutagenesis and NMR to elucidate that the WH-B subdomain of MAGEA4 interacts with a short helical peptide in the RAD6-binding domain (R6BD) of RAD18 (Figs. 1 and 3). Based on our model, we designed MAGEA4 mutants deficient in RAD18 binding and showed that these mutants are unable to protect RAD18 from autoubiquitination in vitro and fail to stabilize RAD18 in HEK293T cells (Fig. 5B,C). Furthermore, we found a strong correlation between the auto-ubiquitination sites of RAD18 and the MAGEA4-RAD18 cross-linking sites, indicating that MAGEA4 stabilises RAD18 by sterically blocking these ubiquitination sites and preventing degradative autoubiquitination (Fig. 5). Of note, we detected numerous cross-links between the WH-A subdomain of MAGEA4 and RAD18 (Fig. 3F), in apparent contradiction to the AF prediction (Fig. 1B), which implicates only WH-B of MAGEA4 in RAD18 binding. While XL-MS accurately determines the closely associated regions in proteins, it is an indirect technique to unravel the direct interactions. We think the cross-links between WH-A and RAD18 are due to the close association of WH-A and WH-B in the MAGEA4 MHD structure and the resulting proximity of WH-A to various regions of RAD18. In addition, given the dimeric nature of RAD18, we cannot exclude the possibility that the cross-links between WH-A and RAD18 occur with the RAD18 monomer that is not already bound to MAGEA4. NMR experiments conducted between the MHD of MAGEA4 and the R6BD peptide clearly implicate only WH-B of MAGEA4 in RAD18 binding (Figs. 1H and EV1E). Moreover, mutagenesis of residues within and in the close vicinity of the PBG in WH-B (Fig. 1G) is sufficient to abrogate the interaction between MAGEA4 and RAD18, strongly suggesting that MAGEA4-WH-B is mainly implicated in the interaction with RAD18.

It has been found that full-length RAD18 exists as an asymmetric dimer, accommodating only one molecule of RAD6 both in cells and in vitro (Huang et al, 2011; Masuda et al, 2012). Our Mass Photometry data indicates that the RAD18/RAD6/MAGEA4 complex exists as a tetrameric complex consisting of two RAD18s, one RAD6 and one MAGEA4 (Fig. 3G). Our XL-MS,

competition assay and density gradient centrifugation experiments (Fig. 3) strongly suggest that MAGEA4 at least partially displaces RAD6 from the R6BD of RAD18, hinting that MAGEA4 likely binds to the RAD18 monomer that is bound to RAD6, whilst still allowing RAD6 to interact with the RING domain of RAD18. Further studies are required to determine if the R6BD of one of the RAD18 monomers in the dimer is inaccessible to both RAD6 and MAGEA4, or whether MAGEA4 engages with the unoccupied RAD18 R6BD and this causes allosteric changes within the RAD18 dimer that reduce the overall affinity between RAD18 and RAD6.

Here, we also identified a novel intramolecular interaction in RAD18 between the RING and SAP domains that is evolutionarily conserved from yeast to humans (Fig. 4). Previous studies using XL-MS, analytical size exclusion chromatography and pulldown experiments showed that RAD18 adopts an intricate architecture, where the RING domain and R6BD together interact non-exclusively with RAD6 (Huang et al, 2011; Notenboom et al, 2007; Back et al, 2002). We propose that the RING and SAP domains together form an interface that interacts with PCNA and consistent with this, we show that interfering with this conserved interface of RING and SAP abrogates PCNA monoubiquitination (Fig. 4E). RAD18 is one of the critical molecules responsible for the resistance to chemotherapy and radiotherapy in various cancers (Xie et al, 2014; Wu et al, 2019; Li et al, 2022; Du et al, 2022), therefore, targeting the RING-SAP interface provides a unique way to inhibit RAD18 and could be employed in combination with the genotoxicity-inducing chemo/radio therapies. Future studies aimed at determining the experimental structures of full-length RAD18 in complex with RAD6, MAGEA4 and/or PCNA/DNA are needed to understand how various domains of RAD18 come together to enable monoubiquitination of the DNA-sliding clamp PCNA at the stalled replication fork.

We also showed that a peptide based on the R6BD of RAD18 could specifically inhibit MAGEA4-dependent reduction in RAD18 autoubiquitination (Fig. 5F). This R6BD peptide naturally also binds to RAD6 and could competitively inhibit the RAD6 interaction with the R6BD of RAD18. Despite this, the peptide did not affect the RAD18/RAD6-mediated monoubiquitination of PCNA, even at the very high concentrations tested (Fig. 5G). This indicates that the interaction of RAD6 with the RAD18 R6BD may not be critical to substrate ubiquitination but may only play a role in the autoubiquitination of RAD18. Consistent with this, MAGEA4 addition, which partially displaces RAD6 from the R6BD of RAD18, reduces RAD18 autoubiquitination without impacting PCNA monoubiquitination (Fig. 5A). A variant of R6BD-based peptide or peptidomimetic with high affinity to the peptide-binding groove (PBG) of MAGEA4 has the potential to be a therapeutic agent in cancers overexpressing MAGEA4 and RAD18 (Xie et al, 2014; Wu et al, 2019; Li et al, 2022; Du et al, 2022; Bhan et al, 2012; Fujiwara-Kuroda et al, 2018; Sani et al, 2018). Recently, an unbiased screen identified a circular peptide inhibiting the MAGEA4-RAD18 interaction that binds to the same location that we identified as the R6BD peptide-binding groove in MAGEA4 (Fleming et al, 2022).

Our study gives further insights into how MAGE proteins can regulate RING E3 ligases. Unlike other MAGEs, which have been shown to directly alter RING E3 ligase targets or enhance ligase activity (Doyle et al, 2010), MAGEA4 indirectly regulates RAD18 activity by blocking degradative autoubiquitination. Given the ability of some MAGE proteins to modify RING E3 ligase substrate specificity, for example MAGEA11, MAGEA3 and MAGEF1 redirect the RING E3 ligases HUWE1, TRIM28 and NSE1 respectfully to novel substrates (Pineda et al, 2015; Yang et al, 2020a; Weon et al, 2018; Yang et al, 2020b), it is also possible that MAGEA4 could encourage RAD18 to ubiquitinate substrates other than PCNA.

The stabilisation of RAD18 could promote TLS in the absence of DNA damage, promoting mutagenesis and cancer development. Indeed recruitment of Y-family polymerases has been associated with cancer phenotypes and RAD6 inhibitors are in development as cancer therapeutics (Yoon et al, 2009; Kothayer et al, 2016; Saadat et al, 2018). In addition, MAGEA4 overexpression has been implicated in numerous cancers and MAGEA4 has long been identified as a promising target for T-cell-based immunotherapy (Duffour et al, 1999; Shichijo et al, 1995; Davari et al, 2021; Sanderson et al, 2020). RAD18 stabilisation in the cell could have wide ranging consequences since RAD18 has been found to function in the DNA damage response outside of monoubiquitinating PCNA. For instance, RAD18 can form a molecular bridge between polη and the SUMO ligase PIAS1, stimulating polη SUMOylation and enhancing polη recruitment to stalled replication forks (Despras et al, 2016). RAD18 has also been shown to be recruited to damaged chromosomes through its ZnF domain, in a manner dependent on the RNF8-mediated ubiquitination of damaged chromosomes (Huang et al, 2009; Räschle et al, 2015). This recruitment of RAD18 to damaged chromosomes has been shown to promote homologous recombination repair (Huang et al, 2009). The influence of MAGEA4 on these functions of RAD18 is an interesting avenue for future studies.

## Methods

### AlphaFold

To predict the structure of the MHD of MAGEA4 in complex with the binding peptide from RAD18, the amino acid sequences of MAGEA4 (Uniprot P43358 residues 101–317) and RAD18 (Uniprot Q9NS91 residues 340–365) were linked with a 40-residue repeating GSGS linker. The resulting MAGEA4-MHD-GSGS(40)-RAD18 R6BD peptide construct was then used for structural prediction via AlphaFold2 (Jumper et al, 2021). To improve the accuracy of the binding interface, the config.py file was modified to perform a total of 12 iterative refinement cycles (default parameter: 3), as recommended in Supplementary Methods 1.8 in (Jumper et al, 2021). Although all five predicted models created in this manner showed near-identical structures and binding interfaces, the highest-confidence model (as determined by the model pLDDT score) was chosen to validate the interface biochemically. The RAD18 dimer (Uniprot Q9NS91 residues 1–495) was predicted similarly, with the exception that AlphaFold multimer was used (preprint: Evans et al, 2021.

### Protein expression and purification

Proteins were expressed in *E. coli* BL21star by growing the bacteria at 37 °C in lysogeny broth (LB), supplemented with the appropriate antibiotic, to an OD600 of 0.6 and inducing the cells with 0.2 mM

IPTG at 18 °C for 16 h. RAD18/RAD6, MAGEA4, PCNA and RAD6 all contain an N-terminal histidine tag (located on RAD18 in the case of RAD18/RAD6). MAGEA4 and RAD18 were cloned into PET15C. PCNA and RAD6 were cloned into pCoofy1 and the N-terminal His-tag was removed from pCoofy1-RAD6 by mutagenesis (Appendix Table S2). Following lysis in 200 mM NaCl, 50 mM Tris pH 7.5, 10% Glycerol (lysis buffer), each protein was incubated rolling on TALON® Superflow™ resin for 1 h at 4 °C and eluted with 300 mM imidazole, following numerous washes with lysis buffer and low amounts of imidazole. Proteins were then subjected to size exclusion chromatography (SEC) and eluted in 100 mM NaCl, 20 mM HEPES pH 7.5, 1 mM TCEP (SEC buffer). RAD18 and MAGEA4 mutants were generated by site-directed mutagenesis with primers listed in Appendix Table S2.

## NMR

All samples for NMR were measured in 200 mM NaCl, 20 mM Hepes, 1 mM TCEP pH 7.5, 10% $D_2O$ (for the deuterium lock) and 0.01% $NaN_3$ at 298 K on Bruker Avance III NMR spectrometers with magnetic field strengths corresponding to proton Larmor frequencies of 600 MHz, 800 MHz and 1 GHz (all equipped with triple resonance gradient cryogenic probe heads).

Experiments for backbone assignments were acquired on MAGEA4-MHD (free or in complex with RAD18 R6BD peptide in a 1:3 ratio) $^2H$, $^{13}C$, $^{15}N$ labelled samples. For the acquisition of all spectra, 3D transverse relaxation optimized spectroscopy (TROSY) pulse sequences with $^2H$-decoupling (HNCO, HNCA, HN(CO)CA, CBCA(CO)NH, HNCACB, HN(CA)CO and HN(CO)CACB) (Sattler, 1999; Pervushin et al, 1997; Salzmann et al, 1998) and apodization weighted sampling were used (Simon and Köstler, 2019). The resulting spectra were processed using NMRPipe (Delaglio et al, 1995) and assigned with the program Cara (Keller, 2004). Seventy-five percent of all amides were assigned for the free MAGEA4-MHD and seventy percent for the complex with RAD18 R6BD peptide. Backbone assignments of the MAGEA4-MHD domain alone and of the complex have been deposited in the Biological Magnetic Resonance Bank (BMRB) under accession codes 51823 and 51822, respectively.

For NMR-based interactions studies, the $^2H$, $^{13}C$, $^{15}N$ labelled free MAGEA4-MHD sample from backbone assignments was titrated with 3.0 molar equivalents of unlabelled RAD18-R6BD peptide (residues 340–366). A $^1H$,$^{15}N$-TROSY-HSQC spectrum was recorded for free and saturated MAGEA4-MHD. The peptide stock was highly concentrated to keep dilution effects as small as possible (2.5 mM). For chemical shift perturbation analysis, NMR-FAM Sparky (Lee et al, 2015) was used and the chemical shift perturbations (ppm) (CSP) at a ratio of 1:3 were calculated according to Williamson (2013):

$$CSP(ppm) = \left[ (\Delta H)^2 + (0.2\Delta N)^2 \right]^{1/2}$$

CSPs greater than the average plus the standard deviation of all measured shifts were considered significant. The ratio of MAGEA4-MHD and RAD18-R6BD at which the former is saturated were derived from NMR titrations with a $^{15}N$ labelled MAGEA4-MHD. CSPs were in the slow exchange, therefore we cannot derive a dissociation constant and backbone assignments had to also be performed for the complex.

## His-pulldown assays

Wild-type (WT) and mutants of His-tagged RAD18 were co-expressed with untagged MAGEA4 WT and grown in the same way as for purification, except cells were grown in 50 mL cultures. Following lysis in lysis buffer, each sample was applied to 50 μL TALON® Superflow™ resin and incubated rotating for 1 h at 4 °C. Samples were washed 3 times with lysis buffer and eluted by boiling with 4X laemmli buffer containing beta-mercaptoethanol, before being loaded onto 4–20% Tris glycine gradient gels (Biorad) for SDS PAGE, followed by Western blotting.

## ITC experiments

A 27 amino acid peptide containing amino acids 340–366 of RAD18 was purchased from GenScript and used at 200 μM for ITC binding experiments. Full-length MAGEA4 WT, MAGEA4 M161D, MAGEA4 triple (M161D, I205D and L288D) and RAD6 WT were purified as described in the previous section and concentrated to 20 μM. The ITC experiments were performed using a MicroCal ITC200 (Malvern), with a sample volume of 350 μL and a ligand volume of 75 μL. All experiments were performed at 25 °C and reference power 5. The initial injection volume was 1 μL and all subsequent injections were 2.5 μL at 3 min intervals. All experiments were performed in SEC buffer. Data baseline subtraction, analysis and dissociation constant determination were conducted using Microcal PEAQ-ITC Analysis Software (Malvern).

## Cross-linking

Full-length RAD18/RAD6 and MAGEA4 were expressed and purified as described previously. RAD18/RAD6 was incubated with 0.5 X molar BS3 cross-linker to total lysine residues and the sample was incubated for 10 min at room temperature, shaking. In the case of RAD18/RAD6/MAGEA4, RAD18/RAD6 was incubated with 5 X molar excess MAGEA4 for 1 h at 4 °C prior to the addition of BS3 cross-linker. Samples were then subjected to Trypsin (Savitski et al, 2014) digest in a 1:50 protease to protein ratio and incubated at 37 °C prior to mass spectrometry analysis.

## Ubiquitination assays

For examining RAD18 autoubiquitination in the presence of MAGEA4, 1 μM RAD18/RAD6, 0.3 μM E1, 12 μM ubiquitin and 2.5 mM ATP were incubated in the presence or absence of increasing concentrations of MAGEA4, as specified. The assay testing the effect of the RAD18 R6BD peptide on RAD18 autoubiquitination was conducted in the same way, with the exception that 5 μM MAGEA4 was used and increasing RAD18 R6BD peptide concentrations were added, as indicated. For determining the impact of the RAD18 R6BD peptide on PCNA ubiquitination, 3 μM PCNA was added and MAGEA4 was omitted. The assay involving RAD18 RING and SAP mutants was carried out in the same way but in the absence of the RAD18 R6BD peptide. All reactions were incubated for 30 min at 30 °C. For assays involving MAGEA4, RAD18/RAD6 and MAGEA4 were incubated together on ice for 30 min prior to addition of the other components. For the assay involving single-stranded DNA,

RAD18/RAD6 was first incubated with a 33 oligonucleotide (Appendix Table S2) for 30 min on ice prior to the addition of the other components and the reaction was run for 45 min. For diGly mass spectrometry analysis, 2.5 μM RAD18/RAD6, 0.75 μM E1, 30 μM ubiquitin and 6.25 mM ATP were used and the reaction was performed at 37 °C for 1 h. Samples were run on a SDS PAGE gel, followed by in-gel digestion, as described below.

### In-Gel digestion

Following SDS-PAGE and coomassie staining, the gel streak corresponding to autoubiquitinated RAD18 was excised and subjected to in-gel digestion with trypsin (Savitski et al, 2014) to generate peptides containing the Lys-ε-Gly-Gly (diGLY) remnant. Peptides were extracted from the gel by sonication for 15 min, followed by centrifugation and supernatant collection. A solution of 50:50 water: acetonitrile, 1% formic acid was added for a second extraction and the samples were again sonicated for 15 min, centrifuged and the supernatant was pooled with the first extract. The supernatants were dried down and reconstituted in 10 μL 4% acetonitrile, 1% formic acid in water and analysed by LC-MS/MS.

## Mass spectrometry

For cross-linking mass spectrometry, digested peptides were concentrated and desalted using an OASIS® HLB μElution Plate (Waters) according to manufacturer instructions. Cross-linked peptides were enriched using size exclusion chromatography (Leitner et al, 2012). In brief, desalted peptides were reconstituted with SEC buffer (30% (v/v) ACN in 0.1% (v/v) TFA) and fractionated using a Superdex Peptide PC 3.2/30 column (GE) on a 1200 Infinity HPLC system (Agilent) at a flow rate of 0.05 mL/min. Fractions eluting between 50–70 μl were evaporated to dryness and reconstituted in 30 μl 4% (v/v) ACN in 1% (v/v) FA.

### LC-MS/MS

Collected fractions were analysed by liquid chromatography (LC) - coupled tandem mass spectrometry (MS/MS) using an UltiMate 3000 RSLC nano LC system (Dionex) fitted with a trapping cartridge (μ-Precolumn C18 PepMap 100, 5 μm, 300 μm i.d. × 5 mm, 100 Å) and an analytical column (nanoEase™ M/Z HSS T3 column 75 μm × 250 mm C18, 1.8 μm, 100 Å, Waters). Trapping was carried out with a constant flow of trapping solvent (0.05% trifluoroacetic acid in water) at 30 μL/min onto the trapping column for 6 min. Subsequently, peptides were eluted and separated on the analytical column using a gradient composed of Solvent A (3% DMSO, 0.1% formic acid in water) and solvent B (3% DMSO, 0.1% formic acid in acetonitrile) with a constant flow of 0.3 μL/min. For the diGLY analysis of ubiquitinated RAD18, the percentage of solvent B (0.1% formic acid in acetonitrile, 3% DMSO) was increased from 2% to 8% in 2.2 min, to 23% in 40.9 min, in 5 min from 23 to 38%, followed by an increase of B to 80% in 4 min and a re-equilibration back to 2% B for 6 min. The outlet of the analytical column was coupled directly to an Orbitrap Fusion Lumos (Thermo Scientific, SanJose) mass spectrometer using the nanoFlex source.

The peptides were introduced into the Orbitrap Fusion Lumos via a Pico-Tip Emitter 360 μm OD × 20 μm ID; 10 μm tip (CoAnn Technologies) and an applied spray voltage of 2.1 kV (2.4 kV for DiGly), the instrument was operated in positive mode. The capillary temperature was set at 275 °C. Only charge states of 4–8 were included. The dynamic exclusion was set to 30 s (60 sec for diGLY analysis). and the intensity threshold was 5e4. Full mass scans were acquired for a mass range 350–1700 $m/z$ in profile mode in the orbitrap with resolution of 120,000. The AGC target was set to Standard and the injection time mode was set to Auto. The instrument was operated in data-dependent acquisition (DDA) mode with a cycle time of 3 s between master scans and MSMS scans were acquired in the Orbitrap with a resolution of 30,000, with a fill time of up to 100 ms and a limitation of 2e5 ions (AGC target). A normalized collision energy of 32 (34 for diGLY) was applied. MS2 data was acquired in profile mode. The DiGly samples were processed in the same way with few exceptions, as noted above.

### Data analysis

For cross-linking mass spectrometry, all data were analysed using the cross-linking module in Mass Spec Studio v2.4.0.3524 (Sarpe et al, 2016). Parameters were set as follows: Trypsin (K/R only), charge states 4–8, peptide length 7–50, percent Evalue threshold = 50, MS mass tolerance = 10 ppm, MS/MS mass tolerance = 10, elution width = 0.5 min. BS3 cross-links residue pairs were constrained to K on one end and one of KSTY on the other. Identifications were manually validated, and cross-links with an E-value corresponding to <0.05% FDR were rejected. The data export from the Studio was filtered to retain only cross-links with a unique pair of peptide sequences and a unique set of potential residue sites.

For DiGly analysis, the raw data was processed with Iso-barQuant (Franken et al, 2015) and as search engine Mascot (v2.2.07) was used. Data was searched against the Uniprot *Escherichia coli* database (UP000000625, 4518 entries, including common contaminants and reverse hits, May 2016) with the amino acid sequences of interest added (His-tagged RAD18, His-tagged E1, USP46, RAD6, His-tagged MAGEA4, UBB, UBC, UBD, RL40 and RS27A, all from *homo sapiens*). Carbamidomethyl (C) was set as fixed modification, Acetyl (Protein N-term), Oxidation (M) and GlyGly (K) as variable modifications. The mass error tolerance for the full scan MS spectra was set to 10 ppm, for MS/MS spectra to 0.02 Da. Enzyme was set to Trypsin/P. A maximum of 2 missed cleavages were allowed. A false discovery rate below 0.01 were required on the peptide and protein level. The diGly score represents the Mascot score for peptide identification, a cut-off of 32 is used for a reliable database match of the MS2 spectra, such that peptides are only considered to be reliably identified if their Mascot score (diGly score) is above 32.

## Comparative interaction proteomics

MAGEC1 was cloned into pEGFP-C1 and the MAGEC1 L318D mutant was generated by site-directed mutagenesis with primers listed in Appendix Table S2. 293T (Human embryonic kidney cells-ATCC CRL-3216) (HEK293T) cells were plated at 500,000 cells per 10 cm dish and grown for 48 h. Cells were then transfected with GFP-MAGEC1 WT or GFP-MAGEC1 L318D in a 3:1 ratio of PEI (polyethylenimine) to DNA. Cells were harvested after 48 h and lysed in 50 mM Tris pH 7.5, 150 mM NaCl, 1% Triton X-100, protease inhibitor (cOmplete™, Mini, EDTA-free (Roche)). Cells were incubated for 25 min on ice and then sonicated for 5 min at

4 °C. Cells were centrifuged at 14,000 rpm for 20 min at 4 °C. 500 µg lysate was then added to GFP-Trap® Agarose resin (ChromoTek) and samples were incubated rotating at 4 °C for 2 h. Following washing of the beads with 50 mM Tris pH 7.5, 150 mM NaCl, proteins were eluted by the addition of 2x Laemmli buffer and boiled for 10 min at 95 °C. Samples were spun down and the supernatants were sent for mass spectrometry analysis as described below. Experiments were conducted in triplicate.

### Sample preparation SP3 and TMT labelling, OASIS

Reduction of disulphide bridges in cysteine containing proteins was performed with dithiothreitol (56 °C, 30 min, 10 mM in 50 mM HEPES, pH 8.5). Reduced cysteines were alkylated with 2-chloroacetamide (room temperature, in the dark, 30 min, 20 mM in 50 mM HEPES, pH 8.5). Samples were prepared using the SP3 protocol (Mikulášek et al, 2021; Hughes et al, 2019) and trypsin (sequencing grade, Promega) was added in an enzyme to protein ratio 1:50 for overnight digestion at 37 °C. The next day, peptide recovery in HEPES buffer by collecting supernatant on magnet and combining with second elution wash of beads with HEPES buffer.

Peptides were labelled with TMT10plex (Werner et al, 2014). Isobaric Label Reagent (ThermoFisher) according the manufacturer's instructions. Samples were combined for the TMT10plex and for further sample clean up an OASIS® HLB µElution Plate (Waters) was used. Offline high pH reverse phase fractionation was carried out on an Agilent 1200 Infinity high-performance liquid chromatography system, equipped with a Gemini C18 column (3 µm, 110 Å, 100 × 1.0 mm, Phenomenex) (Reichel et al, 2016).

### LC-MS/MS

An UltiMate 3000 RSLC nano LC system (Dionex) fitted with a trapping cartridge (µ-Precolumn C18 PepMap 100, 5 µm, 300 µm i.d. × 5 mm, 100 Å) and an analytical column (nanoEase™ M/Z HSS T3 column 75 µm × 250 mm C18, 1.8 µm, 100 Å, Waters). Trapping was carried out with a constant flow of trapping solution (0.05% trifluoroacetic acid in water) at 30 µL/min onto the trapping column for 6 min. Subsequently, peptides were eluted via the analytical column running solvent A (0.1% formic acid in water, 3% DMSO) with a constant flow of 0.3 µL/min, with increasing percentage of solvent B (0.1% formic acid in acetonitrile, 3% DMSO). The outlet of the analytical column was coupled directly to an Orbitrap Fusion™ Lumos™ Tribrid™ Mass Spectrometer (Thermo) using the Nanospray Flex™ ion source in positive ion mode. The peptides were introduced into the Fusion Lumos via a Pico-Tip Emitter 360 µm OD × 20 µm ID; 10 µm tip (CoAnn Technologies) and an applied spray voltage of 2.4 kV. The capillary temperature was set at 275 °C. Full mass scan was acquired with mass range 375–1500 $m/z$ in profile mode in the orbitrap with resolution of 120,000. The filling time was set at maximum of 50 ms with a limitation of $4 \times 10^5$ ions. Data-dependent acquisition (DDA) was performed with the resolution of the Orbitrap set to 30,000, with a fill time of 94 ms and a limitation of $1 \times 10^5$ ions. A normalized collision energy of 38 was applied. MS2 data was acquired in profile mode.

### MS data analysis—MSFragger

All raw files were converted to mzmL format using MSConvert from Proteowizard59, using peak picking from the vendor

algorithm. Files were then searched using MSFragger v3.7 (Kong et al, 2017) in Fragpipe v19.1 against the Swissprot Homo sapiens database (20,594 entries) containing common contaminants and reversed sequences. The standard settings of the Fragpipe TMT10 workflow were used. The following modifications were included into the search parameters: Carbamidomethyl (C) and TMT10 (K) (fixed modification), Acetyl (Protein N-term), Oxidation (M) and TMT10 (N-term) (variable modifications). For the full scan (MS1) a mass error tolerance of 10 ppm and for MS/MS (MS2) spectra of 0.02 Da was set. Further parameters were set: Trypsin as protease with an allowance of maximum two missed cleavages and a minimum peptide length of seven amino acids was required. The false discovery rate on peptide and protein level was set to 0.01.

The raw output files of FragPipe (protein.tsv files, (Kong et al, 2017)) were processed using the R programming language (R Development Core Team, 2016). Contaminants were filtered out and only proteins that were quantified with at least two unique peptides were considered for the analysis. 4389 proteins passed the quality control filters. Log2 transformed raw TMT reporter ion intensities were first cleaned for batch effects using the 'removeBatchEffects' function of the limma package (Ritchie et al, 2015) and further normalized using the vsn package (variance stabilization normalization (Huber et al, 2002). Proteins were tested for differential expression with a moderated t-test using the limma package. The replicate information was added as a factor in the design matrix given as an argument to the 'lmFit' function of limma. The t-values (output of limma) were analysed with the 'fdrtool' function of the fdrtool package (Strimmer, 2008) in order to extract $p$-values and false discovery rates (fdr - q-values).

## GST-pulldown competition experiment

His-GST-RAD18 R6BD was purified as described previously and dialysed overnight into a buffer containing 100 mM NaCl, 20 mM HEPES pH 8.0, 0.5 mM TCEP, 10% Glycerol and 0.1% NP-40. His-GST-RAD18 R6BD was incubated with Glutathione Sepharose 4B (Cytiva) for 2 h, rotating at 4 °C. The beads were subsequently washed several times and then blocked with 5% BSA for 1 h, rotating at 4 °C. Following numerous washes, a 3x Molar excess of RAD6 to RAD18 R6BD was then incubated with the beads for 4 h, rotating at 4 °C. Following several washes, the beads were incubated with a 5x Molar excess MAGEA4 to RAD18 R6BD overnight, rotating at 4 °C. The beads were gently pelleted in a table top centrifuge and the supernatant was taken for analysis via SDS PAGE. The beads were then washed several times and proteins were eluted from the beads by the addition of 2x laemmli buffer. Gel samples were boiled for 3 min prior to application to the SDS PAGE gel.

## Glycerol gradient experiment

RAD18/RAD6, RAD18/RAD6/MAGE4, RAD6 or MAGEA4 were added to a 5–20% glycerol gradient and the sample was centrifuged at 4 °C for 18 h at 40,000 rpm in an Ultracentrifuge (Optima LE-80K, Beckman), before being fractionated into 200 µL fractions. In the case of RAD18/RAD6/MAGEA4, RAD18/RAD6 was incubated with 5 X molar excess MAGEA4 prior to addition to the gradient. Samples were run on a SDS PAGE gel and analysed by silver staining using the SilverQuest™ Staining kit (Invitrogen).

## Mass photometry

Mass photometry experiments were performed using an OneMP Mass Photometer (Refeyn). Experiments were performed in SEC buffer. Mass calibrations were conducted with NativeMark$^{TM}$ Protein Std (Invitrogen) diluted in SEC buffer in a 1 in 400 dilution. Data was collected using AcquireMP (Refeyn) and analysis was performed using DiscoverMP (Refeyn). For the RAD18/RAD6/MAGEA4 complex, the sample was first cross-linked with BS3 cross-linker for 30 min on ice, using a 1x ratio of BS3 to total lysine residues in the complex, followed by loading on a S200 3.2/300 analytical column to separate out the cross-linked complex from the uncross-linked proteins. The peak fraction was selected for MP analysis.

## Analytical ultracentrifugation

The sedimentation velocity experiment was performed on the RAD18/RAD6 complex using an analytical ultracentrifuge XLI, at 42,000 rpm, 4 °C, with an Anti-50 rotor (Beckman Coulter, Brea, USA) and double-secter cells of optical path length 1.5, 3, or 12 mm equipped with Sapphire windows (Nanolytics, Potsdam, DE). The sample buffer was the same as the SEC buffer and RAD18/RAD6 was used at a concentration of 0.5 mg/ml. Acquisitions were made using absorbance at 280 nm and interference optics. Data was processed using Redate v 1.0.0 and analysis was conducted with SEDFIT, v 16.36 and Gussi 1.4.2.

## Confocal imaging

Human RAD18 and human MAGEA4 were cloned into pEHA-C1 and pEGFP-C1 vectors, respectively. Mutants were generated using primers listed in Appendix Table S2. 293T (Human embryonic kidney cells-ATCC CRL-3216) cells were grown on poly-D Lysine-coated coverslips. The cells were transfected with pEHA-C1-RAD18/pEHA-C1-RAD18(R51E) using PEI (polyethylenimine). Twenty-four hours after transfection, the cells were washed in PBS and exposed to UV (10 miliJules/cm2) in a Stratalinker UV crosslinker oven (Stratagene). 4 hs after the UV irradiation, cells were fixed with paraformaldehyde (4% (wt/vol) in PBS) for 15 min at room temperature (RT), followed by permeabilization and blocking in 5% BSA containing 0.3% Triton X-100 and immunostained overnight at 4 °C with an antibody against HA-Tag. The next day, cells were washed and further treated with Alexa fluor 555 conjugated anti-rabbit Secondary Antibody for 1 h in the dark in a humidified 37 °C incubator. The cells were then mounted with ProLong antifade mountant with DAPI and imaged with a Leica TCS SP5 confocal microscope. Cells not exposed to UV [HA RAD18/ HA RAD18 R51E (−UV)] were taken as control and processed as described for the +UV counterparts.

For statistical analysis, the total number of transfected cells and the number of transfected cells with UV-irradiation induced RAD18 foci were blind counted manually from 8 different fields for each condition. A total of 63 and 66 cells were counted for HA-RAD18 and HA-RAD18 R51E, respectively. For each image, the percentage of cells with distinct RAD18 ionizing radiation induced foci (IRIF) was calculated. The percentage difference of transfected cells with IRIF between HA RAD18 and HA RAD18 R51E was then calculated using GraphPad prism 9. Statistical significance of the data was determined by an unpaired T test and is marked as **** for $p < 0.0001$.

## RAD18 stability in HEK293T cells

293T (Human embryonic kidney cells-ATCC CRL-3216) cells were first transfected with pEHA-C1-RAD18 and further transfected with either pEGFP-C1, pEGFP-C1-MAGEA4 WT or pEGFP-C1-MAGEA4 triple mutant. 24 h after transfection, cells were harvested and lysed to obtain the soluble fraction. Total protein concentration was measured by Bicinchoninic acid (BCA) assay and equal amounts of protein from each sample were run on a SDS-PAGE gel, followed by transfer to a PVDF membrane. After blocking, the membranes were probed with the relevant antibodies (detailed in the Western blotting section). To monitor endogenous levels of RAD18 in the absence of RAD18 overexpression, the experiment was conducted in a similar manner, with the exception that transfection with HA-RAD18 was omitted.

## Western blotting

Samples were loaded onto 4–20% Tris glycine gradient gels (Biorad) and analysed by western blot (WB) with RAD18 (NB100-61063, Bio-Techne Ltd), PCNA PC10 (sc-56, SantaCruz), Ubiquitin P4D1 (sc-8017, SantaCruz), MAGEA 6C1 (sc-20034, SantaCruz), HA-tag (C29F4, Cell Signalling), GFP (sc-9996, SantaCruz), or GAPDH (sc-32233, SantaCruz) antibody in a solution containing 5% BSA, 0.2% Tween20 and PBS. Following numerous washing steps with 0.2% Tween20 in PBS, blots were incubated with secondary antibodies for 1 h at room temperature and visualized by fluorescence. IRDye® 800CW Goat anti-Rabbit IgG Secondary Antibody (Li-Cor) was used as the secondary antibody to detect RAD18 and HA-tagged RAD18. IRDye® 800CW Donkey anti-Mouse IgG Secondary Antibody (Li-Cor) was used to detect MAGEA4, Ubiquitin, PCNA, GFP and GAPDH.

# Data availability

The Mass Spectrometry data has been deposited in the ProteomeXchange Consortium via the PRIDE (Perez-Riverol et al, 2022) partner repository (https://www.ebi.ac.uk/pride/) with the following identifiers: PXD047929 for the XL-MS experiment, PXD047850 for the diGly experiment and PXD047849 for the Comparative Interaction Proteomics experiment. The backbone assignments of the MAGEA4-MHD domain alone and of the complex have been deposited in the Biological Magnetic Resonance Bank (BMRB) under accession codes 51823 and 51822, respectively.

# Peer review information

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

## Acknowledgements

We thank the EMBL Mass spectrometry core facility, especially Jennifer Jasmin Schwarz and Frank Stein for the data acquisition and analysis, Martin Pelosse for training and maintenance of the EMBL eukaryotic expression facility, Caroline Mas for training and access to the biophysics platform. We thank Aline Le Roy and Christine Ebel, for assistance with the Analytical Ultracentrifugation (AUC) experiments. This work used the platforms of the Grenoble Instruct-ERIC center (ISBG; UAR 3518 CNRS-CEA-UGA-EMBL) within the Grenoble Partnership for Structural Biology (PSB), supported by FRISBI (ANR-10-INBS-0005-02) and GRAL, financed within the University Grenoble Alpes graduate school (Écoles Universitaires de Recherche) CBH-EUR-GS (ANR-17-EURE-0003). The work was supported by a grant from the French Agence Nationale de la Recherche (ANR-21-CE11-0013) to SB.

## Author contributions

**Simonne Griffith-Jones**: Investigation; Writing—review and editing. **Lucía Álvarez**: Investigation. **Urbi Mukhopadhyay**: Investigation. **Sarah Gharbi**: Investigation. **Mandy Rettel**: Investigation. **Michael Adams**: Investigation. **Janosch Hennig**: Investigation. **Sagar Bhogaraju**: Conceptualization; Supervision; Funding acquisition; Visualization; Writing—original draft; Writing—review and editing.

## funding

## Disclosure and competing interests statement

SGJ and SB have submitted a patent application "Method for identifying inhibitors of Rad18" (EP23206838).

# Expanded View Figures

**Figure EV1.   MAGEA4 utilizes residues within the WH2 motif to bind the RAD18 R6BD.**

(**A**) RAD18 R6BD/MAGEA4 AlphaFold2 model coloured by pLDDT score. (**B**) Predicted alignment error (PAE) plot of the RAD18 R6BD/MAGEA4 Alphafold model. (**C**) Size exclusion chromatography profiles of MAGEA4 WT and mutant proteins. (**D**) ITC analysis of the binding between MAGEA4 M161D and the RAD18 R6BD peptide. The ITC experiment was performed three times with similar results. (**E**) NMR analysis of MAGEA4/RAD18 R6BD peptide. Overlay of the $^1$H-$^{15}$N TROSY-HSQC spectra of free [$^2$H,$^{13}$C,$^{15}$N] MAGEA4-MHD (red) and in a 3:1 molar ratio complex with unlabelled RAD18-R6BD (blue). Significant chemical shift perturbations are indicated with an arrow and observed in residues from the WH-B motif. To optimize data acquisition, a reduced $^{15}$N spectral width was used, leading to folding of some peaks which are boxed. Source data are available online for this figure.

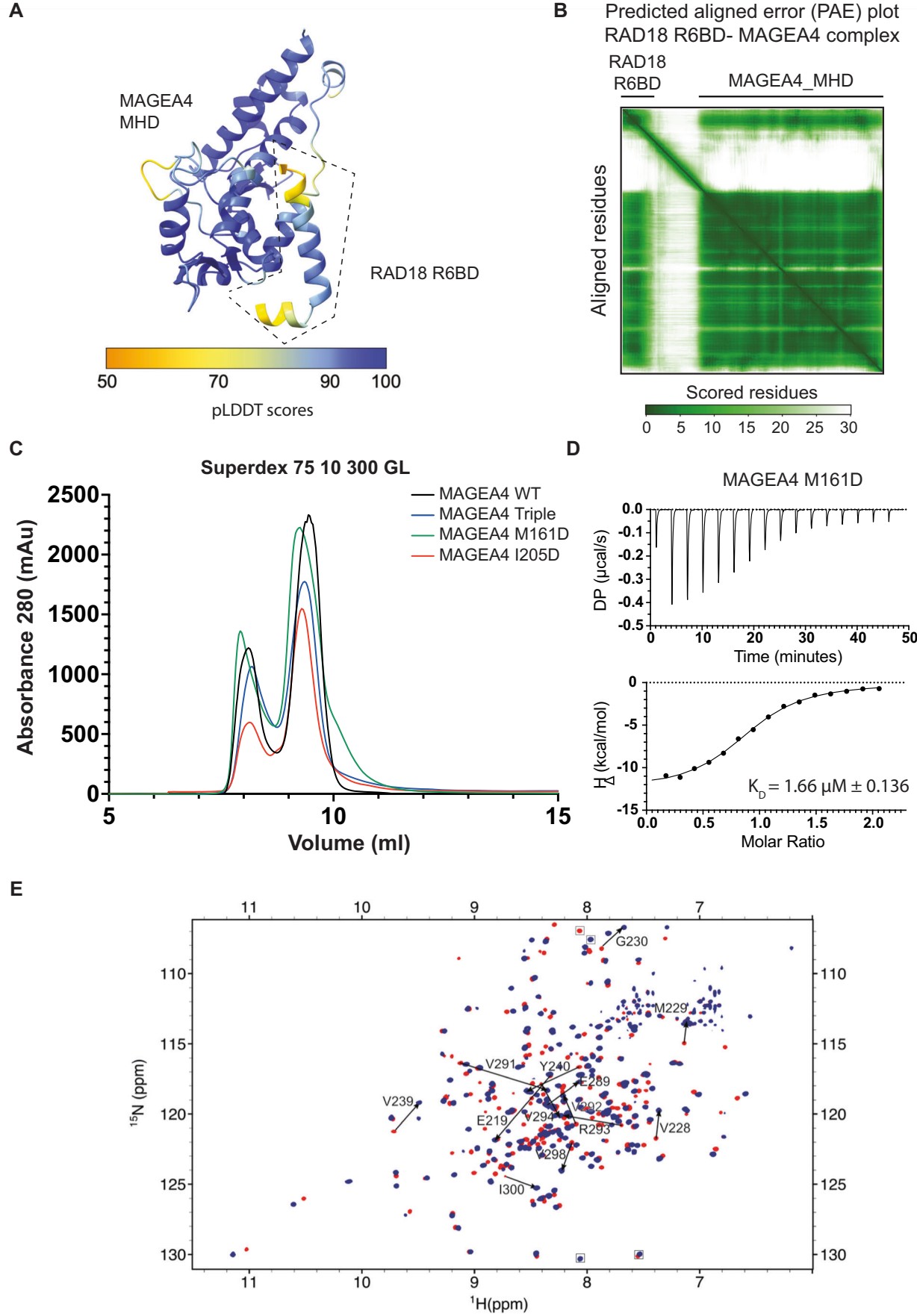

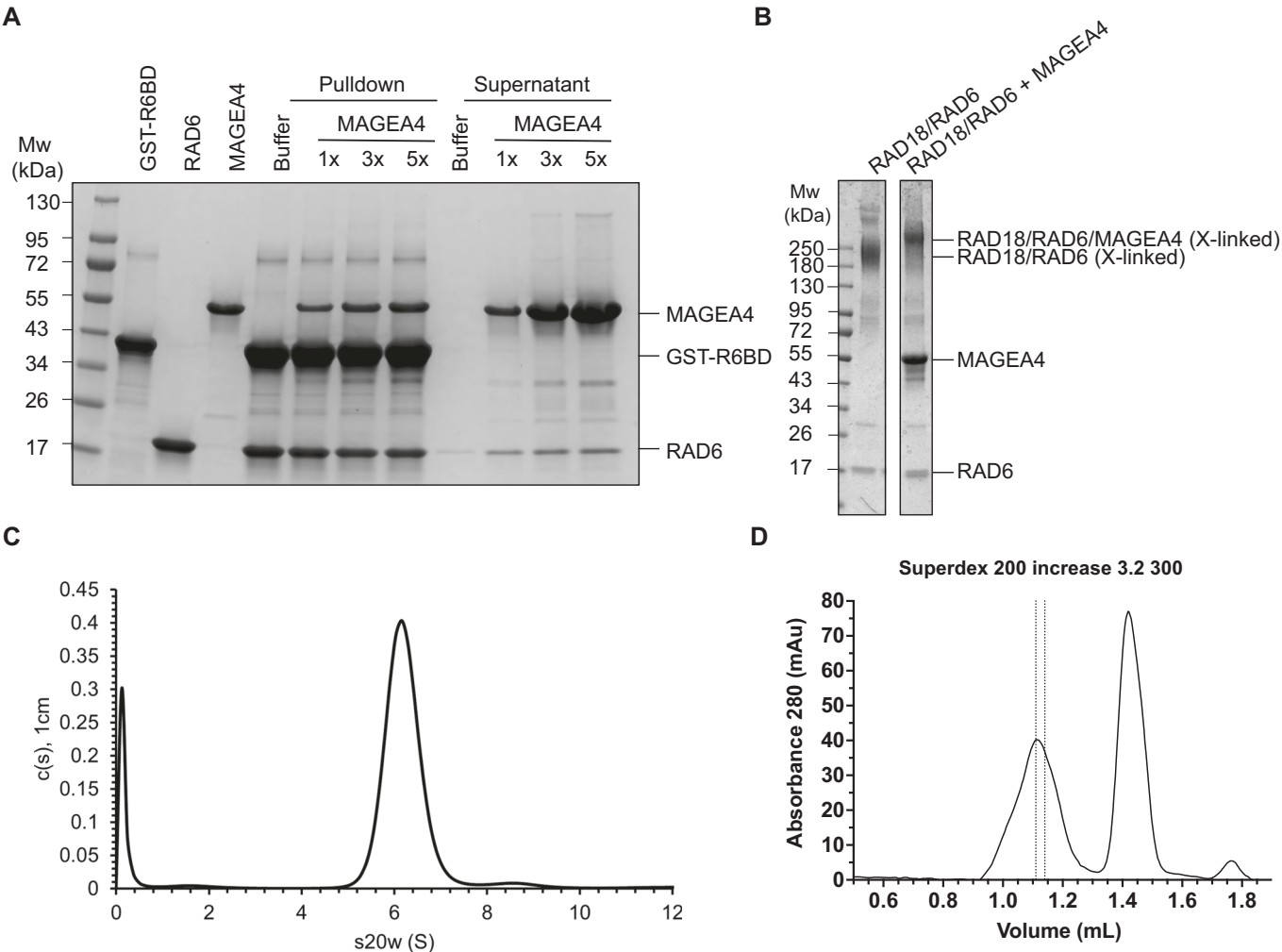

**Figure EV2.  Examining the impact of MAGEA4 binding on the RAD18/RAD6 complex.**

(A) GST-tagged RAD18 R6BD was loaded onto glutathione beads and incubated with a 3x excess of RAD6. The immobilized GST-RAD18 R6BD was then incubated with an increasing concentration of MAGEA4 WT. The supernatant, containing proteins not bound to the GST-RAD18 R6BD, and the pulldown, containing proteins bound to the GST-RAD18 R6BD, from each condition were loaded onto a SDS PAGE gel and subjected to Coomassie staining to monitor RAD6 dissociation from the RAD18 R6BD. (B) Coomassie gel of RAD18/RAD6 cross-linked in the presence or absence of 5x molar MAGEA4. (C) Analytical ultracentrifugation of the RAD18/RAD6 complex. A sedimentation coefficient s20w of 6.2 S was obtained, corresponding to a molecular weight of 131.7 kDa. (D) Size exclusion chromatography profile of RAD18/RAD6/ MAGEA4. The complex was first subjected to BS3 cross-linking, followed by size exclusion chromatography. The peak fraction was taken for analysis by Mass Photometry. Source data are available online for this figure.

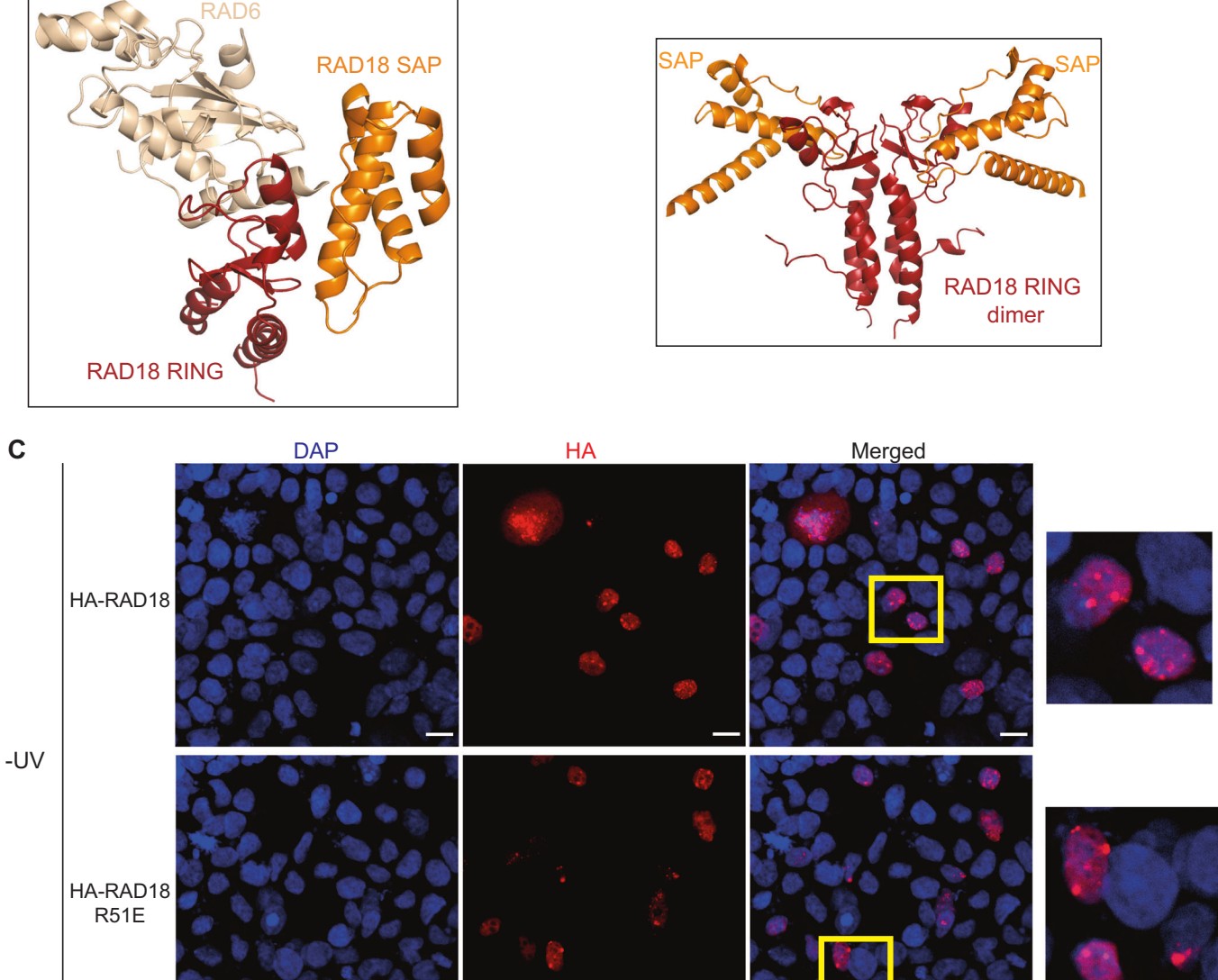

**Figure EV3. Structural analysis of the interaction between the RING and SAP domains of RAD18.**

(A) Model of the RAD18-RING/RAD6/SAP complex. RAD18 RING-SAP model is as shown in Fig. 4B. The RAD18 RING-RAD6 model was created by superposing each molecule on the respective RING (RNF12) and E2 (Ube2e2) of another RING-E2 (PDB: 6W9A). (B) The RING domain of the RAD18 RING/SAP Alphafold model is superposed onto the RING domain of the RAD18 RING domain dimer structure (2Y43). This resulted in the model of a RAD18 dimer containing two RING and two SAP domains without any clashes. (C) HEK293T cells were transfected with WT or R51E HA-RAD18. HA-RAD18 was immunostained using anti-HA antibody and the nucleus was stained using DAPI. Scale bar 12 μm. Source data are available online for this figure.

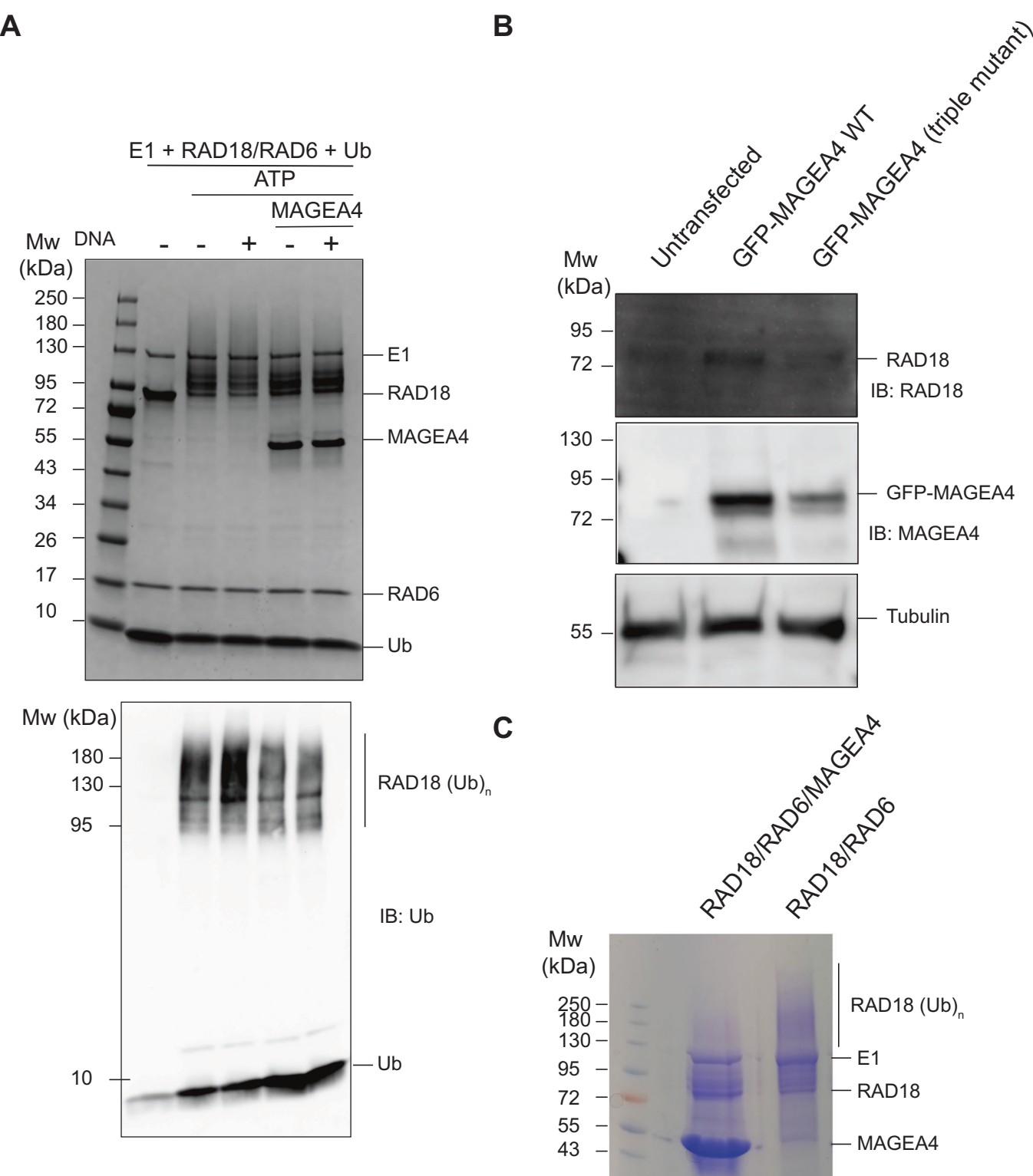

**Figure EV4.  MAGEA4 protects RAD18 from degradative autoubiquitination regardless of DNA.**

(**A**) In vitro ubiquitination reactions performed with the RAD18/RAD6 complex in the presence or absence of a 33 bp single-stranded DNA, with or without a 3x excess of MAGEA4, as indicated. Reactions were subjected to SDS PAGE, followed by Coomassie staining (top) or WB using an antibody against ubiquitin (bottom). (**B**) GFP-tagged MAGEA4 WT or MAGEA4 triple mutant were transfected into HEK293T cells and the levels of endogenous RAD18 were monitored. The soluble fraction was subjected to SDS PAGE and WB using antibodies against RAD18 or GFP. Tubulin levels were monitored as a loading control. (**C**) In vitro ubiquitination reactions were performed using the RAD18/RAD6 complex in the presence or absence of MAGEA4. The reaction mix was analysed using SDS-PAGE followed by Coomassie staining. Autoubiquitinated RAD18 in the RAD18/RAD6 sample, as indicated in the figure, was analysed using mass spectrometry to identify ubiquitination sites. Source data are available online for this figure.

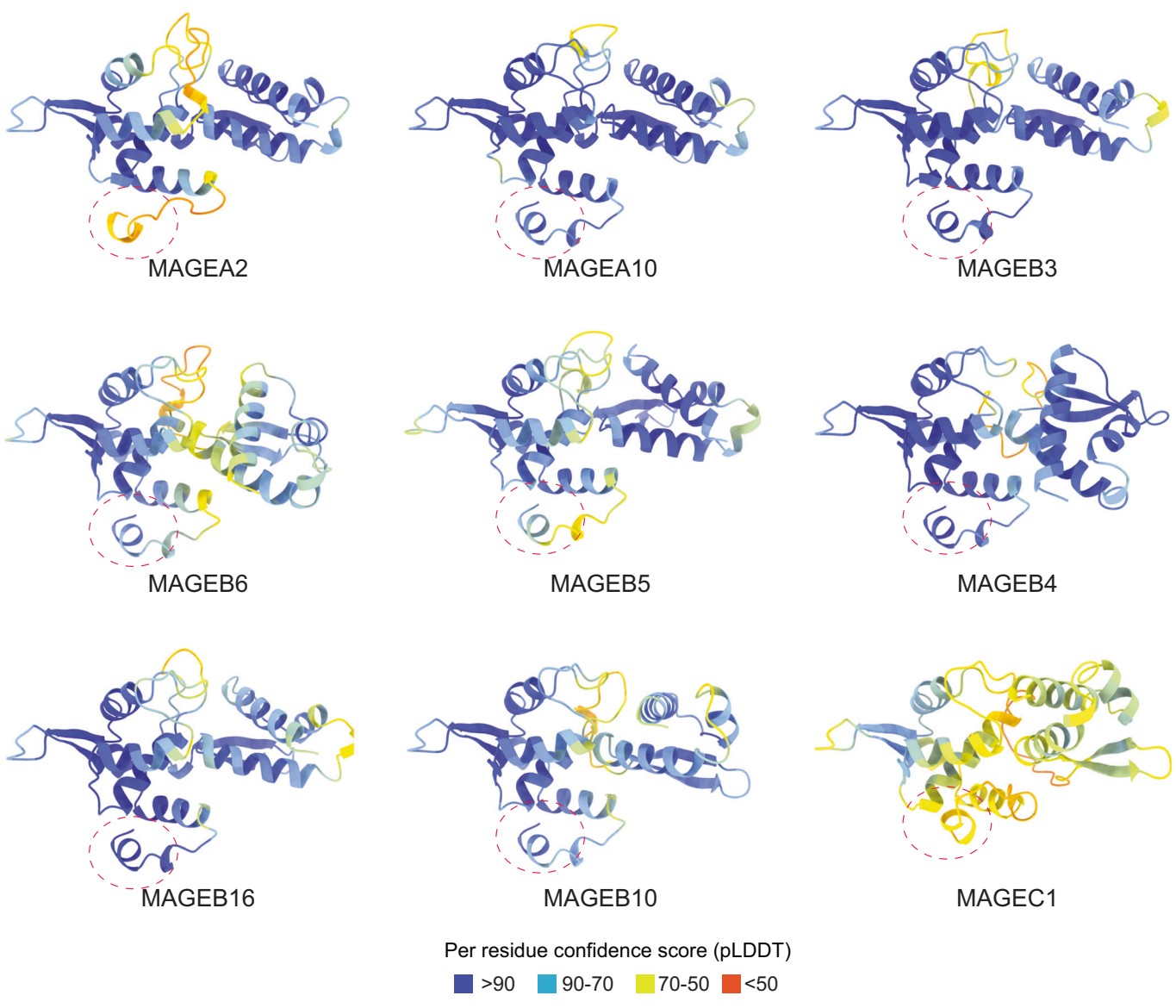

MAGEA2 MAGEA10 MAGEB3 MAGEB6 MAGEB5 MAGEB4 MAGEB16 MAGEB10 MAGEC1

Per residue confidence score (pLDDT)

■ >90  ■ 90-70  ■ 70-50  ■ <50

**Figure EV5. AlphaFold predictions of various Type 1 MAGEs, coloured by pLDDT score.**

The red circle marks the peptide binding groove of various MAGE proteins.

