## [Peer Review File · The EMBO Journal]

Structural basis for RAD18 regulation by MAGEA4 and its implications for RING ubiquitin ligase binding by MAGE family proteins

Sagar Bhogaraju, Simonne Griffith-Jones, Lucía Álvarez, Urbi Mukhopadhyay, Sarah Gharbi, Mandy Rettel, Michael Adams, and Janosch Hennig

Corresponding author(s): Sagar Bhogaraju (bhogaraju@embl.fr)

Review Timeline:

Submission Date:	17th Jul 23
Editorial Decision:	7th Sep 23
Revision Received:	20th Dec 23
Editorial Decision:	30th Jan 24
Revision Received:	6th Feb 24
Accepted:	7th Feb 24

Editor: Hartmut Vodermaier

Transaction Report:

Dear Sagar,

Thank you again for submitting your manuscript on MAGEA1 and Rad18, and my apologies for the delay in its evaluation. We originally sent the manuscript to three expert referees, and granted one of them an extension over the period of my vacation-related absence. Unfortunately, this referee was eventually not able to provide comments even within this extended deadline, leaving us with a set of only two reports, which I am copying below for your information. We have in the meantime discussed these reports in depth within our editorial team, and unfortunately found them to offer insufficient support for publication in The EMBO Journal, at least in the present form. As it stands, both reviewers request further evidence for physiological consequences of MAGEA4-mediated regulation beyond mere effects on Rad18 autoubiquitination. Moreover, referee 2 also raises several substantive concerns with the biochemical analyses and the mechanistic model of regulation itself.

It is not clear if these issues could all be adequately addressed in a straightforward manner during a regular, single-round revision, but I would nevertheless give you a chance to consider the reports, and to get back to me with a tentative revision plan explaining what you might (and might not) be able to add, should you be given the opportunity to revise this work for The EMBO Journal. Based on such a proposal and preliminary point-by-point response to the referees' comments, I could then determine whether a major revision for The EMBO Journal would seem realistic, or whether a less substantively revised (but still mechanistically strengthened) version might at least be suitable for one of our sister journals. I'd be happy to talk through such a revision proposal with you if needed.

Looking forward to hearing from you,

Best regards,

Hartmut Hartmut Vodermaier, PhD

Senior Editor, The EMBO Journal

h.vodermaier@embojournal.org

Referee #1 (Report for Author)

The manuscript by Griffith-Jones, Bhogaraju and colleagues describes a structure-activity study of the ubiquitin ligase Rad18 and its interaction with melanom-associated antigen 4 (MAGE-A4) a cancer-testis antigen and the ubiquitin E2 enzyme Rad6. The study uses AlphaFold2 models to design point mutations to disrupt the interactions and test the consequences in in vitro ubiquitination assays. Cross-linking mass spectrometry (XL-MS) and NMR spectroscopy are used to confirm and extend the AlphaFold2 models toward a model of the ternary complex. The major conclusions are that 1) a Rad6-binding domain of Rad18 binds to the C-terminal WH2 motif of MAGE-A4, 2) this causes a conformational change in Rad18 and affects Rad6 binding, 3) interactions between the RING and SAP domains of Rad18 are important for UV resistance in

HEK293T cells, 4) MAGE-A4 specifically inhibits RAD18 autoubiquitination, and 5) the MAGE-A4 ubiquitin ligase-binding cleft is functional in other type-I MAGE proteins.

The major problem with the manuscript is the absence of results from cellular or animal models to validate MAGE-A4 regulation of Rad18 levels. There is no evidence to show that loss of the MAGE-A4-Rad18 interface has physiological consequences. The paper builds on a previous report (Gao, Nat Comm, 2016) that reported loss of MAGE-A4 decreased Rad18 half-life in cells and MAGE-A4 over-expression decreased Rad18 ubiquitination in cells. The authors of the current work need to extend their results beyond the demonstration of a decrease in Rad18 autoubiquitination. They show that an R51E mutation in Rad18 disrupts PCNA ubiquitination and the response of HEK293T cells to ionizing radiation (Fig 3F). It's unclear why they didn't use similar experiments to demonstrate the effects of the MAGE-A4-Rad18 interaction.

The work also suffers from being a collection of short vignettes rather than a complete story. The validation of the RING-SAP interface in Fig 3 is interesting but doesn't advance the narrative that MAGE-A4 regulates Rad18-mediated ubiquitination of PCNA. Similarly, the presentation of the MAGE-C2-TRIM28 interaction is only tangentially related to Rad18. Coming after the model of MAGE-A4-Rad6-Rad18 complex in Fig 4G, the analysis of other MAGE proteins appears as an afterthought. The topic is underdeveloped and deserves (requires) more experimental validation.

In conclusion, Rad18 is a fascinating ubiquitin ligase but the study doesn't attempt to address the key unresolved questions about the stoichiometry of Rad6 binding and its specificity for monoubiquitination of PCNA. The biological aspects need to be expanded to justify the authors' conclusion that their "data reveal crucial insights into Rad18-mediated ubiquitination of PCNA and its regulation by MAGE-A4."

Specific points.

The displacement of Rad6 by MAGE-A4 binding Rad18 (Fig 2C) is not very convincing. Given the 60-fold higher affinity, it is surprising that a 5-fold excess of MAGE-A4 was needed.

What is the affinity of MAGE-A4 for intact Rad18? How does the complex behave on glycerol gradient. The "higher glycerol concentration" (line 220) of the MAGEA4/Rad18/Rad6 complex needs to be better documented. Analytical ultracentrifugation (AUC) analysis of the complexes might provide more accurate mass estimates and insights into their stoichiometry.

Does Rad18 sediment as a dimer? What is the stoichiometry of Rad6 binding? What is the affinity of Rad6 for full-length Rad18?

Does ubiquitin-charged Rad6 behave the same as the uncharged protein?

It is hard to assess / interpret the XL-MS experiments. Fig 2E shows most of the Rad6-Rad18 crosslinks are still detected (albeit at lower levels) in the presence of MAGE-A4.

The authors are unclear about whether a ternary complex exists. They suggest the absence of crosslinks detected between MAGE-A4 and Rad6 in Fig 2E is evidence of mutually exclusive binding but also state that "Rad6 does not dissociate completely from Rad18 in the presence of MAGEA4."

The decrease in vitro autoubiquitination of Rad18 by MAGE-A4 is clear in Fig 3A but only affects a portion of Rad18. A significant amount (1/3?) of unmodified Rad18 is visible in all conditions. Is this large enough to explain the biological effects of MAGE-A4 on Rad18? The authors should measure Rad18 levels or half-life in cells.

Why are Rad18 foci present in the absence of UV irradiation in Fig S6E?

The authors should use AlphaFold2 to predict the structure of the Rad18 dimer rather than the monomer.

The MS data should be deposited in a publicly accessible database. The di-Gly score should be defined (what does a score of 150 for K462 mean?). How do the scores compare with previously published autoubiquitination sites (Notenboom, NAR 2007)? Do the sites change when MAGEA4 is present or with R51E Rad18 mutant? Does Rad6 or MAGE-A4 become ubiquitinated?

Referee #2 (Report for Author)

In this manuscript, the authors provide a biochemical and biophysical characterisation of the interaction between the human ubiquitin ligase (E3) Rad18 and the E3 regulatory protein MAGEA4. This interaction is critical for the regulation of replication stress tolerance and may have implications for cancer development. A rough picture was previously available, but the authors now provide a large data set with additional detail and develop a definitive hypothesis of how MAGEA4 regulates Rad18 activity. They propose that MAGEA4 competes with the E2, Rad6, for a binding site on Rad18, and the higher affinity of MAGEA4 compared to Rad6 leads to a blocking of Rad18 autoubiquitylation and thus stabilization of the E3. They support their hypothesis by a combination of AlphaFold prediction, NMR, cross-linking mass spectrometry and in vitro assays.

The topic is suitable for the journal and timely. Overall, the authors' hypothesis appears reasonable and may well apply. The manuscript is reasonably well written, and the referee appreciates the breadth of experimental approaches to support the mechanistic claims. That said, some of the most important pieces of evidence on which the authors' model rests are somewhat shaky. In addition, they use and propose a much too simplistic model of the Rad6-Rad18 complex and its actions on the substrate, PCNA. In my opinion, a considerable amount of work would be needed to put their evidence on to more solid ground.

My major concerns are as follows:

1. Rad18 ubiquitylates PCNA while both PCNA and Rad18 itself are bound to a primed piece of ss/dsDNA, where Rad18 is recruited partially via DNA binding and partially via interaction with the ssDNA-binding RPA complex. The domain arrangement of Rad18 and what is known about intramolecular domain-domain interactions (including the authors' own findings) suggest that multiple folded domains are interspersed with less structured connectors. The overall domain arrangement may therefore be quite variable. Thus, interaction with MAGEA4 and the effects of MAGEA4 on Rad18 activity and interaction with Rad6 could be very different depending on whether DNA is involved. This does not necessarily mean that the authors' conclusions are wrong, but it would be important to know whether the pool of Rad18 that is subject to autoubiquitylation,

degradation, and/or MAGEA4-mediated protection is soluble or bound to DNA and/or RPA. The authors should at least make an attempt to approach this important question by studying interactions and MAGEA4 effects also in the presence of DNA and RPA and using loaded PCNA as a ubiquitylation substrate.

2. The other major difference between the authors analysis and the physiological arrangement of Rad18 is its propensity to dimerise. The authors mention this briefly, but they do not appear to make any attempt to consider this in their analysis. I am not a structural biologist and can therefore not fully evaluate the biophysical assays that they use to make claims about domain-domain arrangements and interactions, but I suspect that they may not be able to differentiate between inter- and intramolecular interactions. I therefore do not understand the basis of their conclusions about longer-ranging interactions in their complex. For example, could the WH1 domain make contact with the 'other' Rad18 molecule in the dimer as opposed to the one engaged in interaction with the WH2 domain? There are multiple other analogous scenarios possible, including e.g. the interaction between the RING and SAP domains of Rad18.

3. I do not understand the authors' reasoning with respect to their XL-MS data (Figs. 2D-E and S5B-C). Firstly, they observe that the crosslinks between Rad6 and Rad18 are predominantly not within, but N-terminally of the R6B domain. Do they have an explanation for this? Secondly, in the presence of MAGEA4 the overall number of contacts between Rad6 and Rad18 is reduced, but the RELATIVE number of contacts between Rad6 and the R6B domain actually increases from 9/59 to 1/29. How can the authors conclude from these data that MAGEA4 competes with Rad6 for the R6B domain if the binding appears to become more strongly biased towards that region rather than shifting towards the RING domain?

4. It would strengthen the manuscript considerably if the authors could provide some in cellulo evidence for their model, e.g. an effect of relevant MAGEA4 mutations on Rad18 half-life, TLS efficiency, survival of DNA damage or any other read-out. An approach in cells might also help to clarify whether it is the chromatin-bound (and substrate-engaged) or the free pool of Rad18 that MAGEA4 acts upon.

5. The authors use pull-down assays to demonstrate a reduction in the interactions between Rad18 and MAGEA4 mutants. This is a very crude approach that should probably be complemented by a more quantitative method. The present approach leaves open the possibility that the overall folding and solubility of the mutants is affected.

6. Likewise, the sedimentation approach to show the competition between MAGEA4 and Rad6 is not very convincing. The authors might want to set up more sophisticated competition binding assays.

7. The observation that a human protein is less well produced in E. coli cells is insufficient to postulate a folding problem. It could be a translation efficiency/premature termination issue.

Minor points:

8. It is unclear how exactly the authors define cells with Rad18 foci. There doesn't seem to be an image of a transfected cell where Rad18 doesn't form foci. At least the authors' quantification method should be thoroughly explained (selection of data, thresholding, automated or manual counting, how many cells etc.).

9. The first paragraph is phrased a bit too simplistic; they might want to limit their discussion of E3 function to RING E3s, but they should probably be a bit more detailed with respect to those.

10. The authors should probably contrast the situation of human Rad18 to that in fungi, where regulation appears to be different. They also might want to call the human protein RAD18 rather than Rad18.

Referee #1 (Report for Author)

The manuscript by Griffith-Jones, Bhogaraju and colleagues describes a structure-activity study of the ubiquitin ligase Rad18 and its interaction with melanom-associated antigen 4 (MAGE-A4) a cancer-testis antigen and the ubiquitin E2 enzyme Rad6. The study uses AlphaFold2 models to design point mutations to disrupt the interactions and test the consequences in in vitro ubiquitination assays. Cross-linking mass spectrometry (XL-MS) and NMR spectroscopy are used to confirm and extend the AlphaFold2 models toward a model of the ternary complex. The major conclusions are that 1) a Rad6-binding domain of Rad18 binds to the C-terminal WH2 motif of MAGE-A4, 2) this causes a conformational change in Rad18 and affects Rad6 binding, 3) interactions between the RING and SAP domains of Rad18 are important for UV resistance in HEK293T cells, 4) MAGE-A4 specifically inhibits RAD18 autoubiquitination, and 5) the MAGE-A4 ubiquitin ligase-binding cleft is functional in other type-I MAGE proteins.

The major problem with the manuscript is the absence of results from cellular or animal models to validate MAGE-A4 regulation of Rad18 levels. There is no evidence to show that loss of the MAGE-A4-Rad18 interface has physiological consequences. The paper builds on a previous report (Gao, Nat Comm, 2016) that reported loss of MAGE-A4 decreased Rad18 half-life in cells and MAGE-A4 over-expression decreased Rad18 ubiquitination in cells. The authors of the current work need to extend their results beyond the demonstration of a decrease in Rad18 autoubiquitination. They show that an R51E mutation in Rad18 disrupts PCNA ubiquitination and the response of HEK293T cells to ionizing radiation (Fig 3F). It's unclear why they didn't use similar experiments to demonstrate the effects of the MAGE-A4-Rad18 interaction.

The work also suffers from being a collection of short vignettes rather than a complete story. The validation of the RING-SAP interface in Fig 3 is interesting but doesn't advance the narrative that MAGE-A4 regulates Rad18-mediated ubiquitination of PCNA. Similarly, the presentation of the MAGE-C2-TRIM28 interaction is only tangentially related to Rad18. Coming after the model of MAGE-A4-Rad6-Rad18 complex in Fig 4G, the analysis of other MAGE proteins appears as an afterthought. The topic is underdeveloped and deserves (requires) more experimental validation.

In conclusion, Rad18 is a fascinating ubiquitin ligase but the study doesn't attempt to address the key unresolved questions about the stoichiometry of Rad6 binding and its specificity for monoubiquitination of PCNA. The biological aspects need to be expanded to justify the authors' conclusion that their "data reveal crucial insights into Rad18-mediated ubiquitination of PCNA and its regulation by MAGE-A4."

Specific points.

The displacement of Rad6 by MAGE-A4 binding Rad18 (Fig 2C) is not very convincing. Given the 60-fold higher affinity, it is surprising that a 5-fold excess of MAGE-A4 was needed.

We thank the reviewer for this comment. This apparent discrepancy can be explained by the observations made by us and others. It is important to note that Rad6 interacts with two distant regions in Rad18—namely the RING domain and the Rad6-binding domain (R6BD) (1–3). The 60-fold difference is only when we consider the affinity measure through ITC (Figure 1F and Figure 2B) of interactions between Rad6-R6BD(Rad18) and MAGEA4-R6BD(Rad18). We also discuss as a prelude to the data in Figure 2D (Figure 2C in the previous version of the manuscript) in lines 242–245. In Figure 2D, we used FL Rad18, Rad6 and MAGEA4 proteins in the co-sedimentation assay.

To further strengthen this aspect of our data and complement our ITC experiments (Figure 1F and 2B) and glycerol gradient experiment (Figure 2D), we have performed a competition assay utilizing GST-tagged RAD18 R6BD immobilized on glutathione beads. We have loaded RAD6 onto the immobilized GST-RAD18 R6BD and subjected the complex to a 3x excess of MAGEA4 WT or MAGEA4 triple mutant (Figure 2C). We found that a notable amount of RAD6 dissociated in the presence of MAGEA4 WT, whereas the MAGEA4 triple mutant was unable to displace RAD6 from

the RAD18 R6BD. We found that MAGEA4 was still unable to fully displace RAD6 from the GST-tagged RAD18 R6BD (lines 233-242) (Figure 2C, S7). This was a bit surprising to us because in this case, we did not have the FL RAD18. We think this could be due to the GST tag that was fused to the R6BD which could have influenced the binding with MAGEA4.

This competition aspect is also intricately connected to the stoichiometry of the complex as this reviewer points out in the comment below. We also subjected both RAD18/RAD6 and RAD18/RAD6/MAGEA4 to AUC experiments and we obtained a sedimentation coefficient s_{20w} of 6.2S for RAD18/RAD6 (Figure S9, lines 288-293). However, we were unable to measure the molecular weight of the RAD18/RAD6/MAGEA4 complex accurately through AUC, likely due to the dynamic nature of the complex. Therefore, we employed Mass Photometry, which is another biophysical method to measure the size of macromolecules very accurately. Using mass photometry, we obtained a Mw of 171 kDa for RAD18/RAD6/MAGEA4 (Figure 2G) (lines 295-304). This indicates that we have 2 molecules of His-tagged RAD18 ($58.6 \times 2 = 114$ kDa), 1 molecule of RAD6 (17.3 kDa) and 1 molecule of His-tagged MAGEA4 (37.3 kDa) in this complex.

What is the affinity of MAGE-A4 for intact Rad18? How does the complex behave on glycerol gradient. The "higher glycerol concentration" (line 220) of the MAGEA4/Rad18/Rad6 complex needs to be better documented. Analytical ultracentrifugation (AUC) analysis of the complexes might provide more accurate mass estimates and insights into their stoichiometry.

- We thank the reviewer for this comment. Full length RAD18 is insoluble when expressed without RAD6 or MAGEA4. So it was not possible for us to measure the affinity between MAGEA4 and FL RAD18. But in our ITC experiment shown in Figure 1F, we measure the affinity between the FL MAGEA4 and the peptide of RAD18 which binds to it and determined the Kd as 405 nM. Since we show that this part of RAD18 is the main region interacting with MAGEA4 in Figure 1E, we expect the affinity between FL MAGEA4 and Rad18 to be similar to the one we measured using ITC. The term "higher concentration" has moved to the line 251-252 in the revised manuscript and here we now added the glycerol percentages where the molecules sedimented to avoid the vagueness in the statement.
- We were able to measure the molecular weight of the RAD18/RAD6/MAGEA4 complex accurately using Mass Photometry (Figure 2G) and we determined a Mw of 171 kDa, which is consistent with a tetrameric complex containing two His-tagged RAD18s (58.6×2 kDa), one RAD6 (17.3 kDa) and one His-tagged MAGEA4 (37.3 kDa) (Figure 2G) (lines 295-304).

Does Rad18 sediment as a dimer? What is the stoichiometry of Rad6 binding? What is the affinity of Rad6 for full-length Rad18?

- We thank the reviewer for these questions. RAD18 dimerization was shown previously (4, 5). The RING domain was shown to be responsible for the dimerization of RAD18 later (1). The stoichiometry of RAD18 and RAD6 has been a debated issue because earlier studies (1) indicated that RAD18/RAD6 is a tetramer with two molecules of each present in the complex. But later, it was shown that two FL molecules of RAD18 only bind to one RAD6 monomer (2, 3). The affinity of FL RAD18 to FL RAD6 has not been measured by us or anyone previously, since RAD18 is insoluble without RAD6 and the purification of RAD18 FL separately from RAD6 is necessary to measure the affinity between the two. However, we have measured the affinity of the R6B domain of RAD18 to RAD6 which is 25 μ M (Figure 2B) and previously, Huang et al., have measured the affinity of RAD18 RING domain with RAD6, which measured at a Kd of 35 μ M (2).
- We used Mass Photometry to assess the stoichiometry of our purified RAD18/RAD6 and RAD18/RAD6/MAGEA4 complexes. Our Mass Photometry data indicates that RAD18/RAD6 is a hetero trimer with a measured molecular mass of 128 kDa (Figure 2G). This supports the notion that there are two RAD18 molecules and one RAD6 in the RAD18/RAD6 complex. Our Mass Photometry of RAD18/RAD6/MAGEA4 indicates that RAD18 exists as a dimer in this complex, such that dimeric RAD18 binds to one RAD6 and one MAGEA4 (Figure 2G, lines 295-304). We also performed AUC experiments on

RAD18/RAD6 and found that RAD18 does indeed sediment as a dimer, in line with previous findings as described above. We determined a molecular weight of 131 kDa for RAD18/RAD6, supporting the notion that RAD18 is an asymmetric dimer capable of binding to only one RAD6 molecule (Figure S9, lines 288-293).

Does ubiquitin-charged Rad6 behave the same as the uncharged protein?

- We thank the reviewer for this interesting question. We think RAD6 charged with Ub would have more affinity towards RAD18, since ubiquitin ligases have been shown to interact with both E2 and Ubiquitin during ubiquitination reactions (6). But this is incredibly difficult to measure and in fact it has not been measured for any E2-E3 interaction that we know of. We hope that the reviewer agrees that this aspect, although very interesting, falls outside the scope of our current manuscript.

It is hard to assess / interpret the XL-MS experiments. Fig 2E shows most of the Rad6-Rad18 crosslinks are still detected (albeit at lower levels) in the presence of MAGE-A4.

- We thank the reviewer for raising this issue, we realize that our XL-MS data explanation needs more clarity. So we have included a discussion on especially the non-quantitative nature of the experiment plus how the XL-MS will not tell us whether the cross-links between especially MAGEA4 and RAD18 are from the same molecule of RAD18 or the second molecule in the RAD18 as pointed out by Reviewer 2. Now the XL-MS data is only used to assess which regions of the proteins interact with each other and support our AF predictions and previous findings. We have included these points in lines 274-277, 279-282 and 485-487 of the revised manuscript.

The authors are unclear about whether a ternary complex exists. They suggest the absence of crosslinks detected between MAGE-A4 and Rad6 in Fig 2E is evidence of mutually exclusive binding but also state that "Rad6 does not dissociate completely from Rad18 in the presence of MAGEA4."

- We thank the reviewer for this comment. We would like to draw the attention of the reviewer to the fact that RAD18 has two binding sites to RAD6 (1-3). So even if MAGEA4 displaces RAD6 from R6BD region of RAD18, RAD6 will still bind to the RING domain of RAD18 in a MAGEA4 independent manner. This would explain how RAD6 and MAGEA4 binding to the R6B site of RAD18 is mutually exclusive but they can still be binding to two different sites of RAD18. In the lines 227-229, 242-245, 255-257 we worded exactly this point carefully. Additionally, we have performed Mass Photometry on RAD18/RAD6/MAGEA4 and shown that the complex has a molecular weight of 171 kDa, indicating that two RAD18s, one RAD6 and one MAGEA4 are present within the complex (Figure 2G, lines 295-304).

The decrease in vitro autoubiquitination of Rad18 by MAGE-A4 is clear in Fig 3A but only affects a portion of Rad18. A significant amount (1/3?) of unmodified Rad18 is visible in all conditions. Is this large enough to explain the biological effects of MAGE-A4 on Rad18? The authors should measure Rad18 levels or half-life in cells.

We thank the reviewer for this comment. We agree that only cell-based measurements would answer this question. To measure RAD18 levels directly in cells, we ectopically co-expressed both HA-RAD18 and GFP-MAGEA4 WT or triple mutant in HEK293T cells (Figure 4C). We found that both the endogenous RAD18 levels and HA-RAD18 levels were increased in cells transfected with MAGEA4 WT, compared to cells transfected with MAGEA4 triple mutant or GFP-alone. Although MAGEA4 triple mutant expressed comparatively less well compared to MAGEA4 WT in our experiments, RAD18 levels are much lower in conditions where MAGEA4 triple mutant was expressed compared to the GFP only control, therefore, our experiment is able to show that the MAGEA4 interaction is important for RAD18 stabilization

in cells. We also performed the same experiment without the exogenous expression of HA-RAD18 and monitored the endogenous RAD18 levels (Figure S13). This experiment resulted in similar findings. These results are now described in lines 389-396.

Why are Rad18 foci present in the absence of UV irradiation in Fig S6E?

- We thank the reviewer for this comment. In the absence of UV, most of the RAD18 is still nuclear and localizes to large puncta with only a partial co-localization with PCNA. And upon UV treatment, RAD18 is dispersed into smaller puncta and exhibits perfect co-localization with PCNA. This has been previously observed as well (7) and is also found in the RAD18 antibody manufacturers' page here.

The authors should use AlphaFold2 to predict the structure of the Rad18 dimer rather than the monomer.

- We thank the reviewer for this comment. We have now predicted RAD18 dimer using Alphafold and used that for our analysis of RING-SAP interaction. (Figure S1).

The MS data should be deposited in a publicly accessible database. The di-Gly score should be defined (what does a score of 150 for K462 mean?). How do the scores compare with previously published autoubiquitination sites (Notenboom, NAR 2007)? Do the sites change when MAGEA4 is present or with R51E Rad18 mutant?

- The diGly scores represent the Mascot score for peptide identification, our mass spectrometry experts have decided on a diGly score of 32 as the cut-off for a reliable database match of the MS2 spectra for a modified peptide We have added this definition to the materials and methods section and we have submitted all our MS data to the PRIDE database with the following identifiers: PXD047850 for the diGly experiment, PXD047849 for the Comparative interaction proteomics experiment with MAGE-C2 and PDX047929 for the XL-MS experiment. We have added these identifiers to the data availability section. We have compared the ubiquitination sites that we identified to those already reported in the literature (lines 401-406).
- Reviewer login accounts for access to the MS data (currently private until online publication of the manuscript) are as follows:
Comparative Interaction Proteomics with MAGE-C2
-Username: reviewer_pxd047849@ebi.ac.uk
-Password: dNgTCzPI
diGly experiment for identifying sites of ubiquitination:
-Username: reviewer_pxd047850@ebi.ac.uk
-Password: WZdmB0vC
XL-MS experiment:
-Username: reviewer_pxd047929@ebi.ac.uk
-Password: sHV3wrv4

Does Rad6 or MAGE-A4 become ubiquitinated?

- We do not observe any notable ubiquitination of RAD6 or MAGEA4 in our experiments (Figure 4).

Referee #2 (Report for Author)

In this manuscript, the authors provide a biochemical and biophysical characterisation of the interaction between the human ubiquitin ligase (E3) Rad18 and the E3 regulatory protein MAGEA4. This interaction is critical for the regulation of replication stress tolerance and may have implications for cancer development. A rough picture was previously available, but the authors now provide a large data set with additional detail and develop a definitive hypothesis of how MAGEA4 regulates Rad18

activity. They propose that MAGEA4 competes with the E2, Rad6, for a binding site on Rad18, and the higher affinity of MAGEA4 compared to Rad6 leads to a blocking of Rad18 autoubiquitylation and thus stabilization of the E3. They support their hypothesis by a combination of AlphaFold prediction, NMR, cross-linking mass spectrometry and in vitro assays.

The topic is suitable for the journal and timely. Overall, the authors' hypothesis appears reasonable and may well apply. The manuscript is reasonably well written, and the referee appreciates the breadth of experimental approaches to support the mechanistic claims. That said, some of the most important pieces of evidence on which the authors' model rests are somewhat shaky. In addition, they use and propose a much too simplistic model of the Rad6-Rad18 complex and its actions on the substrate, PCNA. In my opinion, a considerable amount of work would be needed to put their evidence on to more solid ground.

My major concerns are as follows:

1. Rad18 ubiquitylates PCNA while both PCNA and Rad18 itself are bound to a primed piece of ss/dsDNA, where Rad18 is recruited partially via DNA binding and partially via interaction with the ssDNA-binding RPA complex. The domain arrangement of Rad18 and what is known about intramolecular domain-domain interactions (including the authors' own findings) suggest that multiple folded domains are interspersed with less structured connectors. The overall domain arrangement may therefore be quite variable. Thus, interaction with MAGEA4 and the effects of MAGEA4 on Rad18 activity and interaction with Rad6 could be very different depending on whether DNA is involved. This does not necessarily mean that the authors' conclusions are wrong, but it would be important to know whether the pool of Rad18 that is subject to autoubiquitylation, degradation, and/or MAGEA4-mediated protection is soluble or bound to DNA and/or RPA. The authors should at least make an attempt to approach this important question by studying interactions and MAGEA4 effects also in the presence of DNA and RPA and using loaded PCNA as a ubiquitylation substrate.

- We thank the reviewer for this important comment. It is indeed true that RAD18-mediated ubiquitination of PCNA itself is heavily dependent on the DNA and RPA coating the single stranded DNA. However, it was shown by Gao et al., 2018 (Figure 3C in the reference) that MAGEA4 always remains soluble irrespective of the UV treatment, whereas RAD18 forms clear nuclear puncta upon UV irradiation of the cells (8). This showed that MAGEA4 affects the soluble form of Rad18 which is not interacting with chromatin. **This is now mentioned in lines 372-374.** Moreover, it was also shown that auto ubiquitinated or ubiquitin fused Rad18 stays soluble even under UV treatment conditions, indicating Rad18 ubiquitination also likely happens in the soluble form and retains the molecule there (9) (Figure 7 in Zeman et al., 2014).
- We performed the in vitro ubiquitination assays in the presence of DNA to see how the RAD18 interaction with the DNA affects the binding of MAGEA4 to Rad18. **We have performed an in vitro ubiquitination assay in the presence of single stranded DNA (RAD18 was shown to bind to single stranded DNA more tightly than other forms (10) and found that MAGEA4 still offered protection of RAD18 from autoubiquitination, indicating that DNA does not have any notable impact on MAGEA4 binding to RAD18 (Figure S12, lines 376-380).**
- To measure RAD18 levels directly in cells, we ectopically co-expressed both HA-RAD18 and GFP-MAGEA4 WT or triple mutant in HEK293T cells (Figure 4C). We found that the levels of soluble endogenous RAD18 and the overexpressed HA-RAD18 levels were increased in cells transfected with MAGEA4 WT, compared to cells transfected with MAGEA4 triple mutant or GFP-alone. Although the MAGEA4 triple mutant expressed comparatively less well compared to MAGEA4 WT in our experiments, RAD18 levels are much lower in conditions where MAGEA4 triple mutant was expressed compared to the GFP only control, therefore, our experiment is able to show that the MAGEA4 interaction is important for RAD18 stabilization in cells. We also performed the same experiment without the exogenous expression of HA-RAD18 and monitored the endogenous RAD18 levels (Figure S13). This experiment resulted in similar findings. These results are now described in lines (389-396).

2. The other major difference between the authors analysis and the physiological arrangement of Rad18 is its propensity to dimerise. The authors mention this briefly, but they do not appear to make any attempt to consider this in their analysis. I am not a structural biologist and can therefore not fully evaluate the biophysical assays that they use to make claims about domain-domain arrangements and interactions, but I suspect that they may not be able to differentiate between inter- and intramolecular interactions. I therefore do not understand the basis of their conclusions about longer-ranging interactions in their complex. For example, could the WH1 domain make contact with the 'other' Rad18 molecule in the dimer as opposed to the one engaged in interaction with the WH2 domain? There are multiple other analogous scenarios possible, including e.g. the interaction between the RING and SAP domains of Rad18.

- We thank the reviewer for this comment. Indeed, these interactions between MAGEA4 and the “other” Rad18 molecule are possible explanations for some of the XL-MS data that we have in the manuscript. We have included these points in the revised manuscript, in lines 274-277, 326-329 and 485-487 to shed light on these possible inter and intra molecular interactions. But we hope the reviewer agrees that these possible alternative interactions of WH-A of MAGEA4 do not change any major conclusions of the paper.

3. I do not understand the authors' reasoning with respect to their XL-MS data (Figs. 2D-E and S5B-C). Firstly, they observe that the crosslinks between Rad6 and Rad18 are predominantly not within, but N-terminally of the R6B domain. Do they have an explanation for this? Secondly, in the presence of MAGEA4 the overall number of contacts between Rad6 and Rad18 is reduced, but the RELATIVE number of contacts between Rad6 and the R6B domain actually increases from 9/59 to 1/29. How can the authors conclude from these data that MAGEA4 competes with Rad6 for the R6B domain if the binding appears to become more strongly biased towards that region rather than shifting towards the RING domain?

We thank the reviewer for this comment.

- We could think of a few possible reasons why RAD6 predominantly gets crosslinked to just the N-terminus of R6B domain. In Figure 1A and Figure 2E we labelled the domain boundaries (for R6B it is 340-395) based on previous publications (1), however, based on the AF predictions, it is likely that R6B is from 330-372. We have now changed all the domain boundary labels consistent with AF predictions, aside from the RING domain and ZnF which were already well defined experimentally (2, 11). This reassignment of R6B domain boundaries explains some of the cross-links (Figure. 2) The R6B domain is extremely small (<10%) compared to the size of RAD18 and, moreover, in the RAD18/RAD6 complex, the R6B domain is buried at the interaction interface (12). So the likelihood of forming chemical cross-links is low when the residues are already engaged in a tight interaction. Whereas the region N-terminus to the R6B domain of Rad18 is predicted to be flexible and can easily be in close contact with Rad18 and more importantly available for cross-linking. We have added this point in lines 265-268 to accurately describe our data.
- For the second part of the question, we consulted our mass spectrometry expert colleagues to find an answer. And it seems that since these XL-MS experiments using RAD18/RAD6 complexes with and without MAGEA4 are performed separately and analyzed independently in mass spectrometry, it is not recommended to compare this fraction of cross-links found in one experiment vs another. So, we have removed one instance of comparison (lines 238-241 in the previous version of the manuscript) where we talked about the overall reduced number of cross-links between Rad18/Rad6 in the presence of MAGEA4. In addition, we have included the non-quantitative nature of our mass spectrometry data in lines 279-282. We also understand that the placement of numbers representing the number of cross-links in panels E and F of Figure 2 could lead to direct comparisons even if it is not done in the text. Therefore, we have replaced these panels with diagrams showing all cross-links identified (previously in Figure S5).

4. It would strengthen the manuscript considerably if the authors could provide some in cellulo evidence for their model, e.g. an effect of relevant MAGEA4 mutations on Rad18 half-life, TLS efficiency, survival of DNA damage or any other read-out. An approach in cells might also help to clarify whether it is the chromatin-bound (and substrate-engaged) or the free pool of Rad18 that MAGEA4 acts upon.

We thank the reviewer for this suggestion. To measure RAD18 levels directly in cells, we ectopically co-expressed both HA-RAD18 and GFP-MAGEA4 WT or triple mutant in HEK293T cells (Figure 4C). We found the levels of soluble endogenous RAD18 and the overexpressed HA-RAD18 levels were increased in cells transfected with MAGEA4 WT, compared to cells transfected with MAGEA4 triple mutant or GFP-alone. Although the MAGEA4 triple mutant expressed comparatively less well compared to MAGEA4 WT in our experiments, RAD18 levels are much lower in conditions where MAGEA4 triple mutant was expressed compared to the GFP only control, therefore, our experiment is able to show that the MAGEA4 interaction is important for RAD18 stabilization in cells. We also performed the same experiment without the exogenous expression of HA-RAD18 and monitored the endogenous RAD18 levels (Figure S13). This experiment resulted in similar findings. These results are now described in lines (389-396). We direct the reviewer to please also refer to our response to the first question regarding the effect of MAGEA4 on chromatin-bound or soluble RAD18.

5. The authors use pull-down assays to demonstrate a reduction in the interactions between Rad18 and MAGEA4 mutants. This is a very crude approach that should probably be complemented by a more quantitative method. The present approach leaves open the possibility that the overall folding and solubility of the mutants is affected.

- We thank the reviewer for this comment. We would like to draw reviewer's attention to the isothermal titration calorimetry (ITC) data presented in Figure 1F, G and Figure S4 (previously S2) where we quantified the affinity between MAGEA4 WT and mutants and Rad18 R6B.

6. Likewise, the sedimentation approach to show the competition between MAGEA4 and Rad6 is not very convincing. The authors might want to set up more sophisticated competition binding assays.

- We thank the reviewer for this comment. We have performed Mass Photometry on RAD18/RAD6/MAGEA4 and determined a molecular weight of 171 kDa, in line with two RAD18s, one RAD6 and one MAGEA4 (Figure 2G, lines 295-304). Although we were able to analyse RAD18/RAD6 by AUC, we found that the dynamic nature of the RAD18/RAD6/MAGEA4 complex rendered the complex unsuitable for analysis by AUC. We performed alternate competition assays where we immobilized the GST-tagged R6B domain of RAD18 and first incubated with RAD6 and then MAGE-A4 (Figure 2C, Figure S7, lines 233-242). This competition assay also showed that WT MAGEA4 is able to outcompete RAD6 but not the triple mutant MAGEA4, which is deficient in binding to RAD18.

7. The observation that a human protein is less well produced in E. coli cells is insufficient to postulate a folding problem. It could be a translation efficiency/premature termination issue.

We agree with this comment, we purified the MAGEA4 L121/122A mutant and found that the protein eluted in the void volume during size exclusion chromatography, indicating that the protein is aggregated (Figure S5), likely due to misfolding from mutating these residues buried within the core of the MHD. Additionally, we have analyzed the crystal structures of MAGEA3, MAGEA11 and MAGEB1 and found that the dileucine motif of these MAGEs also lies within the hydrophobic core of the MHD, indicating that mutating the dileucine motif of these proteins likely also results in instability of the MHD (Figure S5). The corresponding lines in the manuscript are 185-198.

Minor points:

8. It is unclear how exactly the authors define cells with Rad18 foci. There doesn't seem to be an image of a transfected cell where Rad18 doesn't form foci. At least the authors' quantification method should be thoroughly explained (selection of data, thresholding, automated or manual counting, how many cells etc.).

We thank the reviewer for this comment. In response to DNA damage, many proteins of the DNA damage response pathways e.g. BRCA1 form distinct nuclear puncta (13) which have been termed ionizing radiation induced foci (IRIF). RAD18 was also shown to form small IRIF upon exposure to UV radiation, since RAD18 is recruited to sites of DNA damage (14). We observe these IRIF clearly in Figure 3F with RAD18 WT but much less so with RAD18 R51E. In the absence of UV radiation, RAD18 still forms nuclear puncta but these are larger and fewer in number, since RAD18 is not chromatin-bound. In Figure S11C we show HEK cells transfected with HA-RAD18 WT in the absence of UV and we can see that the RAD18 puncta are more diffuse. We agree that this needed to be further clarified within the text and we do so in lines 341-344. We have performed a manual blinded counting of cells with foci. For the statistical analysis comparing IRIF following exposure to UV, we collected 8 fields of view each for HA-RAD18 WT and HA-RAD18 R51E. The total number of transfected cells counted was 63 for HA-RAD18 WT and 66 for HA-RAD18 R51E. For each image, we calculated the percentage of cells displaying the distinct RAD18 IRIF and performed an unpaired T test using GraphPad Prism to compare the percentage difference between the two conditions. We have added these details to the methods section of the revised manuscript.

9. The first paragraph is phrased a bit too simplistic; they might want to limit their discussion of E3 function to RING E3s, but they should probably be a bit more detailed with respect to those.

- We have made the first paragraph more detailed (lines 70-76).

10. The authors should probably contrast the situation of human Rad18 to that in fungi, where regulation appears to be different. They also might want to call the human protein RAD18 rather than Rad18.

- We thank the reviewer for this suggestion, we have changed the text accordingly.

References:

1. V. Notenboom, R. G. Hibbert, S. E. Van Rossum-Fikkert, J. V. Olsen, M. Mann, T. K. Sixma, Functional characterization of Rad18 domains for Rad6, ubiquitin, DNA binding and PCNA modification. *Nucleic Acids Research* **35**, 5819–5830 (2007).
2. A. Huang, R. G. Hibbert, R. N. De Jong, D. Das, T. K. Sixma, R. Boelens, Symmetry and Asymmetry of the RING–RING Dimer of Rad18. *Journal of Molecular Biology* **410**, 424–435 (2011).
3. Y. Masuda, M. Suzuki, H. Kawai, F. Suzuki, K. Kamiya, Asymmetric nature of two subunits of RAD18, a RING-type ubiquitin ligase E3, in the human RAD6A–RAD18 ternary complex. *Nucleic Acids Research* **40**, 1065–1076 (2012).
4. H. D. Ulrich, Two RING finger proteins mediate cooperation between ubiquitin-conjugating enzymes in DNA repair. *The EMBO Journal* **19**, 3388–3397 (2000).

5. S. Miyase, S. Tateishi, K. Watanabe, K. Tomita, K. Suzuki, H. Inoue, M. Yamaizumi, Differential Regulation of Rad18 through Rad6-dependent Mono- and Polyubiquitination. *Journal of Biological Chemistry* **280**, 515–524 (2005).
6. A. Plechanovová, E. G. Jaffray, M. H. Tatham, J. H. Naismith, R. T. Hay, Structure of a RING E3 ligase and ubiquitin-loaded E2 primed for catalysis. *Nature* **489**, 115–120 (2012).
7. K. Watanabe, S. Tateishi, M. Kawasuji, T. Tsurimoto, H. Inoue, M. Yamaizumi, Rad18 guides poleta to replication stalling sites through physical interaction and PCNA monoubiquitination. *EMBO J* **23**, 3886–3896 (2004).
8. Y. Gao, E. Mutter-Rottmayer, A. M. Greenwalt, D. Goldfarb, F. Yan, Y. Yang, R. C. Martinez-Chacin, K. H. Pearce, S. Tateishi, M. B. Major, C. Vaziri, A neomorphic cancer cell-specific role of MAGE-A4 in trans-lesion synthesis. *Nat Commun* **7**, 12105 (2016).
9. M. K. Zeman, J.-R. Lin, R. Freire, K. A. Cimprich, DNA damage-specific deubiquitination regulates Rad18 functions to suppress mutagenesis. *Journal of Cell Biology* **206**, 183–197 (2014).
10. Y. Tsuji, K. Watanabe, K. Araki, M. Shinohara, Y. Yamagata, T. Tsurimoto, F. Hanaoka, K. Yamamura, M. Yamaizumi, S. Tateishi, Recognition of forked and single-stranded DNA structures by human RAD18 complexed with RAD6B protein triggers its recruitment to stalled replication forks. *Genes Cells* **13**, 343–354 (2008).
11. A. A. Rizzo, P. E. Salerno, I. Bezsonova, D. M. Korzhnev, NMR Structure of the Human Rad18 Zinc Finger in Complex with Ubiquitin Defines a Class of UBZ Domains in Proteins Linked to the DNA Damage Response. *Biochemistry* **53**, 5895–5906 (2014).
12. R. G. Hibbert, A. Huang, R. Boelens, T. K. Sixma, E3 ligase Rad18 promotes monoubiquitination rather than ubiquitin chain formation by E2 enzyme Rad6. *Proc. Natl. Acad. Sci. U.S.A.* **108**, 5590–5595 (2011).
13. T. T. Paull, E. P. Rogakou, V. Yamazaki, C. U. Kirchgessner, M. Gellert, W. M. Bonner, A critical role for histone H2AX in recruitment of repair factors to nuclear foci after DNA damage. *Current Biology* **10**, 886–895 (2000).
14. K. Watanabe, K. Iwabuchi, J. Sun, Y. Tsuji, T. Tani, K. Tokunaga, T. Date, M. Hashimoto, M. Yamaizumi, S. Tateishi, RAD18 promotes DNA double-strand break repair during G1 phase through chromatin retention of 53BP1. *Nucleic Acids Research* **37**, 2176–2193 (2009).

Dr. Sagar Bhogaraju
EMBL Grenoble
Grenoble
France

7th Sep 2023

Re: EMBOJ-2023-115024
Insights into The Role of MAGEA4 in Rad18 Regulation: Implications across the MAGE Protein Family

Dear Sagar,

Thank you for sending me your tentative point-by-point response to the referee reports on your Rad18/MAGE-A4 study. I have now had a chance to consider your detailed answers and plans for revising this work. I was happy to read that you seem to be overall in a good position to adequately address the key issues raised, and would therefore invite you to prepare and resubmit a manuscript revised and extended along the lines proposed in your draft response. Please do keep me updated in case of unexpected problems with the revision work, or in case you should need an extension of the default three-month revision period after all.

I should remind you that our policy to allow only a single round of (major) revision makes it important to carefully revise and answer all points raised to the referees' satisfaction at this point. Competing manuscript published during the course of your revision will have no negative impact on our final decision on your study. Finally, please note the additional information and more detailed guidelines on how to prepare a revision below (and in our online Guide to Authors) - closely adhering to them shall greatly facilitate the editorial process at the time of resubmission.

Thank you again for the opportunity to consider this work, and I look forward to receiving your revision in due time.

With kind regards,

Hartmut

9) Digital image enhancement is acceptable practice, as long as it accurately represents the original data and conforms to community standards. If a figure has been subjected to significant electronic manipulation, this must be clearly noted in the figure legend and/or the 'Materials and Methods' section. The editors reserve the right to request original versions of figures and the original images that were used to assemble the figure. Finally, we generally encourage uploading of numerical as well as gel/blot image source data; for details see: embopress.org/page/journal/14602075/authorguide#sourcedata

At EMBO Press, we ask authors to provide source data for the main manuscript figures. Our source data coordinator will contact you to discuss which figure panels we would need source data for and will also provide you with helpful tips on how to upload and organize the files.

In the interest of ensuring the conceptual advance provided by the work, we recommend submitting a revision within 3 months (6th Dec 2023). Please discuss the revision progress ahead of this time with the editor if you require more time to complete the revisions. Use the link below to submit your revision:

Link Not Available

Referee #1:

The manuscript by Griffith-Jones, Bhogaraju and colleagues describes a structure-activity study of the ubiquitin ligase Rad18 and its interaction with melanom-associated antigen 4 (MAGE-A4) a cancer-testis antigen and the ubiquitin E2 enzyme Rad6. The study uses AlphaFold2 models to design point mutations to disrupt the interactions and test the consequences in in vitro ubiquitination assays. Cross-linking mass spectrometry (XL-MS) and NMR spectroscopy are used to confirm and extend the AlphaFold2 models toward a model of the ternary complex. The major conclusions are that 1) a Rad6-binding domain of Rad18 binds to the C-terminal WH2 motif of MAGE-A4, 2) this causes a conformational change in Rad18 and affects Rad6 binding, 3) interactions between the RING and SAP domains of Rad18 are important for UV resistance in HEK293T cells, 4) MAGE-A4 specifically inhibits RAD18 autoubiquitination, and 5) the MAGE-A4 ubiquitin ligase-binding cleft is functional in other type-I MAGE proteins.

The major problem with the manuscript is the absence of results from cellular or animal models to validate MAGE-A4 regulation of Rad18 levels. There is no evidence to show that loss of the MAGE-A4-Rad18 interface has physiological consequences. The paper builds on a previous report (Gao, Nat Comm, 2016) that reported loss of MAGE-A4 decreased Rad18 half-life in cells and MAGE-A4 over-expression decreased Rad18 ubiquitination in cells. The authors of the current work need to extend their results beyond the demonstration of a decrease in Rad18 autoubiquitination. They show that an R51E mutation in Rad18 disrupts PCNA ubiquitination and the response of HEK293T cells to ionizing radiation (Fig 3F). It's unclear why they didn't use similar experiments to demonstrate the effects of the MAGE-A4-Rad18 interaction.

The work also suffers from being a collection of short vignettes rather than a complete story. The validation of the RING-SAP interface in Fig 3 is interesting but doesn't advance the narrative that MAGE-A4 regulates Rad18-mediated ubiquitination of PCNA. Similarly, the presentation of the MAGE-C2-TRIM28 interaction is only tangentially related to Rad18. Coming after the model of MAGE-A4-Rad6-Rad18 complex in Fig 4G, the analysis of other MAGE proteins appears as an afterthought. The topic is underdeveloped and deserves (requires) more experimental validation.

In conclusion, Rad18 is a fascinating ubiquitin ligase but the study doesn't attempt to address the key unresolved questions about the stoichiometry of Rad6 binding and its specificity for monoubiquitination of PCNA. The biological aspects need to be expanded to justify the authors' conclusion that their "data reveal crucial insights into Rad18-mediated ubiquitination of PCNA and its regulation by MAGE-A4."

Specific points.

The displacement of Rad6 by MAGE-A4 binding Rad18 (Fig 2C) is not very convincing. Given the 60-fold higher affinity, it is surprising that a 5-fold excess of MAGE-A4 was needed.

What is the affinity of MAGE-A4 for intact Rad18? How does the complex behave on glycerol gradient. The "higher glycerol concentration" (line 220) of the MAGEA4/Rad18/Rad6 complex needs to be better documented. Analytical ultracentrifugation (AUC) analysis of the complexes might provide more accurate mass estimates and insights into their stoichiometry.

Does Rad18 sediment as a dimer? What is the stoichiometry of Rad6 binding? What is the affinity of Rad6 for full-length Rad18?

Does ubiquitin-charged Rad6 behave the same as the uncharged protein?

It is hard to assess / interpret the XL-MS experiments. Fig 2E shows most of the Rad6-Rad18 crosslinks are still detected (albeit at lower levels) in the presence of MAGE-A4.

The authors are unclear about whether a ternary complex exists. They suggest the absence of crosslinks detected between MAGE-A4 and Rad6 in Fig 2E is evidence of mutually exclusive binding but also state that "Rad6 does not dissociate completely from Rad18 in the presence of MAGEA4."

The decrease in vitro autoubiquitination of Rad18 by MAGE-A4 is clear in Fig 3A but only affects a portion of Rad18. A significant amount (1/3?) of unmodified Rad18 is visible in all conditions. Is this large enough to explain the biological effects of MAGE-A4 on Rad18? The authors should measure Rad18 levels or half-life in cells.

Why are Rad18 foci present in the absence of UV irradiation in Fig S6E?

The authors should use AlphaFold2 to predict the structure of the Rad18 dimer rather than the monomer.

The MS data should be deposited in a publicly accessible database. The di-Gly score should be defined (what does a score of 150 for K462 mean?). How do the scores compare with previously published autoubiquitination sites (Notenboom, NAR 2007)? Do the sites change when MAGEA4 is present or with R51E Rad18 mutant? Does Rad6 or MAGE-A4 become ubiquitinated?

Referee #2:

In this manuscript, the authors provide a biochemical and biophysical characterisation of the interaction between the human ubiquitin ligase (E3) Rad18 and the E3 regulatory protein MAGEA4. This interaction is critical for the regulation of replication stress tolerance and may have implications for cancer development. A rough picture was previously available, but the authors now provide a large data set with additional detail and develop a definitive hypothesis of how MAGEA4 regulates Rad18 activity. They propose that MAGEA4 competes with the E2, Rad6, for a binding site on Rad18, and the higher affinity of MAGEA4 compared to Rad6 leads to a blocking of Rad18 autoubiquitylation and thus stabilization of the E3. They support their hypothesis by a combination of AlphaFold prediction, NMR, cross-linking mass spectrometry and in vitro assays.

The topic is suitable for the journal and timely. Overall, the authors' hypothesis appears reasonable and may well apply. The manuscript is reasonably well written, and the referee appreciates the breadth of experimental approaches to support the mechanistic claims. That said, some of the most important pieces of evidence on which the authors' model rests are somewhat shaky. In addition, they use and propose a much too simplistic model of the Rad6-Rad18 complex and its actions on the substrate, PCNA. In my opinion, a considerable amount of work would be needed to put their evidence on to more solid ground.

My major concerns are as follows:

1. Rad18 ubiquitylates PCNA while both PCNA and Rad18 itself are bound to a primed piece of ss/dsDNA, where Rad18 is recruited partially via DNA binding and partially via interaction with the ssDNA-binding RPA complex. The domain arrangement of Rad18 and what is known about intramolecular domain-domain interactions (including the authors' own findings) suggest that multiple folded domains are interspersed with less structured connectors. The overall domain arrangement may therefore be quite variable. Thus, interaction with MAGEA4 and the effects of MAGEA4 on Rad18 activity and interaction with Rad6 could be very different depending on whether DNA is involved. This does not necessarily mean that the authors' conclusions are wrong, but it would be important to know whether the pool of Rad18 that is subject to autoubiquitylation, degradation, and/or MAGEA4-

mediated protection is soluble or bound to DNA and/or RPA. The authors should at least make an attempt to approach this important question by studying interactions and MAGEA4 effects also in the presence of DNA and RPA and using loaded PCNA as a ubiquitylation substrate.

2. The other major difference between the authors analysis and the physiological arrangement of Rad18 is its propensity to dimerise. The authors mention this briefly, but they do not appear to make any attempt to consider this in their analysis. I am not a structural biologist and can therefore not fully evaluate the biophysical assays that they use to make claims about domain-domain arrangements and interactions, but I suspect that they may not be able to differentiate between inter- and intramolecular interactions. I therefore do not understand the basis of their conclusions about longer-ranging interactions in their complex. For example, could the WH1 domain make contact with the 'other' Rad18 molecule in the dimer as opposed to the one engaged in interaction with the WH2 domain? There are multiple other analogous scenarios possible, including e.g. the interaction between the RING and SAP domains of Rad18.

3. I do not understand the authors' reasoning with respect to their XL-MS data (Figs. 2D-E and S5B-C). Firstly, they observe that the crosslinks between Rad6 and Rad18 are predominantly not within, but N-terminally of the R6B domain. Do they have an explanation for this? Secondly, in the presence of MAGEA4 the overall number of contacts between Rad6 and Rad18 is reduced, but the RELATIVE number of contacts between Rad6 and the R6B domain actually increases from 9/59 to 1/29. How can the authors conclude from these data that MAGEA4 competes with Rad6 for the R6B domain if the binding appears to become more strongly biased towards that region rather than shifting towards the RING domain?

4. It would strengthen the manuscript considerably if the authors could provide some in cellulo evidence for their model, e.g. an effect of relevant MAGEA4 mutations on Rad18 half-life, TLS efficiency, survival of DNA damage or any other read-out. An approach in cells might also help to clarify whether it is the chromatin-bound (and substrate-engaged) or the free pool of Rad18 that MAGEA4 acts upon.

5. The authors use pull-down assays to demonstrate a reduction in the interactions between Rad18 and MAGEA4 mutants. This is a very crude approach that should probably be complemented by a more quantitative method. The present approach leaves open the possibility that the overall folding and solubility of the mutants is affected.

6. Likewise, the sedimentation approach to show the competition between MAGEA4 and Rad6 is not very convincing. The authors might want to set up more sophisticated competition binding assays.

7. The observation that a human protein is less well produced in E. coli cells is insufficient to postulate a folding problem. It could be a translation efficiency/premature termination issue.

Minor points:

8. It is unclear how exactly the authors define cells with Rad18 foci. There doesn't seem to be an image of a transfected cell where Rad18 doesn't form foci. At least the authors' quantification method should be thoroughly explained (selection of data, thresholding, automated or manual counting, how many cells etc.).

9. The first paragraph is phrased a bit too simplistic; they might want to limit their discussion of E3 function to RING E3s, but they should probably be a bit more detailed with respect to those.

10. The authors should probably contrast the situation of human Rad18 to that in fungi, where regulation appears to be different. They also might want to call the human protein RAD18 rather than Rad18.

Referee #1 (Report for Author)

The manuscript by Griffith-Jones, Bhogaraju and colleagues describes a structure-activity study of the ubiquitin ligase Rad18 and its interaction with melanom-associated antigen 4 (MAGE-A4) a cancer-testis antigen and the ubiquitin E2 enzyme Rad6. The study uses AlphaFold2 models to design point mutations to disrupt the interactions and test the consequences in in vitro ubiquitination assays. Cross-linking mass spectrometry (XL-MS) and NMR spectroscopy are used to confirm and extend the AlphaFold2 models toward a model of the ternary complex. The major conclusions are that 1) a Rad6-binding domain of Rad18 binds to the C-terminal WH2 motif of MAGE-A4, 2) this causes a conformational change in Rad18 and affects Rad6 binding, 3) interactions between the RING and SAP domains of Rad18 are important for UV resistance in HEK293T cells, 4) MAGE-A4 specifically inhibits RAD18 autoubiquitination, and 5) the MAGE-A4 ubiquitin ligase-binding cleft is functional in other type-I MAGE proteins.

The major problem with the manuscript is the absence of results from cellular or animal models to validate MAGE-A4 regulation of Rad18 levels. There is no evidence to show that loss of the MAGE-A4-Rad18 interface has physiological consequences. The paper builds on a previous report (Gao, Nat Comm, 2016) that reported loss of MAGE-A4 decreased Rad18 half-life in cells and MAGE-A4 over-expression decreased Rad18 ubiquitination in cells. The authors of the current work need to extend their results beyond the demonstration of a decrease in Rad18 autoubiquitination. They show that an R51E mutation in Rad18 disrupts PCNA ubiquitination and the response of HEK293T cells to ionizing radiation (Fig 3F). It's unclear why they didn't use similar experiments to demonstrate the effects of the MAGE-A4-Rad18 interaction.

The work also suffers from being a collection of short vignettes rather than a complete story. The validation of the RING-SAP interface in Fig 3 is interesting but doesn't advance the narrative that MAGE-A4 regulates Rad18-mediated ubiquitination of PCNA. Similarly, the presentation of the MAGE-C2-TRIM28 interaction is only tangentially related to Rad18. Coming after the model of MAGE-A4-Rad6-Rad18 complex in Fig 4G, the analysis of other MAGE proteins appears as an afterthought. The topic is underdeveloped and deserves (requires) more experimental validation.

In conclusion, Rad18 is a fascinating ubiquitin ligase but the study doesn't attempt to address the key unresolved questions about the stoichiometry of Rad6 binding and its specificity for monoubiquitination of PCNA. The biological aspects need to be expanded to justify the authors' conclusion that their "data reveal crucial insights into Rad18-mediated ubiquitination of PCNA and its regulation by MAGE-A4."

Specific points.

The displacement of Rad6 by MAGE-A4 binding Rad18 (Fig 2C) is not very convincing. Given the 60-fold higher affinity, it is surprising that a 5-fold excess of MAGE-A4 was needed.

We thank the reviewer for this comment. This apparent discrepancy can be explained by the observations made by us and others. It is important to note that Rad6 interacts with two distant regions in Rad18-namely the RING domain and the Rad6-binding domain (R6BD) (1-3). The 60-fold difference is only when we consider the affinity measure through ITC (Figure 1F and Figure 2B) of interactions between Rad6-R6BD(Rad18) and MAGEA4-R6BD(Rad18). We also discuss as a prelude to the data in Figure 2D (Figure 2C in the previous version of the manuscript) in lines 242-245. In Figure 2D, we used FL Rad18, Rad6 and MAGEA4 proteins in the co-sedimentation assay.

To further strengthen this aspect of our data and complement our ITC experiments (Figure 1F and 2B) and glycerol gradient experiment (Figure 2D), we have performed a competition assay utilizing GST-tagged RAD18 R6BD immobilized on glutathione beads. We have loaded RAD6 onto the immobilized GST-RAD18 R6BD and subjected the complex to a 3x excess of MAGEA4 WT or MAGEA4 triple mutant (Figure 2C). We found that a notable amount of RAD6 dissociated in the presence of MAGEA4 WT, whereas the MAGEA4 triple mutant was unable to displace RAD6 from

the RAD18 R6BD. We found that MAGEA4 was still unable to fully displace RAD6 from the GST-tagged RAD18 R6BD (lines 233-242) (Figure 2C, S7). This was a bit surprising to us because in this case, we did not have the FL RAD18. We think this could be due to the GST tag that was fused to the R6BD which could have influenced the binding with MAGEA4.

This competition aspect is also intricately connected to the stoichiometry of the complex as this reviewer points out in the comment below. We also subjected both RAD18/RAD6 and RAD18/RAD6/MAGEA4 to AUC experiments and we obtained a sedimentation coefficient s_{20w} of 6.2S for RAD18/RAD6 (Figure S9, lines 288-293). However, we were unable to measure the molecular weight of the RAD18/RAD6/MAGEA4 complex accurately through AUC, likely due to the dynamic nature of the complex. Therefore, we employed Mass Photometry, which is another biophysical method to measure the size of macromolecules very accurately. Using mass photometry, we obtained a Mw of 171 kDa for RAD18/RAD6/MAGEA4 (Figure 2G) (lines 295-304). This indicates that we have 2 molecules of His-tagged RAD18 ($58.6 \times 2 = 114$ kDa), 1 molecule of RAD6 (17.3 kDa) and 1 molecule of His-tagged MAGEA4 (37.3 kDa) in this complex.

What is the affinity of MAGE-A4 for intact Rad18? How does the complex behave on glycerol gradient. The "higher glycerol concentration" (line 220) of the MAGEA4/Rad18/Rad6 complex needs to be better documented. Analytical ultracentrifugation (AUC) analysis of the complexes might provide more accurate mass estimates and insights into their stoichiometry.

- We thank the reviewer for this comment. Full length RAD18 is insoluble when expressed without RAD6 or MAGEA4. So it was not possible for us to measure the affinity between MAGEA4 and FL RAD18. But in our ITC experiment shown in Figure 1F, we measure the affinity between the FL MAGEA4 and the peptide of RAD18 which binds to it and determined the K_d as 405 nM. Since we show that this part of RAD18 is the main region interacting with MAGEA4 in Figure 1E, we expect the affinity between FL MAGEA4 and Rad18 to be similar to the one we measured using ITC. The term "higher concentration" has moved to the line 251-252 in the revised manuscript and here we now added the glycerol percentages where the molecules sedimented to avoid the vagueness in the statement.
- We were able to measure the molecular weight of the RAD18/RAD6/MAGEA4 complex accurately using Mass Photometry (Figure 2G) and we determined a Mw of 171 kDa, which is consistent with a tetrameric complex containing two His-tagged RAD18s (58.6×2 kDa), one RAD6 (17.3 kDa) and one His-tagged MAGEA4 (37.3 kDa) (Figure 2G) (lines 295-304).

Does Rad18 sediment as a dimer? What is the stoichiometry of Rad6 binding? What is the affinity of Rad6 for full-length Rad18?

- We thank the reviewer for these questions. RAD18 dimerization was shown previously (4, 5). The RING domain was shown to be responsible for the dimerization of RAD18 later (1). The stoichiometry of RAD18 and RAD6 has been a debated issue because earlier studies (1) indicated that RAD18/RAD6 is a tetramer with two molecules of each present in the complex. But later, it was shown that two FL molecules of RAD18 only bind to one RAD6 monomer (2, 3). The affinity of FL RAD18 to FL RAD6 has not been measured by us or anyone previously, since RAD18 is insoluble without RAD6 and the purification of RAD18 FL separately from RAD6 is necessary to measure the affinity between the two. However, we have measured the affinity of the R6B domain of RAD18 to RAD6 which is 25 μ M (Figure 2B) and previously, Huang et al., have measured the affinity of RAD18 RING domain with RAD6, which measured at a K_d of 35 μ M (2).
- We used Mass Photometry to assess the stoichiometry of our purified RAD18/RAD6 and RAD18/RAD6/MAGEA4 complexes. Our Mass Photometry data indicates that RAD18/RAD6 is a hetero trimer with a measured molecular mass of 128 kDa (Figure 2G). This supports the notion that there are two RAD18 molecules and one RAD6 in the RAD18/RAD6 complex. Our Mass Photometry of RAD18/RAD6/MAGEA4 indicates that RAD18 exists as a dimer in this complex, such that dimeric RAD18 binds to one RAD6 and one MAGEA4 (Figure 2G, lines 295-304). We also performed AUC experiments on

RAD18/RAD6 and found that RAD18 does indeed sediment as a dimer, in line with previous findings as described above. We determined a molecular weight of 131 kDa for RAD18/RAD6, supporting the notion that RAD18 is an asymmetric dimer capable of binding to only one RAD6 molecule (Figure S9, lines 288-293).

Does ubiquitin-charged Rad6 behave the same as the uncharged protein?

- We thank the reviewer for this interesting question. We think RAD6 charged with Ub would have more affinity towards RAD18, since ubiquitin ligases have been shown to interact with both E2 and Ubiquitin during ubiquitination reactions (6). But this is incredibly difficult to measure and in fact it has not been measured for any E2-E3 interaction that we know of. We hope that the reviewer agrees that this aspect, although very interesting, falls outside the scope of our current manuscript.

It is hard to assess / interpret the XL-MS experiments. Fig 2E shows most of the Rad6-Rad18 crosslinks are still detected (albeit at lower levels) in the presence of MAGE-A4.

- We thank the reviewer for raising this issue, we realize that our XL-MS data explanation needs more clarity. So we have included a discussion on especially the non-quantitative nature of the experiment plus how the XL-MS will not tell us whether the cross-links between especially MAGEA4 and RAD18 are from the same molecule of RAD18 or the second molecule in the RAD18 as pointed out by Reviewer 2. Now the XL-MS data is only used to assess which regions of the proteins interact with each other and support our AF predictions and previous findings. We have included these points in lines 274-277, 279-282 and 485-487 of the revised manuscript.

The authors are unclear about whether a ternary complex exists. They suggest the absence of crosslinks detected between MAGE-A4 and Rad6 in Fig 2E is evidence of mutually exclusive binding but also state that "Rad6 does not dissociate completely from Rad18 in the presence of MAGEA4."

- We thank the reviewer for this comment. We would like to draw the attention of the reviewer to the fact that RAD18 has two binding sites to RAD6 (1-3). So even if MAGEA4 displaces RAD6 from R6BD region of RAD18, RAD6 will still bind to the RING domain of RAD18 in a MAGEA4 independent manner. This would explain how RAD6 and MAGEA4 binding to the R6B site of RAD18 is mutually exclusive but they can still be binding to two different sites of RAD18. In the lines 227-229, 242-245, 255-257 we worded exactly this point carefully. Additionally, we have performed Mass Photometry on RAD18/RAD6/MAGEA4 and shown that the complex has a molecular weight of 171 kDa, indicating that two RAD18s, one RAD6 and one MAGEA4 are present within the complex (Figure 2G, lines 295-304).

The decrease in vitro autoubiquitination of Rad18 by MAGE-A4 is clear in Fig 3A but only affects a portion of Rad18. A significant amount (1/3?) of unmodified Rad18 is visible in all conditions. Is this large enough to explain the biological effects of MAGE-A4 on Rad18? The authors should measure Rad18 levels or half-life in cells.

We thank the reviewer for this comment. We agree that only cell-based measurements would answer this question. To measure RAD18 levels directly in cells, we ectopically co-expressed both HA-RAD18 and GFP-MAGEA4 WT or triple mutant in HEK293T cells (Figure 4C). We found that both the endogenous RAD18 levels and HA-RAD18 levels were increased in cells transfected with MAGEA4 WT, compared to cells transfected with MAGEA4 triple mutant or GFP-alone. Although MAGEA4 triple mutant expressed comparatively less well compared to MAGEA4 WT in our experiments, RAD18 levels are much lower in conditions where MAGEA4 triple mutant was expressed compared to the GFP only control, therefore, our experiment is able to show that the MAGEA4 interaction is important for RAD18 stabilization

in cells. We also performed the same experiment without the exogenous expression of HA-RAD18 and monitored the endogenous RAD18 levels (Figure S13). This experiment resulted in similar findings. These results are now described in lines 389-396.

Why are Rad18 foci present in the absence of UV irradiation in Fig S6E?

- We thank the reviewer for this comment. In the absence of UV, most of the RAD18 is still nuclear and localizes to large puncta with only a partial co-localization with PCNA. And upon UV treatment, RAD18 is dispersed into smaller puncta and exhibits perfect co-localization with PCNA. This has been previously observed as well (7) and is also found in the RAD18 antibody manufacturers' page here.

The authors should use AlphaFold2 to predict the structure of the Rad18 dimer rather than the monomer.

- We thank the reviewer for this comment. We have now predicted RAD18 dimer using Alphafold and used that for our analysis of RING-SAP interaction. (Figure S1).

The MS data should be deposited in a publicly accessible database. The di-Gly score should be defined (what does a score of 150 for K462 mean?). How do the scores compare with previously published autoubiquitination sites (Notenboom, NAR 2007)? Do the sites change when MAGEA4 is present or with R51E Rad18 mutant?

- The diGly scores represent the Mascot score for peptide identification, our mass spectrometry experts have decided on a diGly score of 32 as the cut-off for a reliable database match of the MS2 spectra for a modified peptide We have added this definition to the materials and methods section and we have submitted all our MS data to the PRIDE database with the following identifiers: PXD047850 for the diGly experiment, PXD047849 for the Comparative interaction proteomics experiment with MAGE-C2 and PDX047929 for the XL-MS experiment. We have added these identifiers to the data availability section. We have compared the ubiquitination sites that we identified to those already reported in the literature (lines 401-406).
- Reviewer login accounts for access to the MS data (currently private until online publication of the manuscript) are as follows:
Comparative Interaction Proteomics with MAGE-C2
-Username: reviewer_pxd047849@ebi.ac.uk
-Password: dNgTCzPI
diGly experiment for identifying sites of ubiquitination:
-Username: reviewer_pxd047850@ebi.ac.uk
-Password: WZdmB0vC
XL-MS experiment:
-Username: reviewer_pxd047929@ebi.ac.uk
-Password: sHV3wrv4

Does Rad6 or MAGE-A4 become ubiquitinated?

- We do not observe any notable ubiquitination of RAD6 or MAGEA4 in our experiments (Figure 4).

Referee #2 (Report for Author)

In this manuscript, the authors provide a biochemical and biophysical characterisation of the interaction between the human ubiquitin ligase (E3) Rad18 and the E3 regulatory protein MAGEA4. This interaction is critical for the regulation of replication stress tolerance and may have implications for cancer development. A rough picture was previously available, but the authors now provide a large data set with additional detail and develop a definitive hypothesis of how MAGEA4 regulates Rad18

activity. They propose that MAGEA4 competes with the E2, Rad6, for a binding site on Rad18, and the higher affinity of MAGEA4 compared to Rad6 leads to a blocking of Rad18 autoubiquitylation and thus stabilization of the E3. They support their hypothesis by a combination of AlphaFold prediction, NMR, cross-linking mass spectrometry and *in vitro* assays.

The topic is suitable for the journal and timely. Overall, the authors' hypothesis appears reasonable and may well apply. The manuscript is reasonably well written, and the referee appreciates the breadth of experimental approaches to support the mechanistic claims. That said, some of the most important pieces of evidence on which the authors' model rests are somewhat shaky. In addition, they use and propose a much too simplistic model of the Rad6-Rad18 complex and its actions on the substrate, PCNA. In my opinion, a considerable amount of work would be needed to put their evidence on to more solid ground.

My major concerns are as follows:

1. Rad18 ubiquitylates PCNA while both PCNA and Rad18 itself are bound to a primed piece of ss/dsDNA, where Rad18 is recruited partially via DNA binding and partially via interaction with the ssDNA-binding RPA complex. The domain arrangement of Rad18 and what is known about intramolecular domain-domain interactions (including the authors' own findings) suggest that multiple folded domains are interspersed with less structured connectors. The overall domain arrangement may therefore be quite variable. Thus, interaction with MAGEA4 and the effects of MAGEA4 on Rad18 activity and interaction with Rad6 could be very different depending on whether DNA is involved. This does not necessarily mean that the authors' conclusions are wrong, but it would be important to know whether the pool of Rad18 that is subject to autoubiquitylation, degradation, and/or MAGEA4-mediated protection is soluble or bound to DNA and/or RPA. The authors should at least make an attempt to approach this important question by studying interactions and MAGEA4 effects also in the presence of DNA and RPA and using loaded PCNA as a ubiquitylation substrate.

- We thank the reviewer for this important comment. It is indeed true that RAD18-mediated ubiquitination of PCNA itself is heavily dependent on the DNA and RPA coating the single stranded DNA. However, it was shown by Gao et al., 2018 (Figure 3C in the reference) that MAGEA4 always remains soluble irrespective of the UV treatment, whereas RAD18 forms clear nuclear puncta upon UV irradiation of the cells (8). This showed that MAGEA4 affects the soluble form of Rad18 which is not interacting with chromatin. **This is now mentioned in lines 372-374.** Moreover, it was also shown that auto ubiquitinated or ubiquitin fused Rad18 stays soluble even under UV treatment conditions, indicating Rad18 ubiquitination also likely happens in the soluble form and retains the molecule there (9) (Figure 7 in Zeman et al., 2014).
- We performed the *in vitro* ubiquitination assays in the presence of DNA to see how the RAD18 interaction with the DNA affects the binding of MAGEA4 to Rad18. **We have performed an *in vitro* ubiquitination assay in the presence of single stranded DNA (RAD18 was shown to bind to single stranded DNA more tightly than other forms (10) and found that MAGEA4 still offered protection of RAD18 from autoubiquitination, indicating that DNA does not have any notable impact on MAGEA4 binding to RAD18 (Figure S12, lines 376-380).**
- To measure RAD18 levels directly in cells, we ectopically co-expressed both HA-RAD18 and GFP-MAGEA4 WT or triple mutant in HEK293T cells (Figure 4C). We found that the levels of soluble endogenous RAD18 and the overexpressed HA-RAD18 levels were increased in cells transfected with MAGEA4 WT, compared to cells transfected with MAGEA4 triple mutant or GFP-alone. Although the MAGEA4 triple mutant expressed comparatively less well compared to MAGEA4 WT in our experiments, RAD18 levels are much lower in conditions where MAGEA4 triple mutant was expressed compared to the GFP only control, therefore, our experiment is able to show that the MAGEA4 interaction is important for RAD18 stabilization in cells. We also performed the same experiment without the exogenous expression of HA-RAD18 and monitored the endogenous RAD18 levels (Figure S13). This experiment resulted in similar findings. These results are now described in lines (389-396).

2. The other major difference between the authors analysis and the physiological arrangement of Rad18 is its propensity to dimerise. The authors mention this briefly, but they do not appear to make any attempt to consider this in their analysis. I am not a structural biologist and can therefore not fully evaluate the biophysical assays that they use to make claims about domain-domain arrangements and interactions, but I suspect that they may not be able to differentiate between inter- and intramolecular interactions. I therefore do not understand the basis of their conclusions about longer-ranging interactions in their complex. For example, could the WH1 domain make contact with the 'other' Rad18 molecule in the dimer as opposed to the one engaged in interaction with the WH2 domain? There are multiple other analogous scenarios possible, including e.g. the interaction between the RING and SAP domains of Rad18.

- We thank the reviewer for this comment. Indeed, these interactions between MAGEA4 and the “other” Rad18 molecule are possible explanations for some of the XL-MS data that we have in the manuscript. We have included these points in the revised manuscript, in lines 274-277, 326-329 and 485-487 to shed light on these possible inter and intra molecular interactions. But we hope the reviewer agrees that these possible alternative interactions of WH-A of MAGEA4 do not change any major conclusions of the paper.

3. I do not understand the authors' reasoning with respect to their XL-MS data (Figs. 2D-E and S5B-C). Firstly, they observe that the crosslinks between Rad6 and Rad18 are predominantly not within, but N-terminally of the R6B domain. Do they have an explanation for this? Secondly, in the presence of MAGEA4 the overall number of contacts between Rad6 and Rad18 is reduced, but the RELATIVE number of contacts between Rad6 and the R6B domain actually increases from 9/59 to 1/29. How can the authors conclude from these data that MAGEA4 competes with Rad6 for the R6B domain if the binding appears to become more strongly biased towards that region rather than shifting towards the RING domain?

We thank the reviewer for this comment.

- We could think of a few possible reasons why RAD6 predominantly gets crosslinked to just the N-terminus of R6B domain. In Figure 1A and Figure 2E we labelled the domain boundaries (for R6B it is 340-395) based on previous publications (1), however, based on the AF predictions, it is likely that R6B is from 330-372. We have now changed all the domain boundary labels consistent with AF predictions, aside from the RING domain and ZnF which were already well defined experimentally (2, 11). This reassignment of R6B domain boundaries explains some of the cross-links (Figure. 2) The R6B domain is extremely small (<10%) compared to the size of RAD18 and, moreover, in the RAD18/RAD6 complex, the R6B domain is buried at the interaction interface (12). So the likelihood of forming chemical cross-links is low when the residues are already engaged in a tight interaction. Whereas the region N-terminus to the R6B domain of Rad18 is predicted to be flexible and can easily be in close contact with Rad18 and more importantly available for cross-linking. We have added this point in lines 265-268 to accurately describe our data.
- For the second part of the question, we consulted our mass spectrometry expert colleagues to find an answer. And it seems that since these XL-MS experiments using RAD18/RAD6 complexes with and without MAGEA4 are performed separately and analyzed independently in mass spectrometry, it is not recommended to compare this fraction of cross-links found in one experiment vs another. So, we have removed one instance of comparison (lines 238-241 in the previous version of the manuscript) where we talked about the overall reduced number of cross-links between Rad18/Rad6 in the presence of MAGEA4. In addition, we have included the non-quantitative nature of our mass spectrometry data in lines 279-282. We also understand that the placement of numbers representing the number of cross-links in panels E and F of Figure 2 could lead to direct comparisons even if it is not done in the text. Therefore, we have replaced these panels with diagrams showing all cross-links identified (previously in Figure S5).

4. It would strengthen the manuscript considerably if the authors could provide some in cellulo evidence for their model, e.g. an effect of relevant MAGEA4 mutations on Rad18 half-life, TLS efficiency, survival of DNA damage or any other read-out. An approach in cells might also help to clarify whether it is the chromatin-bound (and substrate-engaged) or the free pool of Rad18 that MAGEA4 acts upon.

We thank the reviewer for this suggestion. To measure RAD18 levels directly in cells, we ectopically co-expressed both HA-RAD18 and GFP-MAGEA4 WT or triple mutant in HEK293T cells (Figure 4C). We found the levels of soluble endogenous RAD18 and the overexpressed HA-RAD18 levels were increased in cells transfected with MAGEA4 WT, compared to cells transfected with MAGEA4 triple mutant or GFP-alone. Although the MAGEA4 triple mutant expressed comparatively less well compared to MAGEA4 WT in our experiments, RAD18 levels are much lower in conditions where MAGEA4 triple mutant was expressed compared to the GFP only control, therefore, our experiment is able to show that the MAGEA4 interaction is important for RAD18 stabilization in cells. We also performed the same experiment without the exogenous expression of HA-RAD18 and monitored the endogenous RAD18 levels (Figure S13). This experiment resulted in similar findings. These results are now described in lines (389-396). We direct the reviewer to please also refer to our response to the first question regarding the effect of MAGEA4 on chromatin-bound or soluble RAD18.

5. The authors use pull-down assays to demonstrate a reduction in the interactions between Rad18 and MAGEA4 mutants. This is a very crude approach that should probably be complemented by a more quantitative method. The present approach leaves open the possibility that the overall folding and solubility of the mutants is affected.

- We thank the reviewer for this comment. We would like to draw reviewer's attention to the isothermal titration calorimetry (ITC) data presented in Figure 1F, G and Figure S4 (previously S2) where we quantified the affinity between MAGEA4 WT and mutants and Rad18 R6B.

6. Likewise, the sedimentation approach to show the competition between MAGEA4 and Rad6 is not very convincing. The authors might want to set up more sophisticated competition binding assays.

- We thank the reviewer for this comment. We have performed Mass Photometry on RAD18/RAD6/MAGEA4 and determined a molecular weight of 171 kDa, in line with two RAD18s, one RAD6 and one MAGEA4 (Figure 2G, lines 295-304). Although we were able to analyse RAD18/RAD6 by AUC, we found that the dynamic nature of the RAD18/RAD6/MAGEA4 complex rendered the complex unsuitable for analysis by AUC. We performed alternate competition assays where we immobilized the GST-tagged R6B domain of RAD18 and first incubated with RAD6 and then MAGE-A4 (Figure 2C, Figure S7, lines 233-242). This competition assay also showed that WT MAGEA4 is able to outcompete RAD6 but not the triple mutant MAGEA4, which is deficient in binding to RAD18.

7. The observation that a human protein is less well produced in E. coli cells is insufficient to postulate a folding problem. It could be a translation efficiency/premature termination issue.

We agree with this comment, we purified the MAGEA4 L121/122A mutant and found that the protein eluted in the void volume during size exclusion chromatography, indicating that the protein is aggregated (Figure S5), likely due to misfolding from mutating these residues buried within the core of the MHD. Additionally, we have analyzed the crystal structures of MAGEA3, MAGEA11 and MAGEB1 and found that the dileucine motif of these MAGEs also lies within the hydrophobic core of the MHD, indicating that mutating the dileucine motif of these proteins likely also results in instability of the MHD (Figure S5). The corresponding lines in the manuscript are 185-198.

Minor points:

8. It is unclear how exactly the authors define cells with Rad18 foci. There doesn't seem to be an image of a transfected cell where Rad18 doesn't form foci. At least the authors' quantification method should be thoroughly explained (selection of data, thresholding, automated or manual counting, how many cells etc.).

We thank the reviewer for this comment. In response to DNA damage, many proteins of the DNA damage response pathways e.g. BRCA1 form distinct nuclear puncta (13) which have been termed ionizing radiation induced foci (IRIF). RAD18 was also shown to form small IRIF upon exposure to UV radiation, since RAD18 is recruited to sites of DNA damage (14). We observe these IRIF clearly in Figure 3F with RAD18 WT but much less so with RAD18 R51E. In the absence of UV radiation, RAD18 still forms nuclear puncta but these are larger and fewer in number, since RAD18 is not chromatin-bound. In Figure S11C we show HEK cells transfected with HA-RAD18 WT in the absence of UV and we can see that the RAD18 puncta are more diffuse. We agree that this needed to be further clarified within the text and we do so in lines 341-344. We have performed a manual blinded counting of cells with foci. For the statistical analysis comparing IRIF following exposure to UV, we collected 8 fields of view each for HA-RAD18 WT and HA-RAD18 R51E. The total number of transfected cells counted was 63 for HA-RAD18 WT and 66 for HA-RAD18 R51E. For each image, we calculated the percentage of cells displaying the distinct RAD18 IRIF and performed an unpaired T test using GraphPad Prism to compare the percentage difference between the two conditions. We have added these details to the methods section of the revised manuscript.

9. The first paragraph is phrased a bit too simplistic; they might want to limit their discussion of E3 function to RING E3s, but they should probably be a bit more detailed with respect to those.

- We have made the first paragraph more detailed (lines 70-76).

10. The authors should probably contrast the situation of human Rad18 to that in fungi, where regulation appears to be different. They also might want to call the human protein RAD18 rather than Rad18.

- We thank the reviewer for this suggestion, we have changed the text accordingly.

References:

1. V. Notenboom, R. G. Hibbert, S. E. Van Rossum-Fikkert, J. V. Olsen, M. Mann, T. K. Sixma, Functional characterization of Rad18 domains for Rad6, ubiquitin, DNA binding and PCNA modification. *Nucleic Acids Research* **35**, 5819–5830 (2007).
2. A. Huang, R. G. Hibbert, R. N. De Jong, D. Das, T. K. Sixma, R. Boelens, Symmetry and Asymmetry of the RING–RING Dimer of Rad18. *Journal of Molecular Biology* **410**, 424–435 (2011).
3. Y. Masuda, M. Suzuki, H. Kawai, F. Suzuki, K. Kamiya, Asymmetric nature of two subunits of RAD18, a RING-type ubiquitin ligase E3, in the human RAD6A–RAD18 ternary complex. *Nucleic Acids Research* **40**, 1065–1076 (2012).
4. H. D. Ulrich, Two RING finger proteins mediate cooperation between ubiquitin-conjugating enzymes in DNA repair. *The EMBO Journal* **19**, 3388–3397 (2000).

5. S. Miyase, S. Tateishi, K. Watanabe, K. Tomita, K. Suzuki, H. Inoue, M. Yamaizumi, Differential Regulation of Rad18 through Rad6-dependent Mono- and Polyubiquitination. *Journal of Biological Chemistry* **280**, 515–524 (2005).
6. A. Plechanovová, E. G. Jaffray, M. H. Tatham, J. H. Naismith, R. T. Hay, Structure of a RING E3 ligase and ubiquitin-loaded E2 primed for catalysis. *Nature* **489**, 115–120 (2012).
7. K. Watanabe, S. Tateishi, M. Kawasuji, T. Tsurimoto, H. Inoue, M. Yamaizumi, Rad18 guides poleta to replication stalling sites through physical interaction and PCNA monoubiquitination. *EMBO J* **23**, 3886–3896 (2004).
8. Y. Gao, E. Mutter-Rottmayer, A. M. Greenwalt, D. Goldfarb, F. Yan, Y. Yang, R. C. Martinez-Chacin, K. H. Pearce, S. Tateishi, M. B. Major, C. Vaziri, A neomorphic cancer cell-specific role of MAGE-A4 in trans-lesion synthesis. *Nat Commun* **7**, 12105 (2016).
9. M. K. Zeman, J.-R. Lin, R. Freire, K. A. Cimprich, DNA damage-specific deubiquitination regulates Rad18 functions to suppress mutagenesis. *Journal of Cell Biology* **206**, 183–197 (2014).
10. Y. Tsuji, K. Watanabe, K. Araki, M. Shinohara, Y. Yamagata, T. Tsurimoto, F. Hanaoka, K. Yamamura, M. Yamaizumi, S. Tateishi, Recognition of forked and single-stranded DNA structures by human RAD18 complexed with RAD6B protein triggers its recruitment to stalled replication forks. *Genes Cells* **13**, 343–354 (2008).
11. A. A. Rizzo, P. E. Salerno, I. Bezsonova, D. M. Korzhnev, NMR Structure of the Human Rad18 Zinc Finger in Complex with Ubiquitin Defines a Class of UBZ Domains in Proteins Linked to the DNA Damage Response. *Biochemistry* **53**, 5895–5906 (2014).
12. R. G. Hibbert, A. Huang, R. Boelens, T. K. Sixma, E3 ligase Rad18 promotes monoubiquitination rather than ubiquitin chain formation by E2 enzyme Rad6. *Proc. Natl. Acad. Sci. U.S.A.* **108**, 5590–5595 (2011).
13. T. T. Paull, E. P. Rogakou, V. Yamazaki, C. U. Kirchgessner, M. Gellert, W. M. Bonner, A critical role for histone H2AX in recruitment of repair factors to nuclear foci after DNA damage. *Current Biology* **10**, 886–895 (2000).
14. K. Watanabe, K. Iwabuchi, J. Sun, Y. Tsuji, T. Tani, K. Tokunaga, T. Date, M. Hashimoto, M. Yamaizumi, S. Tateishi, RAD18 promotes DNA double-strand break repair during G1 phase through chromatin retention of 53BP1. *Nucleic Acids Research* **37**, 2176–2193 (2009).

Dr. Sagar Bhogaraju
EMBL Grenoble
Grenoble
France

30th Jan 2024

Re: EMBOJ-2023-115024R
Insights into The Role of MAGEA4 in Rad18 Regulation: Implications across the MAGE Protein Family

Dear Sagar,

Thank you for submitting your revised manuscript, as well as for your patience during its re-review. With some delay due to busy reviewers at the start of the year, we have now finally received the comments from one of the original referees. As you will see from the report copied below, the reviewer is generally supportive of publication, but still has some queries that I would ask you to respond to in a final point-by-point response letter prior to publication. In addition, there are a number of editorial issues that need to be addressed at this stage:

1) Importantly, please pay careful attention to our Guide to Authors and reformat the whole submission accordingly:

- Please reorganize the figures - there are currently only 5 main figures (i.e. "level 1", of which we could have 7-8) and 16 "supplementary" figures, which no longer exist at EMBO Press (see www.embopress.org/page/journal/14602075/authorguide#expandedview for more information). We allow up to FIVE EXPANDED VIEW FIGURES (i.e. "level 2" - referenced as "FIGURE EV1/2/3...") that would also be type-set and directly accessible in the HTML version; while the remainder should be collated in a single APPENDIX PDF (headed by a Table Of Contents) as "APPENDIX FIGURE S1/2/3..." (i.e. 3rd level of importance).
- All main and EV figures need to be uploaded as individual files with sufficient resolution/quality for production.
- "Supplementary Tables" 1 and 2 should be converted/renamed (both in the file and in the text) to EXPANDED VIEW DATASETS (call-out: "Dataset EV1/2"), and the spreadsheets each need a separate "Legend" tab containing dataset title and legend information
- "Supplementary Tables" 1 and 2 should become APPENDIX TABLES - being moved together with all Appendix figures into a single Appendix PDF, and listed in its ToC. The should be renamed into "Appendix Table S1/2" (please make sure to also update all in-text references accordingly), and have their respective title/legend above then
- Please adjust the format of the reference list and of the in-text citations according to EMBO Journal format (alphabetical order, author name et al + year.../up to 10 author names in the reference list before et al / please refer to our Guide to Authors for additional information on EMBO J reference format). Also, please adjust the format for citation of preprints as specified in our author guidelines. The citation in the text should be: "(preprint: NAME1 et al, YEAR)"; in the reference list: "Author NAME1, Author NAME2, ... (YEAR) article title. bioRxiv doi: XXX".

2) Other presentational issues:

- On the abstract page of the manuscript, please include 4-5 general keyword terms to enhance searchability.
- Please rename the Conflict of Interest section into "Disclosure and Competing Interests Statement", in accordance with our updated Guide to Authors (<https://www.embopress.org/competing-interests>)
- As we are switching from a free-text author contribution statement towards a more formal statement based on Contributor Role Taxonomy (CRediT) terms, please remove the present Author Contribution section and instead specify each author's contribution(s) directly in the Author Information page of our submission system during upload of the final manuscript. See <https://casrai.org/credit/> for more information.
- Please remove review access information, and ensure that data listed in the Data Availability section becomes publicly accessible at this point, latest upon formal acceptance.
- Please double-check to make sure to all relevant funding information in the manuscript is congruent with the info entered into our submission system. Some grants are currently only mentioned in the text but not in the system: FRISBI (ANR-10-INBS-0005-02) and GRAL; University Grenoble Alpes graduate school (Écoles Universitaires de Recherche) CBH-EUR-GS (ANR-17-EURE-0003)
- To ensure conciseness and explicitness of the title, I would propose the following altered title (happy to discuss): "Structural basis of RAD18 regulation by MAGEA4 and it implications for ubiquitin ligase binding by MAGE family proteins"

3) Source Data:

- Please upload the Source Data file folders, which are currently all combined in a single ZIP archive, as one separate archive per each main figure; plus one combined ZIP archive for all "Expanded View Figure" source data: and another combined ZIP for all "Appendix Figure" source data.

4) Synopsis material:

- Please provide suggestions for a short 'blurb' text prefacing and summing up the conceptual aspect of the study in two sentences (max. 250 characters), followed by 3-5 one-sentence 'bullet points' with brief factual statements of key results of the paper; they will form the basis of an editor-written 'Synopsis' accompanying the online version of the article. Please also upload a synopsis image, which can be used as a "visual title" for the synopsis section of your paper. The image should be in PNG or JPG format, and please make sure that it remains in the modest dimensions of (exactly) 550 pixels wide and 300-600 pixels high.

I am therefore returning the manuscript to you for a final round of minor revision, to allow you to make these adjustments and upload all modified files. Once we will have received them, we should hopefully be ready to swiftly proceed with formal acceptance and production of the manuscript.

With kind regards,

Hartmut Vodermaier

9) Digital image enhancement is acceptable practice, as long as it accurately represents the original data and conforms to community standards. If a figure has been subjected to significant electronic manipulation, this must be clearly noted in the figure legend and/or the 'Materials and Methods' section. The editors reserve the right to request original versions of figures and the original images that were used to assemble the figure. Finally, we generally encourage uploading of numerical as well as gel/blot

image source data; for details see: embopress.org/page/journal/14602075/authorguide#sourcedata

At EMBO Press, we ask authors to provide source data for the main manuscript figures. Our source data coordinator will contact you to discuss which figure panels we would need source data for and will also provide you with helpful tips on how to upload and organize the files.

In the interest of ensuring the conceptual advance provided by the work, we recommend submitting a revision within 3 months (29th Apr 2024). Please discuss the revision progress ahead of this time with the editor if you require more time to complete the revisions. Use the link below to submit your revision:

Link Not Available

Referee #2:

The authors have addressed the reviewers' comments in detail in their written statement. They have provided additional explanations and performed some additional experiments. Overall, this has strengthened some of their conclusions. I still found it a bit unsatisfying that some of their model is based more on conjecture and alternatives haven't been rigorously excluded.

Examples:

- The authors now show RAD18 stability data in cells - it would have been nice to see some 'physiological' effects, e.g. on damage resistance or mutagenesis.
- Intra- and intermolecular interactions cannot be distinguished in their XL-MS experiments. Could this not be done by mixing two differentially isotope-labelled versions of the same proteins?

Although the manuscript is not one of the strongest based on the uncertainty of some of the conclusions, I find the topic still interesting and the data potentially valid.

Referee #2:

The authors have addressed the reviewers' comments in detail in their written statement. They have provided additional explanations and performed some additional experiments. Overall, this has strengthened some of their conclusions. I still found it a bit unsatisfying that some of their model is based more on conjecture and alternatives haven't been rigorously excluded.

We thank the reviewer for this comment. A minor part of our model of how MAGE-A4 contributes to the stability of RAD18 indeed is uncertain, however, we clearly indicated this uncertainty in the model itself. For eg. We have shown two possibilities of how MAGE-A4 binds to one of the RAD18 molecules in the RAD18 dimer (Figure 5H). Text lines 356-366, 387-393 and 409-412 also describe the uncertainty concerning MAGE-A4 binding to RAD18 in the context of full-length complexes of RAD18/RAD6.

Examples:

- The authors now show RAD18 stability data in cells - it would have been nice to see some 'physiological' effects, e.g. on damage resistance or mutagenesis.

We thank the reviewer for this comment. We agree that having additional physiological significance of our findings would increase the impact. The role of RAD18 levels in damage resistance and mutagenesis has been clearly demonstrated previously and we hope the reviewer agrees that our data provides the molecular and structural basis for these physiological observations. We are continuing to pursue the effects of increased stability of RAD18 and its effects especially in select cancer cell types.

- Intra- and intermolecular interactions cannot be distinguished in their XL-MS experiments. Could this not be done by mixing two differentially isotope-labelled versions of the same proteins?

We thank the reviewer for this comment. This is in principle a valid approach in differentiating between inter and intra molecular cross links. But RAD18 is an obligate dimer, so we cannot purify and mix two differently labelled monomers. We are currently working to obtain a cryo-EM structure of the full-length RAD18/RAD6 complexes together with MAGEA4 and PCNA, which will conclusively address this aspect of our study.

Although the manuscript is not one of the strongest based on the uncertainty of some of the conclusions, I find the topic still interesting and the data potentially valid.

We thank the reviewer for supporting the publication of our work.

Dr. Sagar Bhogaraju
EMBL Grenoble
Grenoble
France

7th Feb 2024

Re: EMBOJ-2023-115024R1

Structural basis for RAD18 regulation by MAGEA4 and its implications for RING ubiquitin ligase binding by MAGE family proteins

Dear Sagar,

Thank you for submitting your final revised manuscript for our consideration. I am pleased to inform you that we have now accepted it for publication in The EMBO Journal.

With kind regards,

Hartmut
